# Over-parameterised Shallow Neural Networks with Asymmetrical Node Scaling: Global Convergence Guarantees and Feature Learning

**François Caron**                                    *caron@stats.ox.ac.uk*
*Department of Statistics*
*University of Oxford, United Kingdom*

**Fadhel Ayed**                                    *fadhel.ayed@gmail.com*
*Huawei Technologies*
*Paris, France*

**Paul Jung**                                    *paul.jung@gmail.com*
*Department of Mathematics*
*Fordham University, USA*

**Hoil Lee**                                    *hoil.lee@alumni.kaist.ac.kr*
*Samsung SDS*
*South Korea*

**Juho Lee**                                    *juholee@kaist.ac.kr*
*Kim Jaechul Graduate School of AI*
*KAIST, South Korea*

**Hongseok Yang**                                    *hongseok.yang@kaist.ac.kr*
*School of Computing*
*KAIST, South Korea*

**Reviewed on OpenReview:** *https://openreview.net/forum?id=Sx1khIIi95*

## Abstract

We consider gradient-based optimisation of wide, shallow neural networks, where the output of each hidden node is scaled by a positive parameter. The scaling parameters are non-identical, differing from the classical Neural Tangent Kernel (NTK) parameterisation. We prove that for large such neural networks, with high probability, gradient flow and gradient descent converge to a global minimum *and* can learn features in some sense, unlike in the NTK parameterisation. We perform experiments illustrating our theoretical results and discuss the benefits of such scaling in terms of prunability and transfer learning.

## 1 Introduction

The training of neural networks typically involves the minimisation of a non-convex objective function. However, first-order optimisation methods, such as gradient descent (GD) and its variants, often find solutions with low training error. To gain a better understanding of this phenomenon, one fruitful direction of research has been to analyse properties of GD training of over-parameterised, large-width neural networks; that is, neural networks where the number $m$ of hidden nodes in a given layer is very large. In particular, under a "$\sqrt{1/m}$" scaling of the hidden nodes, Jacot et al. (2018) have shown that, as the number of nodes $m$ tends to infinity, the solution obtained by GD achieves zero training error, and coincides with that of kernel regression under a so-called limiting *Neural Tangent Kernel* (NTK). Under the same node scaling, called *NTK scaling*,

quantitative theoretical guarantees for the global convergence and generalisation properties have then been obtained for large (but finite) width neural networks (Du et al., 2019b;a; Oymak & Soltanolkotabi, 2020; Arora et al., 2019a; Bartlett et al., 2021). However, it has been noted in a number of articles (Chizat et al., 2019; Yang, 2019; Arora et al., 2019a; Yang & Hu, 2021) that under NTK scaling, feature learning does not occur and GD training is performed in a *lazy-training* regime, in contrast with the typical feature-learning regime exhibited in deep neural networks.

**Main contributions.** We investigate global convergence properties and feature learning in gradient-type training of large-width feedforward neural networks (FFNNs) under a more general asymmetrical node scaling. In particular, each hidden node $j = 1, \ldots, m$ has a fixed node-specific scaling $\sqrt{\lambda_{m,j}}$ with

$$\lambda_{m,j} = \gamma \cdot \frac{1}{m} + (1 - \gamma) \cdot \frac{\widetilde{\lambda}_j}{\sum_{k=1}^m \widetilde{\lambda}_k} \tag{1}$$

where $\gamma \in [0,1]$ and $1 \geq \widetilde{\lambda}_1 \geq \widetilde{\lambda}_2 \geq \ldots \geq 0$ are nonnegative fixed scalars with $\sum_{j=1}^\infty \widetilde{\lambda}_j = 1$. Note that $\gamma = 1$ corresponds to the $\sqrt{1/m}$ NTK scaling. If $\gamma < 1$, the node scaling is necessarily asymmetrical for large-width networks. Two typical examples of the scalars $(\widetilde{\lambda}_j)_{j \geq 1}$ are (a) $\widetilde{\lambda}_j = 6\pi^{-2} j^{-2}$ for all $j \geq 1$, and (b) $\widetilde{\lambda}_1 = \ldots = \widetilde{\lambda}_K = 1/K$ and $\widetilde{\lambda}_j = 0$ for all $j > K$, for a fixed $K$.

We consider a shallow FFNN with a smooth activation function and without bias, where the first layer weights are trained via gradient flow or descent and empirical risk minimisation under the $\ell_2$ loss. We show that, under similar assumptions as in Du et al. (2019b;a) on the data, activation function, and initialisation, when the number of hidden nodes $m$ is sufficiently large: (i) if $\gamma > 0$, the training error goes to 0 at a linear rate with high probability; and (ii) feature learning (in the sense of the definitions given in Section 7.1) occurs if and only if $\gamma < 1$. We provide numerical experiments which illustrate the theoretical results and demonstrate empirically that such node-scaling is also useful for pruning and for transfer learning.

**Organisation of the paper.** Section 2 discusses related work. Section 3 introduces the FFNN model with asymmetrical node scaling, gradient flow or gradient descent updates, and the main assumptions on the data, activation function, and initialisation. Section 4 discusses the properties of the NTK of such a model at initialisation, and its infinite-width limit. Sections 5 and 6 derive our main results on the convergence to a global minimum of gradient flow and gradient descent and sketch their proofs. Section 7 gives the main results regarding feature learning. Section 8 describes our experiments on simulated and real datasets, whose results illustrate our theoretical results and their potential applications. The Supplementary Material contains detailed proofs, as well as results on additional convergence of gradient flow and feature learning when using the ReLU activation function.

**Notations.** For an integer $n \geq 1$, let $[n] = \{1, \ldots, n\}$. For a multivariate real-valued function $f : \mathbb{R}^n \to \mathbb{R}$, the gradient $\nabla_{\mathbf{v}} f(\mathbf{v})$ is the $n$-dimensional column vector of partial derivatives $\nabla_{\mathbf{v}} f(\mathbf{v}) = (\frac{\partial f}{\partial v_1}(\mathbf{v}), \ldots, \frac{\partial f}{\partial v_n}(\mathbf{v}))^\top$ where $\mathbf{v} = (v_1, \ldots, v_n)^\top$. For a square matrix $B$, we denote its minimum and maximum eigenvalues by $\text{eig}_{\min}(B)$ and $\text{eig}_{\max}(B)$, respectively. For a vector $\mathbf{v} \in \mathbb{R}^n$, we write $B = \text{diag}(\mathbf{v})$ for the $n$-by-$n$ diagonal matrix with $B_{ii} = v_i$ for $i \in [n]$.

## 2 Related work

**Large-width FFNNs.** The analysis of large-width FFNNs goes back to Neal (1996) who showed the connection between Gaussian processes and FFNNs in the large-width limit. Recent work has explored this connection under varying assumptions (Matthews et al., 2018; Lee et al., 2018; Yang, 2019; Favaro et al., 2020; Bracale et al., 2021; Lee et al., 2023; Jung et al., 2023).

**Large-width FFNNs under NTK scaling.** Following the seminal work of Jacot et al. (2018), a number of articles have investigated the benefits of over-parameterisation for gradient descent training, with the "$1/\sqrt{m}$" NTK scaling (Arora et al., 2019b; Du et al., 2019b;a; Lee et al., 2019; Zou & Gu, 2019; Oymak & Soltanolkotabi, 2020; Zou et al., 2020). Crucially, when the width of the network is large enough with respect

to the size of the training set, the training loss converges to a global minimum at a linear rate under gradient flow or gradient descent. However, under this symmetrical NTK scaling, the hidden-layer features do not move significantly when the width is large, and the scaling has been coined lazy-training regime for this reason (Chizat et al., 2019; Woodworth et al., 2020).

**Large-width FFNNs under mean-field scaling.** An alternative scaling is the "$1/m$" mean-field scaling (Rotskoff & Vanden-Eijnden, 2018; Mei et al., 2018; 2019; Chizat et al., 2019; Sirignano & Spiliopoulos, 2020; Ghorbani et al., 2020; Chen et al., 2021; Tao et al., 2021). This scaling is also equivalent, up to the so-called abc-scaling symmetry (Yang & Hu, 2021), to the $\mu P$ parameterisation of Yang & Hu (2021) in the case of shallow networks. Feature learning is known to occur under this mean-field scaling. Also, Chizat & Bach (2018) showed that under the same scaling, if the training of the model converges, it converges to a global minimum.

**Asymmetrical scaling in FFNNs.** The idea of using asymmetrical scaling parameters in the context of GD optimisation of deep FFNNs has been previously introduced by Wolinski et al. (2020). The focus of Wolinski et al. (2020) was on the (empirical) usefulness in terms of pruning. Indeed, our experiments in Section 8 are also in line with their findings. The work of Wolinski et al. (2020), however, only considered asymmetrical scaling with $\gamma = 0$ (no fixed part), and did not investigate global convergence properties under such scaling. The properties of random FFNNs under *random* asymmetrical node scaling in the large-width limit has also been considered by Lee et al. (2023); but this paper did not investigate the training properties under gradient flow or gradient descent.

## 3 Problem setup

### 3.1 Statistical model

We consider a shallow FFNN with one hidden layer of $m \geq 1$ hidden nodes and a scalar output. To simplify the analysis, we assume that there is no bias term. Let $\mathbf{x} \in \mathbb{R}^d$ be some input vector, where $d \geq 1$ is the input dimension. The model is defined as

$$f_m(\mathbf{x}; \mathbf{W}) = \sum_{j=1}^{m} \sqrt{\lambda_{m,j}} a_j \sigma(Z_j(\mathbf{x}; \mathbf{W})) \quad \text{with} \quad Z_j(\mathbf{x}; \mathbf{W}) = \frac{1}{\sqrt{d}} \mathbf{w}_j^\top \mathbf{x} \quad \text{for } j \in [m] \tag{2}$$

where $f_m(\mathbf{x}; \mathbf{W})$ is the scalar output of the FFNN, $Z_j(\mathbf{x}; \mathbf{W})$ is the pre-activation of the $j$-th hidden node, $\sigma : \mathbb{R} \to \mathbb{R}$ is the activation function, $\mathbf{w}_j \in \mathbb{R}^d$ is the column vector of weights between node $j$ of the hidden layer and the input nodes, and $a_j \in \mathbb{R}$ is the weight between the hidden node $j$ and the output node. The $\lambda_{m,j}$'s for $j \in [m]$ are nonnegative scaling parameters for the hidden nodes. The parameters to be optimised are contained in $\mathbf{W}$ which is an $md$-dimensional column vector $(\mathbf{w}_1^\top, \ldots, \mathbf{w}_m^\top)^\top$. We assume that $\sigma$ admits a derivative $\sigma'$.

For $n \geq 1$, let $\boldsymbol{\sigma} : \mathbb{R}^n \to \mathbb{R}^n$ (resp. $\boldsymbol{\sigma}' : \mathbb{R}^n \to \mathbb{R}^n$) be the vector-valued multivariate function that applies the real-valued function $\sigma$ (resp. $\sigma'$) element-wise on each of the $n$ input variables. To simplify the analysis, we assume throughout this article that the output weights $(a_j)_{j \in [m]}$ are randomly initialised and fixed afterwards:

$$a_j \overset{\text{iid}}{\sim} \text{Uniform}(\{-1, 1\}), \quad j \geq 1. \tag{3}$$

This simplifying assumption is often made when analysing large shallow networks (see e.g. (Du et al., 2019b; Bartlett et al., 2021)), and the analysis can also be extended to the case where both layers are trained. The scaling parameters $(\lambda_{m,j})_{j \in [m]}$ are fixed and parameterised as in Equation (1). By construction, we have $\lambda_{m,1} > 0$ and $\sum_{j=1}^{m} \lambda_{m,j} = 1$ for all $m \geq 1$. Recall that the case $\gamma = 1$ corresponds to NTK scaling. Also, note that our model covers finite FFNNs: when

$$\gamma = 0 \quad \text{and} \quad \widetilde{\lambda}_j = \begin{cases} 1/K & \text{if } j \in [K] \\ 0 & \text{otherwise} \end{cases}$$

for some $K \leq m$, the model becomes a finite network of width $K$. In the experiments, we will consider the special case where $(\widetilde{\lambda}_j)_{j \geq 1}$ are the probability masses of a Zipf law:

$$\widetilde{\lambda}_j = \frac{1}{\zeta(1/\alpha)} \frac{1}{j^{1/\alpha}}, \quad j \geq 1 \tag{4}$$

for some $\alpha \in (0, 1)$, where $\zeta$ is the Riemann zeta function. The parameter $\alpha$ tunes how quickly $\widetilde{\lambda}_j$ decreases with $j$, smaller values corresponding to more rapid decrease and more asymmetry.

### 3.2 Training

Let $\mathcal{D}_n = \{(\mathbf{x}_i, y_i)\}_{i \in [n]}$ be the training dataset, where $n \geq 1$ is the number of observations. Let $\mathbf{X}$ denote the $n$-by-$d$ matrix whose $i$th row is $\mathbf{x}_i^\top$. We want to minimise the empirical risk under $\ell_2$ loss. Consider the objective function

$$L_m(\mathbf{W}) = \frac{1}{2} \sum_{i=1}^{n} (y_i - f_m(\mathbf{x}_i; \mathbf{W}))^2 \tag{5}$$

which is non-convex in general. For a given dataset $\mathcal{D}_n$, width $m \geq 1$, output weights $(a_j)_{j \in [m]}$, and scaling parameters $(\lambda_{m,j})_{j \in [m]}$, we aim to estimate the trainable parameters $\mathbf{W}$ by minimising $L_m(\mathbf{W})$ using gradient flow or gradient descent. Let $\mathbf{W}_0$ be some initialisation. In gradient flow, $(\mathbf{W}_t)_{t>0}$ is the solution to the following ordinary differential equation (ODE):

$$\frac{d\mathbf{W}_t}{dt} = -\nabla_{\mathbf{W}} L_m(\mathbf{W}_t)$$

with $\lim_{t \to 0} \mathbf{W}_t = \mathbf{W}_0$. Let $\mathbf{w}_{tj}$ be the value of the parameter $\mathbf{w}_j$ at time $t$, and define $Z_{tj}(\mathbf{x}) = Z_j(\mathbf{x}; \mathbf{W}_t)$. Note that $\nabla_{\mathbf{w}_j} f_m(\mathbf{x}; \mathbf{W}) = \sqrt{\lambda_{m,j}} a_j \sigma'(Z_j(\mathbf{x}; \mathbf{W})) \cdot \frac{1}{\sqrt{d}} \mathbf{x}$. Under gradient flow, for $j \in [m]$,

$$\frac{d\mathbf{w}_{tj}}{dt} = \left( \sum_{i=1}^{n} (y_i - f_m(\mathbf{x}_i; \mathbf{W}_t)) \nabla_{\mathbf{w}_j} f_m(\mathbf{x}_i; \mathbf{W}_t) \right) = \left( \frac{\sqrt{\lambda_{m,j}} a_j}{\sqrt{d}} \sum_{i=1}^{n} (y_i - f_m(\mathbf{x}_i; \mathbf{W}_t)) \sigma'(Z_{tj}(\mathbf{x}_i)) \mathbf{x}_i \right).$$

Thus, the derivatives associated with each hidden node $j$ are scaled by $\sqrt{\lambda_{m,j}}$. For an input $\mathbf{x} \in \mathbb{R}^d$, the output of the FFNN therefore satisfies the ODE

$$\frac{df_m(\mathbf{x}; \mathbf{W}_t)}{dt} = \left( \nabla_{\mathbf{W}} f_m(\mathbf{x}; \mathbf{W}_t)^\top \frac{d\mathbf{W}_t}{dt} \right) = \left( \sum_{i=1}^{n} (y_i - f_m(\mathbf{x}_i; \mathbf{W}_t)) \Theta_m(\mathbf{x}, \mathbf{x}_i; \mathbf{W}_t) \right),$$

where $\Theta_m : \mathbb{R}^d \times \mathbb{R}^d \to \mathbb{R}$ is the neural tangent kernel for the network $f_m(\mathbf{x}; \mathbf{W})$:

$$\Theta_m(\mathbf{x}, \mathbf{x}'; \mathbf{W}) = \frac{\mathbf{x}^\top \mathbf{x}'}{d} \sum_{j=1}^{m} \lambda_{m,j} \sigma'(Z_j(\mathbf{x}; \mathbf{W})) \sigma'(Z_j(\mathbf{x}'; \mathbf{W})). \tag{6}$$

The associated neural tangent Gram (NTG) matrix $\widehat{\Theta}_m(\mathbf{X}; \mathbf{W})$ is the $n$-by-$n$ positive semi-definite matrix whose $(i, j)$-th entry is $\Theta_m(\mathbf{x}_i, \mathbf{x}_j; \mathbf{W})$. It takes the form

$$\widehat{\Theta}_m(\mathbf{X}; \mathbf{W}) = \frac{1}{d} \sum_{j=1}^{m} \lambda_{m,j} \operatorname{diag}\left( \boldsymbol{\sigma}'\left( \frac{\mathbf{X}\mathbf{w}_j}{\sqrt{d}} \right) \right) \mathbf{X}\mathbf{X}^\top \operatorname{diag}\left( \boldsymbol{\sigma}'\left( \frac{\mathbf{X}\mathbf{w}_j}{\sqrt{d}} \right) \right). \tag{7}$$

Gradient descent is a discretisation of gradient flow. Under gradient descent, the parameters are updated by

$$\mathbf{W}_s = \mathbf{W}_{s-1} - \eta \nabla_{\mathbf{W}} L_m(\mathbf{W}_{s-1}) \quad \text{for all } s \in \mathbb{N}, \tag{8}$$

where $\eta > 0$ is a learning rate. These updates give rise to the family $(\mathbf{W}_s)_{s \in \mathbb{N} \cup \{0\}}$ indexed by discrete time steps $s = 0, 1, 2, \ldots$, rather than continuous times $t \geq 0$.

### 3.3 Main assumptions

Throughout the paper, we assume that the activation function $\sigma$ satisfies the following standard condition: for all random variables $Z \sim \mathcal{N}(0, s^2)$ for some $s > 0$,

$$|\mathbb{E}_Z[\sigma(Z)]| < \infty \qquad \text{and} \qquad 0 < \mathbb{E}_Z[\sigma(Z)^2] < \infty \tag{9}$$

This assumption is made all the time, and so we do not mention its use explicitly in the paper.

The results of this article on global convergence and feature learning use several further assumptions. The first set of these assumptions, which are mild and similar to other assumptions used in the literature, is on the training dataset $\mathcal{D}_n = \{(\mathbf{x}_i, y_i)\}_{i \in [n]}$.

**Assumption 3.1** (Dataset). (a) All inputs are non-zero and have norms at most 1: $0 < \|\mathbf{x}_i\| \leq 1$ for all $i \geq 1$. (b) For all $i \neq i'$ and $c \in \mathbb{R}$, $\mathbf{x}_i \neq c\mathbf{x}_{i'}$. (c) There is $C > 0$ such that $|y_i| \leq C$ for all $i \geq 1$.

The next assumption concerns the activation function $\sigma$. Standard activation functions (softplus, tanh, sigmoid, swish) satisfy this assumption, but not the ReLU. However, some of our results, such as global convergence of gradient flow and feature-learning results, also hold in the ReLU case, as shown in Appendix A in the Supplementary Material.

**Assumption 3.2** (Activation function). The activation function is analytic, with $|\sigma'(x)| \leq 1$ and $|\sigma''(x)| \leq M$ for some $M > 0$, but it is not a polynomial.

The last assumption, which is standard, is on the initialisation of the weights.

**Assumption 3.3** (Initialisation). For $j \in \mathbb{N}$, $\mathbf{w}_{0j} \stackrel{\text{iid}}{\sim} \mathcal{N}(0, \mathbf{I}_d)$, where $\mathbf{I}_d$ is the $d$-by-$d$ identity matrix.

## 4 Neural Tangent Kernel at initialisation and its limit

**Mean NTG at initialisation and its minimum eigenvalue.** Let $\mathbf{W}_0$ be a random initialisation from Assumption 3.3. Consider the mean NTK at initialisation

$$\Theta^*(\mathbf{x}, \mathbf{x}') = \mathbb{E}\left[\Theta_m(\mathbf{x}, \mathbf{x}'; \mathbf{W}_0)\right] = \frac{\mathbf{x}^\top \mathbf{x}'}{d} \mathbb{E}\left[\sigma'\left(\frac{1}{\sqrt{d}}\mathbf{w}_{01}^\top \mathbf{x}\right) \sigma'\left(\frac{1}{\sqrt{d}}\mathbf{w}_{01}^\top \mathbf{x}'\right)\right]. \tag{10}$$

The mean NTK, which is also, by the law of large numbers, the limiting NTK under $1/\sqrt{m}$ scaling (Jacot et al., 2018), does not depend on $(\lambda_{m,j})_{j \geq 1}$ nor $m$. Let $\widehat{\Theta}^*(\mathbf{X}) = \mathbb{E}[\widehat{\Theta}_m(\mathbf{X}; \mathbf{W}_0)]$ be the associated $n$-by-$n$ mean NTG matrix at initialisation, whose $(i, j)$-th entry is $\Theta^*(\mathbf{x}_i, \mathbf{x}_j)$. Let $\kappa_n = \text{eig}_{\min}(\widehat{\Theta}^*(\mathbf{X}))$ be the minimum eigenvalue of the mean NTG matrix at initialisation. This minimum eigenvalue plays an important role in the analysis of global convergence properties in the lazy-training regime. Many authors (see e.g. (El Karoui, 2010; Nguyen et al., 2021)) have shown that, under some assumptions on the data, activation function, and initialisation, $\kappa_n$ is strictly positive or bounded away from zero. such a result, under the Assumptions of Section 3.3.

**Proposition 4.1** ((Du et al., 2019b, Theorem 3.1) and (Du et al., 2019a, Proposition F.1)). *When Assumptions 3.1 to 3.3 hold, we have $\kappa_n > 0$.*

*Remark* 4.2. Du et al. (2019b;a) make the assumption that each $\mathbf{x}_i$ has unit norm. But their proof holds under the less strict Assumption 3.1(a). The above proposition also holds if Assumption 3.2 is replaced by the assumption that $\sigma$ is the ReLU function.

**Limiting NTG.** To give some intuition, we now describe the limiting behaviour of the NTG, for a fixed sample size $n$, as the width $m$ goes to infinity. The proof, given in Appendix C in the Supplementary Material, follows from the triangle inequality and the law of large numbers, together with $|\sigma'(z)| \leq 1$ and $\sum_{j \geq 1} \widetilde{\lambda}_j = 1$.

**Proposition 4.3.** *Consider a sequence $(\mathbf{w}_{0j})_{j \geq 1}$ of iid random vectors distributed as in Assumption 3.3. Suppose Assumption 3.2 holds. Then,*

$$\widehat{\Theta}_m(\mathbf{X}; \mathbf{W}_0) \to \widehat{\Theta}_\infty(\mathbf{X}; \mathbf{W}_0) \tag{11}$$

*almost surely as $m \to \infty$, where $\widehat{\Theta}_\infty(\mathbf{X}; \mathbf{W}_0) = \gamma \widehat{\Theta}^*(\mathbf{X}) + (1-\gamma)\widehat{\Theta}_\infty^{(2)}(\mathbf{X}; \mathbf{W}_0)$, with $\widehat{\Theta}_\infty^{(2)}(\mathbf{X}; \mathbf{W}_0)$ being the following random positive semi-definite matrix:*

$$\widehat{\Theta}_\infty^{(2)}(\mathbf{X}; \mathbf{W}_0) = \frac{1}{d}\sum_{j=1}^\infty \widetilde{\lambda}_j \, \text{diag}\left(\boldsymbol{\sigma}'\left(\frac{\mathbf{X}\mathbf{w}_{0j}}{\sqrt{d}}\right)\right)\mathbf{X}\mathbf{X}^\top \text{diag}\left(\boldsymbol{\sigma}'\left(\frac{\mathbf{X}\mathbf{w}_{0j}}{\sqrt{d}}\right)\right). \tag{12}$$

*Also, $\mathbb{E}[\widehat{\Theta}_\infty(\mathbf{X}; \mathbf{W}_0)] = \mathbb{E}[\widehat{\Theta}_\infty^{(2)}(\mathbf{X}; \mathbf{W}_0)] = \widehat{\Theta}^*(\mathbf{X})$, and*

$$\mathbb{E}\left[\|\widehat{\Theta}_\infty(\mathbf{X}; \mathbf{W}_0) - \widehat{\Theta}^*(\mathbf{X})\|_F^2\right] = C_0(\mathbf{X})(1-\gamma)^2\sum_{j\geq 1}\widetilde{\lambda}_j^2 \tag{13}$$

*where $\|\cdot\|_F$ denotes the Frobenius norm, and $C_0(\mathbf{X}) > 0$ is a positive constant equal to*

$$\sum_{1\leq i,i'\leq n}\left(\frac{\mathbf{x}_i^\top \mathbf{x}_{i'}}{d}\right)^2 \text{Var}\left(\sigma'\left(\frac{1}{\sqrt{d}}\mathbf{w}_{01}^\top \mathbf{x}_i\right)\sigma'\left(\frac{1}{\sqrt{d}}\mathbf{w}_{01}^\top \mathbf{x}_{i'}\right)\right).$$

When $\gamma = 1$ (NTK scaling), the NTG converges to a constant matrix, and solutions obtained by gradient flow coincide with that of kernel regression. Whenever $\gamma < 1$, Proposition 4.3 shows that the NTG is random at initialisation, even in the infinite-width limit, contrary to that of NTK scaling. As shown in Equation (13), the departure from the symmetric regime, as measured by the total variance of the limiting random NTG, can be quantified by the nonnegative constant $(1-\gamma)^2\sum_{j\geq 1}\widetilde{\lambda}_j^2 \in [0,1]$. When this constant is close to 0, we approach the kernel regime, and increasing this value leads to a departure from the regime. The quantity $\sum_{j\geq 1}\widetilde{\lambda}_j^2 \in (0,1]$ is always strictly positive. More rapid decrease of the $\widetilde{\lambda}_j$ as $j$ increases will lead to higher values of $\sum_{j\geq 1}\widetilde{\lambda}_j^2$. For example, when using the Zipf weights in Equation (4), we have $\sum_{j\geq 1}\widetilde{\lambda}_j^2 = \frac{\zeta(2/\alpha)}{\zeta(1/\alpha)^2}$, which decreases with $\alpha$, as shown in Figure 1 in the Supplementary Material.

This section has described the behaviour of the NTG at initialisation in the infinite-width limit, and has provided intuition on the node-scaling parameters. The next three sections contain results on global convergence and feature learning properties of large, but finite, FFNNs under such asymmetrical scaling.

## 5 Global convergence analysis for gradient flow

Our global convergence theorem, which is given below, explains what happens during training via gradient flow. Recall that $\kappa_n$ is the minimum eigenvalue of the mean NTG matrix $\widehat{\Theta}^*(\mathbf{X})$ at initialisation. Our theorem says that with high probability, (i) the loss decays exponentially fast with respect to $\kappa_n$ and training time $t$, and (ii) the NTG and weights $\mathbf{w}_{tj}$ change, respectively, by

$$\|\widehat{\Theta}_m(\mathbf{X}; \mathbf{W}_t) - \widehat{\Theta}_m(\mathbf{X}; \mathbf{W}_0)\|_2 = O\left(\frac{n^3\sum_{j=1}^m\lambda_{m,j}^2}{\kappa_n^2 d^3\gamma^2} + \frac{n^2\sqrt{\sum_{j=1}^m\lambda_{m,j}^2}}{\kappa_n d^2\gamma}\right) \quad \text{and} \quad \|\mathbf{w}_{tj} - \mathbf{w}_{0j}\| = O\left(\frac{n\lambda_{m,j}^{1/2}}{\kappa_n d^{1/2}\gamma}\right).$$

Define

$$C_1 = \sup_{c\in(0,1]}\mathbb{E}_{z\sim\mathcal{N}(0,1)}\left[\sigma\left(\frac{cz}{\sqrt{d}}\right)^2\right]. \tag{14}$$

**Theorem 5.1.** *(Global convergence, gradient flow) Let $\delta \in (0,1)$. Suppose Assumptions 3.1 to 3.3 hold, and that*

$$\gamma > 0, \quad and \quad m \geq \max\left(\frac{2^3 n\log\frac{2n}{\delta}}{\kappa_n d}, \frac{2^{10}n^3 M^2(C^2+C_1)}{\kappa_n^3 d^3\gamma^2\delta}, \frac{2^{15}n^4 M^2(C^2+C_1)}{\kappa_n^4 d^4\gamma^2\delta}\right),$$

*where $C$ is the bound on the $y_i$'s in Assumption 3.1. Then, with probability at least $1 - \delta$, the following properties hold for all $t \geq 0$:*

*(a) $\text{eig}_{\min}(\widehat{\Theta}_m(\mathbf{X}; \mathbf{W}_t)) \geq \frac{\gamma\kappa_n}{4}$;*
*(b) $L_m(\mathbf{W}_t) \leq e^{-(\gamma\kappa_n t)/2}L_m(\mathbf{W}_0)$;*

*(c)* $\|\mathbf{w}_{tj} - \mathbf{w}_{0j}\| \leq \frac{n\sqrt{\lambda_{m,j}}}{\kappa_n d^{1/2}} \sqrt{\frac{2^7(C^2+C_1)}{\gamma^2 \delta}}$ *for all* $j \in [m]$;

*(d)* $\|\widehat{\Theta}_m(\mathbf{X}; \mathbf{W}_t) - \widehat{\Theta}_m(\mathbf{X}; \mathbf{W}_0)\|_2 \leq \left( \frac{2^7 n^3 M^2 (C^2+C_1)}{\kappa_n^2 d^3 \gamma^2 \delta} \cdot \sum_{j=1}^m \lambda_{m,j}^2 \right) + \left( \frac{2^5 n^2 M(C^2+C_1)^{1/2}}{\kappa_n d^2 \gamma \delta^{1/2}} \cdot \sqrt{\sum_{j=1}^m \lambda_{m,j}^2} \right).$

The above theorem implies that, whenever $\gamma > 0$, the training error converges to 0 exponentially fast. Additionally, the weight change is bounded by a factor $\sqrt{\lambda_{m,j}}$ and the NTG change is bounded by a factor $\sqrt{\sum_{j=1}^m \lambda_{m,j}^2}$. We have (see Appendix B.2 in the Supplementary Material) that as $m \to \infty$,

$$\lambda_{m,j} \to (1-\gamma)\widetilde{\lambda}_j \text{ for all } j \geq 1 \qquad \text{and} \qquad \sum_{j=1}^m \lambda_{m,j}^2 \to (1-\gamma)^2 \sum_{j=1}^\infty \widetilde{\lambda}_j^2.$$

If $\widetilde{\lambda}_j > 0$ (note that we necessarily have $\widetilde{\lambda}_1 > 0$), the upper bound in (c) is therefore vanishing in the infinite-width limit if and only if $\gamma = 1$ (lazy-training regime). Similarly, the upper bound in (d) is vanishing if and only if $\gamma = 1$. Although we were not able to obtain matching lower bounds, we show in Section 7 that feature learning arises whenever $\gamma < 1$.

*Remark* 5.2. We make two comments on Theorem 5.1. First, although $d$ represents the input dimension, all the occurrences of $d$ in the theorem, such as those in the lower bound of the width $m$, do not come from the complexity of the input dimension. Instead, it comes from the fact that our model uses the $1/\sqrt{d}$ scaling when computing the pre-activation values of the first layer. If this scaling were removed in our model, the statement of the theorem would not include $d$ (i.e., we would have the theorem with $d$ set to 1). Second, a result similar to Theorem 5.1 also holds for the ReLU activation function. See Theorem A.1 in the Supplementary Material.

**Sketch of the proof.** We give here a sketch of the proof of Theorem 5.1 (and of Theorem A.1, its ReLU counterpart, in the Supplementary Material). The detailed proofs are given in Appendices F and G in the Supplementary Material, with secondary lemmas given in Appendices D and E there. The structures of the proofs of Theorems 5.1 and A.1 are similar to that of (Du et al., 2019b, Theorem 3.2), which showed analogous results on the global convergence under NTK scaling. However, there are some key differences which we highlight below.

Gradient flow converges to a global minimum of the objective function if the minimum eigenvalue of the NTG matrix $\widehat{\Theta}_m(\mathbf{X}; \mathbf{W}_t)$ is bounded away from zero, for $m$ sufficiently large, by some positive constant for all $t \geq 0$. In the NTK scaling case ($\gamma = 1$), Du et al. (2019b) showed that the following is satisfied, for $m$ sufficiently large, with high probability: (i) the NTG matrix at initialisation is close to its mean, and the minimum eigenvalue is close to that of the mean NTG, (ii) the weights $\mathbf{w}_{tj}$ are nearly constant in time, which implies that (iii) the NTG matrix is nearly constant in time, hence (iv) the minimum eigenvalue of the NTG matrix at time $t$ is close to its value at initialisation, which is bounded away from zero.

However, in the case of asymmetrical node scaling ($\gamma < 1$), none of the points (i-iv) holds. At initialisation, the random NTG matrix may be significantly different from its mean. Additionally, both the weights and the NTG matrix substantially change over time. This therefore requires a somewhat different approach that we now describe.

Let $\lambda_{m,j}^{(1)} = \gamma/m$ and $\lambda_{m,j}^{(2)} = ((1-\gamma)\widetilde{\lambda}_j)/\sum_{k=1}^m \widetilde{\lambda}_k$, and note that $\lambda_{m,j}^{(1)} + \lambda_{m,j}^{(2)} = \lambda_{m,j}$. For $k \in \{1, 2\}$, let $\widehat{\Theta}_m^{(k)}$ be the $n$-by-$n$ symmetric positive semi-definite matrices defined by Equation (7), with $\lambda_{m,j}$ replaced by either $\lambda_{m,j}^{(k)}$. Note that

$$\widehat{\Theta}_m(\mathbf{X}; \mathbf{W}_t) = \widehat{\Theta}_m^{(1)}(\mathbf{X}; \mathbf{W}_t) + \widehat{\Theta}_m^{(2)}(\mathbf{X}; \mathbf{W}_t) \tag{15}$$

with $\mathbb{E}[\widehat{\Theta}_m^{(1)}(\mathbf{X}; \mathbf{W}_0)] = \gamma\widehat{\Theta}^*(\mathbf{X})$.

The key idea of the proof is to use the above decomposition of the NTG matrix as a sum of two terms, and to show that, while the second term may change over time, the first term is close to its mean at initialisation, and does not change much over time. The important points of the proof are as follows. For large $m$, with

high probability: (i) $\widehat{\Theta}_m^{(1)}(\mathbf{X}; \mathbf{W}_0)$ is close to its mean $\gamma \widehat{\Theta}^*(\mathbf{X})$ and its minimum eigenvalue is therefore lower bounded by $(\gamma \kappa_n)/2$;

The important points of the proof are as follows. For large $m$, with high probability: (i) $\widehat{\Theta}_m^{(1)}(\mathbf{X}; \mathbf{W}_0)$ is close to its mean $\gamma \widehat{\Theta}^*(\mathbf{X})$ and its minimum eigenvalue is therefore lower bounded by $(\gamma \kappa_n)/2$; (ii) while the weights $\mathbf{W}_t$ may change significantly over time, $\widehat{\Theta}_m^{(1)}(\mathbf{X}; \mathbf{W}_t)$ remains nearly constant over time; (iii) as a result, the minimum eigenvalue of $\widehat{\Theta}_m^{(1)}(\mathbf{X}; \mathbf{W}_t)$ can be lower bounded by $(\gamma \kappa_n)/4$; (iv) this implies that the minimum eigenvalue of the overall NTG matrix $\widehat{\Theta}_m(\mathbf{X}; \mathbf{W}_t)$ is lower bounded by $(\gamma \kappa_n)/4$.

Since showing that the first part of the NTK/NTG in Equation (15) does not change in the limit of $m \to \infty$ is a key component of the proof, we give here an outline of it in the simplified case of $d = 1$ and $\sigma$ being smooth. Let

$$\Theta_m^{(1)}(\mathbf{x}, \mathbf{x}'; \mathbf{W}_t) = \mathbf{x}\mathbf{x}' \frac{\gamma}{m} \sum_{j=1}^{m} \sigma'(\mathbf{w}_{tj}\mathbf{x}) \sigma'(\mathbf{w}_{tj}\mathbf{x}')$$

be the first part of the NTK. We have, over gradient flow,

$$\left| \frac{d\Theta_m^{(1)}(\mathbf{x}, \mathbf{x}'; \mathbf{W}_t)}{dt} \right| = \left| \mathbf{x}\mathbf{x}' \frac{\gamma}{m} \sum_{j=1}^{m} \left( \mathbf{x}\sigma''(\mathbf{w}_{tj}\mathbf{x})\sigma'(\mathbf{w}_{tj}\mathbf{x}') + \mathbf{x}'\sigma'(\mathbf{w}_{tj}\mathbf{x})\sigma''(\mathbf{w}_{tj}\mathbf{x}') \right) \frac{d\mathbf{w}_{tj}}{dt} \right|$$

$$\le 2M \frac{\gamma}{m} \sum_{j=1}^{m} \left| \frac{d\mathbf{w}_{tj}}{dt} \right|,$$

where the last inequality follows from the triangle inequality and Assumptions 3.1 and 3.2. Furthermore,

$$\left| \frac{d\mathbf{w}_{tj}}{dt} \right| = \sqrt{\lambda_{m,j}} \left| \sum_{i=1}^{n} (y_i - f_m(\mathbf{x}_i; \mathbf{W}_t)) \sigma'(Z_{tj}(\mathbf{x}_i)) \mathbf{x}_i \right|$$

$$\le \sqrt{\lambda_{m,j}} \sum_{i=1}^{n} |y_i - f_m(\mathbf{x}_i; \mathbf{W}_t)|$$

$$\le \sqrt{\lambda_{m,j}} 2n L_m(\mathbf{W}_t),$$

where the first inequality follows from the triangle inequality and Assumptions 3.1 and 3.2, and the second inequality follows from Cauchy-Schwarz. The change in the first part of the NTK is thus bounded by a quantity involving $\frac{\gamma}{m} \sum_{j=1}^{m} \sqrt{\lambda_{m,j}}$. Under the scaling in Equation (1), $\frac{\gamma}{m} \sum_{j=1}^{m} \sqrt{\lambda_{m,j}} \to 0$ as $m \to \infty$ (see Appendix B.2 in the Supplementary Material). Hence, the change in the first part of the NTK/NTG becomes asymptotically small as the width $m$ increases. It is worth noting that for the full NTK, i.e. the sum of both parts in Equation (15), a similar derivation leads to an upper bound of the order of $(\frac{\gamma}{m} \sum_{j=1}^{m} \sqrt{\lambda_{m,j}}) + (1 - \gamma) \frac{\sum_{j=1}^{m} \widetilde{\lambda}_j \sqrt{\lambda_{m,j}}}{\sum_{k=1}^{m} \widetilde{\lambda}_k}$. Due to the second term, this quantity does not converges to 0 as $m \to \infty$, unless $\gamma = 1$ (symmetric case).

# 6 Global convergence analysis for gradient descent

Let $\mathbf{y} \in \mathbb{R}^n$ be the vector of outputs $(y_1, \ldots, y_n)^\top$ in the training dataset $\mathcal{D}_n = \{(\mathbf{x}_i, y_i)\}_{i \in [n]}$. For each gradient-descent step $s$, let $\mathbf{u}_s \in \mathbb{R}^n$ be the outputs at step $s$ based on the inputs in $\mathcal{D}_n$, that is, $\mathbf{u}_s = (f_m(\mathbf{x}_1; \mathbf{W}_s), \ldots, f_m(\mathbf{x}_n; \mathbf{W}_s))^\top$. The following convergence theorem intuitively says that if the learning rate $\eta$, of gradient descent, is sufficiently small and the width of the network is large enough, then with high probability, the training error of the network decays exponentially fast to 0. We remark that our node scaling is independent of the learning rate $\eta$. Thus, the only conditions on $\eta$ in this work are in this section where we extend our results from gradient flow to gradient descent.

**Theorem 6.1.** *Consider $\delta \in (0, 1)$. Suppose Assumptions 3.1 to 3.3 hold, and that $\gamma > 0$, and*

$$0 < \eta < \min \left( \frac{2}{\gamma \kappa_n}, \frac{\gamma \kappa_n d^2}{8n^2}, \frac{\gamma \kappa_n d^2 \delta^{1/2}}{2^{9/2} n^2 M (C^2 + C_1)^{1/2}} \right),$$

*where $C$ and $C_1$ are from Assumption 3.1 and Equation* (14). *Let $\beta = (1 - \eta\gamma\kappa_n/2)^{1/2}$. If*

$$m \geq \max \left( \frac{2^3 n \log \frac{2n}{\delta}}{\kappa_n d}, \; \frac{2^5 \eta^2 n^3 M^2 (C^2 + C_1)}{\kappa_n d^3 (1 - \beta)^2 \delta}, \; \frac{2^{11} \eta^2 n^4 M^2 (C^2 + C_1)}{\kappa_n^2 d^4 (1 - \beta)^2 \delta} \right),$$

*then with probability at least $1 - \delta$,*

$$\|\mathbf{y} - \mathbf{u}_s\|^2 \leq (1 - \alpha)^s \|\mathbf{y} - \mathbf{u}_0\|^2 \; \textit{for all } s \in \mathbb{N} \cup \{0\}. \tag{16}$$

Note that the condition on the learning rate requires $\eta = O(\gamma\kappa_n/n^2)$. Thus, the best possible convergence rate from the theorem is $(1 - (\eta\gamma\kappa_n/2)) = (1 - (C_0\gamma^2\kappa_n^2/n^2))$ for some constant $C_0$.

The proof is by induction on the gradient-descent step $s$, and is described in detail in Appendix H in the Supplementary Material. It is similar to the proof of (Du et al., 2019a, Theorem 5.1), but the two proofs differ significantly because, as in the case of gradient flow, the weights $\mathbf{w}_{sj}$ and the Gram matrix $\widehat{\Theta}_m(\mathbf{X}; \mathbf{W}_s)$ change during gradient descent in our case, while they remain nearly constant in the case of (Du et al., 2019a).

## 7 Feature learning analysis

In this section, we present some results about feature learning. We focus on node scalings $(\lambda_{m,j})_{j \in [m]}$ of the form in Equation (1), both asymmetrical ($\gamma < 1$) and symmetrical ($\gamma = 1$; that is, $\lambda_{m,j} = 1/m$), but we also discuss alternative parameterisations such as mean-field and $\mu P$. For shallow neural networks trained by gradient descent, mean-field and $\mu P$ parameterisations are equivalent (Yang & Hu, 2021), and correspond to node scalings $\lambda_{m,j} = 1/m^2$ and learning rate $\eta = \eta_0 m$ for some $\eta_0 > 0$. We start with some definitions of feature learning in the context of potentially asymmetric scalings, generalising existing definitions. We then present feature learning results first under a linear activation function, and next under a general nonlinear activation function.

### 7.1 Definitions

**Definition 7.1** (Feature learning)**.** Let $(\mathbf{w}_{0j})_{j \geq 1}$ be a sequence of random initialisations for nodes $j \geq 1$. We will say that *feature learning* occurs during training if[1]

$$\liminf_{m \to \infty} \frac{\sum_{j=1}^m \lambda_{m,j} \left( \sigma(Z_j(\mathbf{x}; \mathbf{W}_t)) - \sigma(Z_j(\mathbf{x}; \mathbf{W}_0)) \right)^2}{\sum_{k=1}^m \lambda_{m,k} \left( \sigma(Z_k(\mathbf{x}; \mathbf{W}_0)) \right)^2} > 0 \tag{17}$$

almost surely, or in probability, for some $t \in (0, \infty]$ and $\mathbf{x}$. Here $t = \infty$ refers to the case that the ratio in the above inequality is the limit as $t$ tends to $\infty$.

The left-hand-side quantity in Equation (17) corresponds to the relative change in the scaled $m$-dimensional feature maps between time $0$ and time $t$. Definition 7.1 matches the definitions of (Yang & Hu, 2021, Definitions 3.5, H.2, H.9) or (Frei et al., 2023, Proposition 3.2) in the case of symmetrical NTK or mean-field node scalings.

*Remark* 7.2. As already noted by (Yang & Hu, 2021, Remark H.10), Definition 7.1 is a relatively weak notion of feature learning. It only requires a change in the feature map for some $t \in (0, \infty]$ and some $\mathbf{x}$, and does not relate to the relevance of the learn features for prediction. However, we show empirically in Section 8 that the feature learning property leads to better performances in terms of prunability and transfer learning.

The previous definition ensures that a change occurs in the feature map. However, it may still be the case that the contributions from all the individual nodes remain asymptotically infinitesimally small, in such a

---

[1]In (Yang & Hu, 2021), feature learning was defined in the large-width limits for stable and nontrivial parameterisations. Nontrivial means that the neural network function $f_m$ is not constant in time. Stable means that both the preactivations and activations have $\Theta(1)$ coordinates at initialisation and $O(1)$ coordinates throughout training. Both of these properties are satisfied by our model.

way that there are no nodes representing important features in the network. This is problematic if one is interested in pruning the nodes of the network, as we show theoretically for a linear activation in Section 7.2, and empirically in Section 8. We introduce below the stronger definition of *non-uniform feature learning*.

**Definition 7.3** (Non-uniform feature learning). Let $(\mathbf{w}_{0j})_{j \geq 1}$ be a sequence of random initialisations for nodes $j \geq 1$. We will say that *non-uniform feature learning* occurs during training if

$$\liminf_{m \to \infty} \frac{\max_{j \in [m]} \lambda_{m,j} \left( \sigma(Z_j(\mathbf{x}; \mathbf{W}_t)) - \sigma(Z_j(\mathbf{x}; \mathbf{W}_0)) \right)^2}{\sum_{k=1}^{m} \lambda_{m,k} \left( \sigma(Z_k(\mathbf{x}; \mathbf{W}_0)) \right)^2} > 0 \tag{18}$$

almost surely or in probability, for some $t \in (0, \infty]$ and $\mathbf{x}$.

Note that non-uniform feature learning implies feature learning, but the converse does not hold. For instance, feature learning holds under the mean-field parameterisation, but not non-uniform feature learning (see the next subsection for an illustration with a linear activation).

## 7.2 Linear activation function

We now describe, with the following theorem, analytic results in the case of a linear activation function. Although for fixed second-layer weights, the NTK does not change in this linear-activation case, the evolution of the weights provides useful insights into the differences between the symmetrical and asymmetrical scalings in terms of weight change.

**Theorem 7.4.** *Assume that the activation function is $\sigma(x) = x$, i.e., the identity map. Let $(\lambda_{m,j})$ be some node scalings, not necessarily of the form $(1)^2$. Let $\mathbf{X} = \mathbf{U}\mathbf{D}\mathbf{V}^\top$ be a reduced singular value decomposition of $\mathbf{X}$, where $\mathbf{U}$ is an $n \times k$ matrix with orthonormal columns, $\mathbf{D}$ is a diagonal $k \times k$ matrix, $\mathbf{V}$ is a $d \times k$ matrix with orthonormal columns, and $k \leq \min(n, d)$ is the rank of $\mathbf{X}$. For all $j \in [m]$, the difference between the solution of gradient descent[3] /flow $\mathbf{w}_{\infty j}$ and the initialisation $\mathbf{w}_{0j}$ is given by*

$$\mathbf{w}_{\infty j} - \mathbf{w}_{0j} = \frac{\sqrt{\lambda_{m,j}}}{\sum_{k=1}^{m} \lambda_{m,k}} a_j (\boldsymbol{\beta}_\infty - \mathbf{V}\mathbf{V}^\top \boldsymbol{\beta}_0) \tag{19}$$

*where $\boldsymbol{\beta}_0 = \sum_{j=1}^{m} \sqrt{\lambda_{m,j}} a_j \mathbf{w}_{0j}$, and $\boldsymbol{\beta}_\infty = \sqrt{d}\,\mathbf{V}\mathbf{D}^{-1}\mathbf{U}^\top\mathbf{y}$ is the minimum-norm solution of*

$$\operatorname*{argmax}_{\boldsymbol{\beta}} \frac{1}{2}\|\mathbf{y} - \frac{1}{\sqrt{d}}\mathbf{X}\boldsymbol{\beta}\|^2.$$

*The learnt function is*

$$f_m(\mathbf{x}; \mathbf{W}_\infty) = \frac{1}{\sqrt{d}}\mathbf{x}^\top \left( \sum_{j=1}^{m} \sqrt{\lambda_{m,j}} a_j \mathbf{w}_{\infty j} \right) = \frac{1}{\sqrt{d}}\mathbf{x}^\top \left( \boldsymbol{\beta}_\infty + (I_d - \mathbf{V}\mathbf{V}^\top)\boldsymbol{\beta}_0 \right). \tag{20}$$

The proof of this theorem is given in Appendix I.1.1 in the Supplementary Material.

Theorem 7.4 says that along the dimensions spanned by the data, the weight vector of a node $j$ moves by a quantity proportional to $\sqrt{\lambda_{m,j}}$ towards the minimum-norm solution. The form of the learnt function in Equation (20) implies that the contribution of each hidden node $j$ to the function's output is proportional to $\sqrt{\lambda_{m,j}} a_j \mathbf{w}_{\infty j}$. The next theorem analyses the asymptotic behaviour of this contribution in the infinite-width limit and the associated feature learning properties. It shows that, under the scaling of Equation (1), both feature learning and non-uniform feature learning occur if and only if $\gamma < 1$.

**Theorem 7.5** (Feature learning - linear activation). *Assume the setting of Theorem 7.4, and that Assumption 3.3 holds.*

---

[2]Note that we assume none of Assumptions 3.1 to 3.3 here.
[3]with a step-size less than $d(D_{\max}^2 \sum_k \lambda_{m,k})^{-1}$, where $D_{\max}$ is the largest entry of $\mathbf{D}$.

- *Under the node scalings (1), both feature learning (Definition 7.1) and non-uniform feature learning (Definition 7.3) hold if and only if $\gamma < 1$.*

- *Under the mean-field scaling, feature learning (Definition 7.1) holds, but non-uniform feature learning (Definition 7.3) does not.*

*The contribution from the $j$th node is marginally normally distributed with*

$$\sqrt{\lambda_{m,j}}a_j\mathbf{w}_{\infty j} \ \sim \ \mathcal{N}\left(\frac{\lambda_{m,j}}{\sum_k \lambda_{m,k}}\boldsymbol{\beta}_\infty, \ \lambda_{m,j}\left(I_d - \frac{\lambda_{m,j}}{\sum_k \lambda_{m,k}}\mathbf{V}\mathbf{V}^\top\right)\right) \quad \textit{for all } m \geq 1 \textit{ and } 1 \leq j \leq m.$$

*This implies that, under the mean-field scaling, contributions are asymptotically vanishing. Under the scaling (1), we have*

$$\sqrt{\lambda_{m,j}}a_j\mathbf{w}_{\infty j} \ \overset{\mathrm{d}}{\to} \ \mathcal{N}\left((1-\gamma)\widetilde{\lambda}_j\boldsymbol{\beta}_\infty, \ (1-\gamma)\widetilde{\lambda}_j\left(I_d - (1-\gamma)\widetilde{\lambda}_j\mathbf{V}\mathbf{V}^\top\right)\right) \quad \textit{as } m \to \infty. \tag{21}$$

*As a result, if $\gamma < 1$ and $\widetilde{\lambda}_j > 0$, the contribution from the $j$th node to the output is non-vanishing in the infinite-width limit.*

Before moving on to the case of the nonlinear activation function, we analyse the consequence of pruning nodes of a linear network based on the scaling parameters $\lambda_{m,j}$. The result of this analysis is given in the following proposition, which shows the benefit of our asymmetric scaling in pruning.

**Proposition 7.6.** *Assume the setting of Theorem 7.4, and that Assumption 3.3 holds. Assume also the node scaling (1). Let $\rho \in (0,1)$. Consider the following pruned network that is obtained by keeping the $\lfloor \rho m \rfloor$ hidden nodes with largest scalings $\lambda_{m,j}$ and pruning the other nodes:*

$$\widetilde{f}_{m,\rho}(\mathbf{x};\mathbf{W}_\infty) = \left(\sum_{j=1}^{\lfloor \rho m \rfloor}\sqrt{\lambda_{m,j}}a_j\frac{\mathbf{w}_{\infty j}^\top\mathbf{x}}{\sqrt{d}}\right) = \frac{1}{\sqrt{d}}\mathbf{x}^\top\left(\sum_{j=1}^{\lfloor \rho m \rfloor}\sqrt{\lambda_{m,j}}a_j\mathbf{w}_{\infty j}\right).$$

*Then, for all $\varepsilon > 0$, we have the following bound on the pruning error:*

$$\Pr\left(\left|\widetilde{f}_{m,\rho}(\mathbf{x};\mathbf{W}_\infty) - f_m(\mathbf{x};\mathbf{W}_\infty)\right| > \varepsilon\right) \ \leq \ \frac{\|\mathbf{x}\|}{\varepsilon\sqrt{d}}\left(\left(\|\boldsymbol{\beta}_\infty\| + \sqrt{d}\right)\left(\sum_{j>\lfloor \rho m \rfloor}\lambda_{m,j}\right) + \sqrt{d\sum_{j>\lfloor \rho m \rfloor}\lambda_{m,j}}\right).$$

*Since $\sum_{j>\lfloor \rho m \rfloor}\lambda_{m,j} \to \gamma(1-\rho)$ as $m \to \infty$, the above bound implies the following. If $\gamma = 0$ (no symmetric part), then $\Pr(|\widetilde{f}_{m,\rho}(\mathbf{x};\mathbf{W}_\infty) - f_m(\mathbf{x};\mathbf{W}_\infty)| > \varepsilon) \to 0$ and so the network can be compressed to a smaller network via pruning. Otherwise, the pruning error is controlled by the proportion of the symmetric part $\gamma$ in the infinite-width limit:*

$$\lim_{m\to\infty}\Pr\left(\left|\widetilde{f}_{m,\rho}(\mathbf{x};\mathbf{W}_\infty) - f_m(\mathbf{x};\mathbf{W}_\infty)\right| > \varepsilon\right) \ \leq \ \frac{\|\mathbf{x}\|}{\varepsilon\sqrt{d}}\left(\left(\|\boldsymbol{\beta}_\infty\| + \sqrt{d}\right)\gamma(1-\rho) + \sqrt{d\gamma(1-\rho)}\right).$$

### 7.3 Nonlinear activation function

We now analyse feature learning in our model when the activation function is nonlinear. Our analysis assumes the following two changes in our setup:

**Assumption 7.7** (Zeroed initialisation). We assume the model has the following form:

$$f_m(\mathbf{x};\mathbf{W}) = \left(\sum_{j=1}^m\sqrt{\lambda_{m,j}}a_j\sigma(Z_j(\mathbf{x};\mathbf{W}))\right) - \left(\sum_{j=1}^m\sqrt{\lambda_{m,j}}a_j\sigma(Z_j(\mathbf{x};\mathbf{W}_0))\right).$$

That is, we subtract from the original model, a duplicate version whose parameters are set to $\mathbf{W}_0$ and are unchanging throughout training. This is a commonly used simplification in theoretical analyses of neural networks. It ensures that at initialisation, the model satisfies

$$f_m(\mathbf{x};\mathbf{W}_0) = 0 \text{ for all } \mathbf{x} \in \mathbb{R}^d.$$

**Assumption 7.8** (Random outputs). We regard the outputs $y_1, \ldots, y_n$ as random variables, so that the probabilities in Equations (17) and (18) refer to the randomness of the outputs as well. We further assume that $y_1, \ldots, y_n$ are independent and continuous (i.e., the distribution of $y_i$ has a density with respect to Lebesgue measure), and that they are also independent from $\mathbf{W}_0$ and the $a_j$'s. Note that we still treat the inputs $x_1, \ldots, x_n$ as deterministic variables. This assumption is met if, for example, there exists a true generating function $f^*$ such that $y_i = f^*(x_i) + \epsilon_i(x_i)$ where the $\epsilon_i(x_i)$ are independent continuous noise variables.

Our analysis on feature learning considers gradient descent with a learning rate $\eta$ that does not depend on $m$ just as we did in Section 6. This is in contrast to parameterisations such as mean-field parameterisation where the learning rate has a scaling dependent on $m$. However, let us note that key reasoning steps in our proofs also apply to such $m$-dependent learning rates after minor modifications, allowing us to recover existing feature-learning results as we will explain shortly after Theorem 7.9. The full proofs of all the theorems in this subsection are given in Appendix I in the Supplementary Material. Also, the theorems in this subsection have counterparts that hold for the ReLU activation function. Appendix A.2 in the Supplementary Material contains those feature-learning results for the ReLU case.

We show that if the activation function is continuously differentiable and its derivative is always positive, as in the case of sigmoid, then after the first gradient-descent step, (i) both feature learning and non-uniform feature learning in Definitions 7.1 and 7.3 occur (Theorem 7.9) and (ii) the squared norm of each weight vector $\mathbf{w}_j$ changes almost surely by the amount $\Omega(\widetilde{\lambda}_1)$ in the infinite-width limit (Theorem 7.10).

**Theorem 7.9.** *Suppose that Assumptions 3.1, 3.3, 7.7 and 7.8 hold. Suppose also that $\gamma < 1$ and that the activation function $\sigma$ is continuously differentiable with $\sigma'(x) > 0$ for all $x \in \mathbb{R}$. Let $i \in [n]$. Then, both feature learning and non-uniform feature learning occur after the first gradient-descent step with respect to the input $\mathbf{x}_i$ in the almost-sure sense, i.e., the following inequalities hold almost surely:*

$$\liminf_{m \to \infty} \frac{\sum_{j=1}^m \lambda_{m,j} \Big( \sigma(Z_j(\mathbf{x}_i; \mathbf{W}_1)) - \sigma(Z_j(\mathbf{x}_i; \mathbf{W}_0)) \Big)^2}{\sum_{j=1}^m \lambda_{m,j} \Big( \sigma(Z_j(\mathbf{x}_i; \mathbf{W}_0)) \Big)^2} > 0$$

$$and \qquad \liminf_{m \to \infty} \frac{\max_{j \in [m]} \lambda_{m,j} \Big( \sigma(Z_j(\mathbf{x}_i; \mathbf{W}_1)) - \sigma(Z_j(\mathbf{x}_i; \mathbf{W}_0)) \Big)^2}{\sum_{j=1}^m \lambda_{m,j} \Big( \sigma(Z_j(\mathbf{x}_i; \mathbf{W}_0)) \Big)^2} > 0.$$

**Sketch of the proof.** Since non-uniform feature learning implies feature learning, we prove the former only. The denominator $\sum_{j=1}^m \lambda_{m,j} (\sigma(Z_j(\mathbf{x}_i; \mathbf{W}_0)))^2$ in the condition for non-uniform feature learning converges to a positive finite value almost surely as $m$ tends to $\infty$. Thus, it is enough to prove that

$$\liminf_{m \to \infty} \left( \max_{j \in [m]} \lambda_{m,j} \left( \sigma(Z_j(\mathbf{x}_i; \mathbf{W}_1)) - \sigma(Z_j(\mathbf{x}_i; \mathbf{W}_0)) \right)^2 \right) > 0 \quad \text{almost surely,}$$

which is implied by

$$\liminf_{m \to \infty} \left( \sigma(Z_1(\mathbf{x}_i; \mathbf{W}_1)) - \sigma(Z_1(\mathbf{x}_i; \mathbf{W}_0)) \right)^2 > 0 \quad \text{almost surely.} \tag{22}$$

Note that the limits from above are not redundant since $\mathbf{W}_1$ depends on $m$. The sufficient condition in Equation (22) can be simplified further. The assumptions of the theorem allow us to use the inverse function theorem to deduce that the condition in Equation (22) holds whenever

$$\liminf_{m \to \infty} \left( Z_1(\mathbf{x}_i; \mathbf{W}_1) - Z_1(\mathbf{x}_i; \mathbf{W}_0) \right)^2 > 0 \quad \text{almost surely.} \tag{23}$$

The majority of the detailed proof concerns proving the condition in Equation (23). To that end, we compute the following $m$-independent lower bound: for all $m$,

$$\left(Z_1(\mathbf{x}_i; \mathbf{W}_1) - Z_1(\mathbf{x}_i; \mathbf{W}_0)\right)^2 = \frac{\eta^2 \lambda_{m,1}}{d^2} \left(\sum_{i'=1}^{n} y_{i'} \left(\sigma'\left(\frac{\mathbf{w}_{01}^\top \mathbf{x}_{i'}}{\sqrt{d}}\right) \mathbf{x}_{i'}^\top \mathbf{x}_i\right)\right)^2 \tag{24}$$

$$\geq \frac{\eta^2 (1-\gamma)\widetilde{\lambda}_1}{d^2} \left(\sum_{i'=1}^{n} y_{i'} \left(\sigma'\left(\frac{\mathbf{w}_{01}^\top \mathbf{x}_{i'}}{\sqrt{d}}\right) \mathbf{x}_{i'}^\top \mathbf{x}_i\right)\right)^2. \tag{25}$$

Then, we show that the right-hand side is almost surely positive, which implies the conclusion of the theorem. The justification of this almost-sure positivity relies on Assumptions 3.1 and 7.8, the positivity of $\sigma'$, and the assumption that $\gamma < 1$. Concretely, the assumption on $\gamma$ implies that $\eta^2(1-\gamma)\widetilde{\lambda}_1/d^2 > 0$, while Assumptions 3.1 and 7.8 and the positivity of $\sigma'$ imply that the squared sum on the right-hand side is almost surely a positive random variable.

We point out that the proof also reveals that if we were to allow the learning rate $\eta$ to depend on $m$ in a particular way, then feature learning could occur even for $\gamma = 1$ (although the condition for non-uniform feature learning might fail). For instance, when $\eta = \sqrt{m}$ and $\gamma = 1$, Equation (24) still holds, and its right-hand side is almost surely positive because $\eta^2 \lambda_{m,1}/d^2 = 1/d^2 > 0$. This almost-sure positivity and the assumptions on $\sigma$ then imply that

$$\mathbb{E}[(\sigma(Z_1(\mathbf{x}_i; \mathbf{W}_1)) - \sigma(Z_1(\mathbf{x}_i; \mathbf{W}_0)))^2] > 0, \tag{26}$$

which in turn ensures that the condition for feature learning holds, i.e., almost surely,

$$\liminf_{m \to \infty} \frac{\sum_{j=1}^{m} \lambda_{m,j} \left(\sigma(Z_j(\mathbf{x}_i; \mathbf{W}_1)) - \sigma(Z_j(\mathbf{x}_i; \mathbf{W}_0))\right)^2}{\sum_{j=1}^{m} \lambda_{m,j} \left(\sigma(Z_j(\mathbf{x}_i; \mathbf{W}_0))\right)^2} = \frac{\lim_{m \to \infty} \frac{1}{m} \sum_{j=1}^{m} \left(\sigma(Z_j(\mathbf{x}_i; \mathbf{W}_1)) - \sigma(Z_j(\mathbf{x}_i; \mathbf{W}_0))\right)^2}{\lim_{m \to \infty} \frac{1}{m} \sum_{j=1}^{m} \sigma(Z_j(\mathbf{x}_i; \mathbf{W}_0))^2}$$

$$= \frac{\mathbb{E}\left[\left(\sigma(Z_1(\mathbf{x}_i; \mathbf{W}_1)) - \sigma(Z_1(\mathbf{x}_i; \mathbf{W}_0))\right)^2\right]}{\mathbb{E}[\sigma(Z_1(\mathbf{x}_i; \mathbf{W}_0))^2]} > 0.$$

However, note that in this case, by (Yang & Hu, 2021, Theorem 3.3), the model becomes unstable and could blow up (cf. the footnote regarding Equation (17)). Also, if $\eta = m$ and we were to use $1/m^2$ for $\lambda_{m,j}$ (i.e., mean-field parameterisation), the essentially same argument would apply and lead to feature learning, although such $\lambda_{m,j}$'s are not covered by our setup. Just as before, Equation (24) would hold and its right-hand side would be almost surely positive in this case, because $\eta^2 \lambda_{m,1}/d^2 = 1/d^2 > 0$. This almost-sure positivity implies the inequality in Equation (26), which then gives the condition for feature learning because the $\liminf$ formula in the definition of feature learning can be simplified to the same ratio of expectations as the one before.

**Theorem 7.10.** *Suppose Assumptions 3.1, 3.3, 7.7 and 7.8 hold. Suppose also that the activation function $\sigma$ is continuously differentiable and satisfies $\sigma'(x) > 0$ for all $x \in \mathbb{R}$. Then, for all $j$, the following holds almost surely:*

$$\liminf_{m \to \infty} \left\| \nabla_{\mathbf{w}_{tj}} L(\mathbf{W}_t)\big|_{t=0} \right\|^2 \geq \frac{(1-\gamma)\widetilde{\lambda}_j}{d} \left| \sum_{i=1}^{n} \sum_{i'=1}^{n} y_i y_{i'} \left(\mathbf{x}_i^\top \mathbf{x}_{i'} \sigma'\left(\frac{\mathbf{w}_{0j}^\top \mathbf{x}_i}{\sqrt{d}}\right) \sigma'\left(\frac{\mathbf{w}_{0j}^\top \mathbf{x}_{i'}}{\sqrt{d}}\right)\right) \right|. \tag{27}$$

*In particular, if $\gamma < 1$ and $\widetilde{\lambda}_j > 0$, then the lower bound is positive almost surely so that we have*

$$\liminf_{m \to \infty} \left\| \nabla_{\mathbf{w}_{tj}} L(\mathbf{W}_t)\big|_{t=0} \right\|^2 > 0$$

*with probability $1$.*

**Sketch of the proof.** The lower bound in Equation (27) is obtained by a relatively straightforward calculation since the calculation of the gradient is simplified due to Assumption 7.7. The second part of the theorem follows from the fact that when conditioned on $\mathbf{w}_{0j}$, the absolute value on the right-hand side of Equation (27) is a continuous random variable that depends only on the $y_i$'s, and so it is strictly positive with probability one.

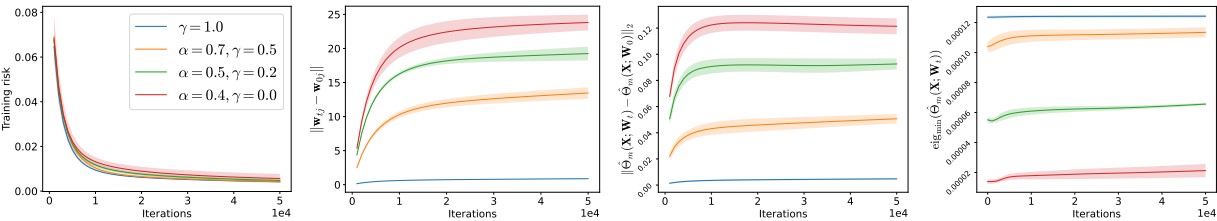

Figure 1: Results on simulated data. From left to right, 1) training risks, 2) differences in weight norms $\|\mathbf{w}_{tj} - \mathbf{w}_{0j}\|$ with the $j$'s being those neurons which have maximal differences at the end of the training, 3) differences in NTG matrices, and 4) minimum eigenvalues of NTG matrices.

# 8 Experiments

We use here a (smooth) swish activation function $\sigma(z) = z/(1 + e^{-z})$. We obtained quantitatively similar results with the ReLU activation function; see Appendix L in the Supplementary Material.

## 8.1 Simulated data.

### 8.1.1 Illustration of the main results

We first illustrate our theory on simulated data.[4] We generate $n = 100$ observations where for $i \in [n]$, $\mathbf{x}_i$ is $d = 50$ dimensional and sampled uniformly on the unit sphere, and $y_i = \frac{5}{d} \sum_{j=1}^{d} \sin(\pi x_{i,j}) + \varepsilon_i$ where $\varepsilon_i \overset{\text{iid}}{\sim} \mathcal{N}(0, 1)$. We use the FFNN of Section 3.1, with the swish activation function, $m = 2000$ hidden nodes, and $\lambda_{m,j}$ as in Equations (1) and (4). We consider the four values $(\gamma, \alpha) \in \{(1, -), (0.5, 0.7), (0.2, 0.5), (0, 0.4)\}$. For each setting, we run GD with a learning rate of 1.0 for 50 000 steps, which is repeated five times to get average results. We summarise the results in Figure 1, which shows the training error and the evolution of the weights, NTG, and minimum eigenvalue of the NTG as a function of the GD iterations. We see a clear correspondence between the theory and the empirical results. For $\gamma > 0$, GD achieves near-zero training error. The minimum eigenvalue and training rates increase with $\gamma$. For $\gamma = 1$, we have the highest minimum eigenvalue and the fastest training rate, but there is no/very little feature learning: the weights and the NTG do not change significantly over the GD iterations. When $\gamma < 1$, there is clear evidence of feature learning: both the weights and the NTG change significantly over time; the smaller the $\gamma$ and $\alpha$, the more feature learning arises.

### 8.1.2 Improved generalization: Single ReLU experiment.

To illustrate the benefits of asymmetrical scaling, we consider here the scenario where the function to learn is a single-unit ReLU, a setting known to be challenging for the lazy-training regime (Malach et al., 2021). Consider the following data-generating process:

$$
\begin{aligned}
X &\sim \mathcal{N}(0, I_d) \\
Y &= \sigma(w_0^T X)
\end{aligned}
$$

where $w_0 = (1, .., 1) \in \mathbb{R}^d$ and $\sigma$ is the ReLU activation function. In this experiment, we sample a training dataset of $n = 100$ samples in dimension $d = 10$. As previously, we train fully connected neural networks composed of a single hidden layer, with different node-scaling strategies. The width in all models is set to $P = 2000$. The generalization error is computed on 5000 samples from the same single ReLU data-generating process. All experiments are repeated 5 times, the training and testing datasets are resampled for each run. The results are reported in Figure 2. We notice that at the end of the training, the error is near zero for all scalings. Examining the error on the test set, one can see that, as expected, the symmetrical $\gamma = 1$ NTK scaling generalizes poorly in this setting. The asymmetrical scalings, on the other hand, perform significantly

---

[4]The code can found at `https://github.com/juho-lee/asymmetrical_scaling`

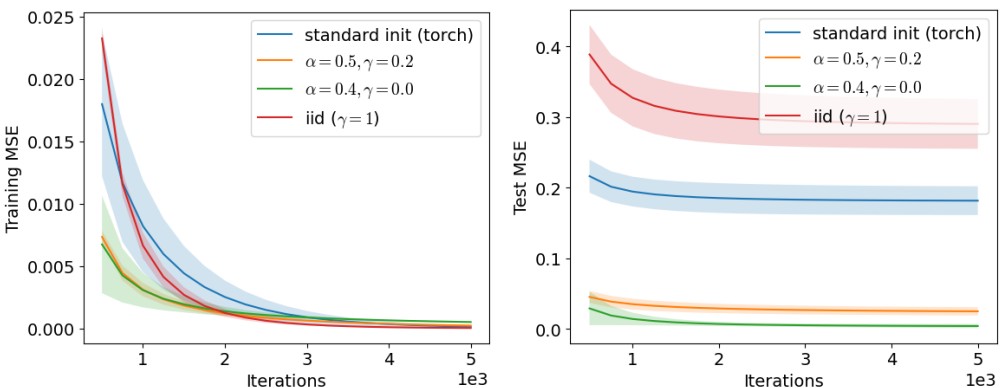

Figure 2: Results on simulated data from a single ReLU unit. Evolution of the training error (left) and test error (right) as a function of the training iteration.

better, illustrating the benefits of this strategy in terms of generalization error. For completeness, we also report the results using the standard pytorch initialization. We can see that in this setting of a sparse generating process, the standard initialization also generalizes poorly compared to the asymmetrical model.

## 8.2   Real data

### 8.2.1   Regression

We also validate our model on four regression datasets from the UCI repository[5]: `concrete` $((n,d)=(1030,9))$, `energy` $((n,d)=(768,8))$, `airfoil` $((n,d)=(1503,6))$, and `plant` $((n,d)=(9568,4))$. We split each dataset into training (40%) , test (20%), and validation sets (40%), and use the validation set to test transfer learning. We use the same parameters as for the simulated data, but train our FFNNs for 100 000 steps in each run. To further highlight the presence of feature learning in our model, we test the transferability of features learnt from our networks as follows. We first split the validation set into a held-out training set (50%) and a test set (50%), and extract features of the held-out training set using the FFNNs trained on the original training set. Features are taken to be the outputs of the hidden layers, so each data point in the validation set is represented with a $m=2000$ dimensional vector. Then, we sort feature dimensions with respect to feature importance measured as $(\lambda_{m,j}\|\mathbf{w}_{tj}\|^2)_{j\in[m]}$ and use the top-$k$ of these to train an external model. The chosen external model is a FFNN with a single hidden layer having 64 neurons and ReLU activation, which is trained for 5000 GD steps with a learning rate of 1.0. Our theory suggests that smaller $\gamma$ and $\alpha$ likely lead to better transfer learning. A subset of our results appears in Figure 3; see Appendix K for additional results. In line with the simulated data experiments, we observe a stronger presence of feature learning, in terms of weight-norm changes and NTG changes, for smaller values of $\gamma$ and $\alpha$. Also, we observe that models with smaller values of $\gamma$ have lower risks when a small number of features are used for the transfer. The interpretation is that those models are able to learn a sufficient number of representative features using relatively fewer neurons.

### 8.2.2   Classification

We apply our model on two image classification tasks. The first is small-scale using the setting assumed in our theory, while the second is larger-scale using a more realistic setting. In addition to the transferability experiment described before, we test the prunability of the FFNNs. We gradually prune hidden nodes with small feature importance and measure risks after pruning. Feature importance is measured as above. Our theory suggests that models with smaller $\gamma$ and/or $\alpha$ values are likely to be more robust with respect to pruning, as long as $\gamma < 1$. Wolinski et al. (2020) had similar empirical findings on the benefits of asymmetrical scaling for network pruning when $\gamma = 0$.

---

[5]https://archive.ics.uci.edu/ml/datasets.php

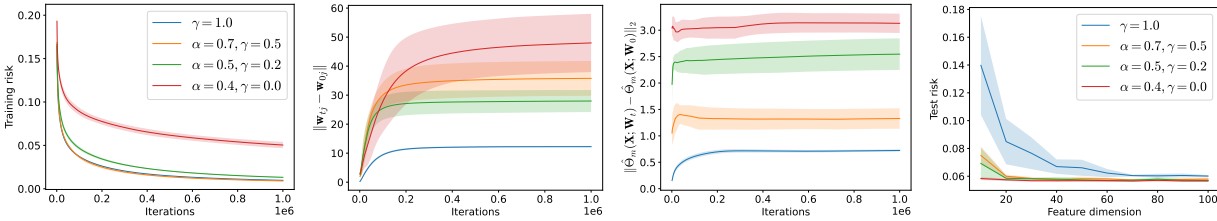

Figure 3: A subset of results for the regression experiments. From left to right, 1) training risks for dataset `concrete` , 2) differences in weight norms $\|\mathbf{w}_{tj} - \mathbf{w}_{0j}\|$ with the $j$'s being the neurons having the maximal difference at the end of the training for dataset `energy`, 3) differences in NTG matrices for dataset `airfoil`, and 4) test risks of transferred models for dataset `plant`.

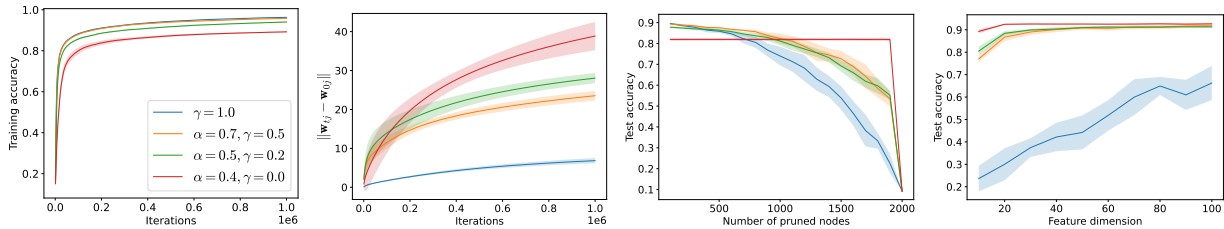

Figure 4: A subset of results for MNIST dataset. From left to right, 1) training risks, 2) differences in weight norms, 3) test accuracies of pruned models, and 4) test accuracies of transferred models.

**MNIST.** We take a subset of size 5000 from the MNIST dataset and train the same models used in the previous experiments. We also test pruning and transfer learning, where we use an additional subset of size 5000 to train an external FFNN having a single hidden layer with 128 nodes. To match our theory, instead of using cross-entropy loss, we use the MSE loss by treating one-hot class labels as continuous-valued targets. The outputs of the models are 10 dimensional, so we compute the NTG matrices using only the first dimension of the outputs. In general, we get similar results in line with our previous experiments. The pruning and transfer learning results are displayed in Figure 4. For other results, see Figure 6 in the Supplementary Material.

**CIFAR.** We consider a more challenging image classification task of CIFAR–10 and CIFAR–100. The datasets have 60 000 images with 50 000 for used training and the rest used for testing. There are, respectively, 10 and 100 different classes. We show the benefits of asymmetrical node scaling hold for this more challenging problem. In many applications, one uses a large model pre-trained on a general task and then performs fine-tuning or transfer learning to adapt it to the task at hand. We implement this approach on a ResNet-18 model, pre-trained on ImageNet data. With this model, we transform each original image to a vector of dimension 512. We then train shallow FFNNs as described in Section 3.1, with $m = 2000$ and output dimension 10 (resp. 100). This experiment differs from previous cases as 1) we use stochastic GD with a mini-batch size of 64 instead of full batch GD; 2) we use cross-entropy loss instead of MSE; and 3) both layers are trained. All experiments are run five times, with the learning rate 5.0. Figure 5 shows the pruning results for the same four values of pairs $(\gamma, \alpha)$ as above, for CIFAR–100. Similar results are obtained for CIFAR–10; see Appendix K. Similar conclusions as before hold here, even though the theory does not apply directly.

## 9 Discussion, limitations and further work

We have shown that under an asymmetrical scaling of the nodes of a neural network, it is possible to achieve both zero training error and feature learning, when the width of the neural network is sufficiently large. We considered two definitions of feature learning. The first definition is a minor generalisation of the notion of feature learning from (Yang & Hu, 2021), and it is defined as a change in the feature map. We proposed a second definition, called non-uniform feature learning, which additionally requires that the contributions of some individual nodes remain non-negligible in the asymptotic limit. We showed that under our asymmetric scaling and additional conditions, both definitions hold, whereas for the standard NTK, neither does, and for

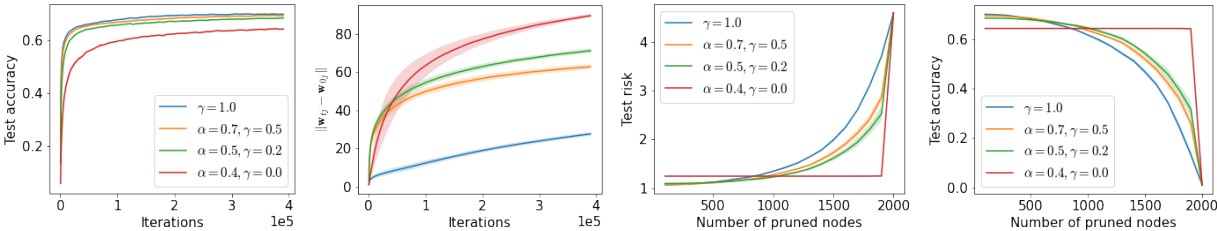

Figure 5: Results for `CIFAR-100`. From left to right, 1) test risk through training, 2) differences in weight norms $\|\mathbf{w}_{tj} - \mathbf{w}_{0j}\|$ with the $j$'s being the neurons having the maximal difference at the end of training, 3) test risks of pruned models, and 4) test accuracies of pruned models.

the mean field, only the standard definition holds. We demonstrate empirically that having non-uniform feature learning is particularly important when we consider transfer learning and pruning. Our definitions of feature learning relate to the change in the feature map. As already noted by (Yang & Hu, 2021), it is a relatively weak definition of feature learning, as it does not connect the weight change with the generalisation properties. Experimentally, we found that in some case (e.g. single ReLU), the asymmetrical, unpruned network provides the best test error, while in others (MNIST and CIFAR), the unpruned symmetrical scaling gave the best test accuracy. An interesting avenue of research is to investigate theoretically the generalisation properties of such asymmetrical scaling. We note that the approaches used for the symmetric NTK (Arora et al., 2019a), which rely on the limiting kernel, cannot be applied to our setting, due to the evolving kernel.

In this article, we have assumed an iid Gaussian initialisation for the weights (Assumption 3.3), which is a standard assumption in the analysis of large-width neural networks (Du et al., 2019b;a; Oymak & Soltanolkotabi, 2020; Nguyen et al., 2021). Our results rely on the fact that the minimum eigenvalue $\kappa_n$ of the mean NTK at initialisation is strictly positive; this result was demonstrated by (Du et al., 2019a, Proposition F.1) under the iid Gaussian initialisation. An interesting direction of research would be to investigate whether the results derived in this paper hold under other, possibly non-iid, initialisation schemes. In particular, the case of orthogonal initialisations would be of particular interest (Hu et al., 2020; Huang et al., 2021).

The asymmetrical parameterisation in Equation (1) is rather general, and only requires the $\widetilde{\lambda}_j$ to be summable. A natural default choice, taken in this article, is to take a power function $\propto j^{-1/\alpha}$ where $0 < \alpha < 1$. Other parameterisations are also possible, such as $\widetilde{\lambda}_j = 1/K$ for $j = 1, \ldots, K$ and 0 otherwise. One could also choose other scalings such as $(e - 1)\exp(-j)$ or $C/(j\log^2(j + 1))$.

## Acknowledgements

We would like to thank Taeyoung Kim for helpful discussions, and the anonymous reviewers for their useful comments that helped improve the paper. HY was supported by the National Research Foundation of Korea (NRF) grant funded by the Korean Government (MSIT) (No. RS-2023-00279680). JL acknowledges support from Institute for Information & communications Technology Planning & Evaluation(IITP) grant funded by the Korea government(MSIT) (RS-2019-II190075, Artificial Intelligence Graduate School Program(KAIST)).

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
