# Over-parameterised Shallow Neural Networks with Asymmetrical Node Scaling: Global Convergence Guarantees and Feature Learning: Supplementary Material

**François Caron**  *caron@stats.ox.ac.uk*
*Department of Statistics*
*University of Oxford, United Kingdom*

**Fadhel Ayed**  *fadhel.ayed@gmail.com*
*Huawei Technologies*
*Paris, France*

**Paul Jung**  *paul.jung@gmail.com*
*Department of Mathematics*
*Fordham University, USA*

**Hoil Lee**  *hoil.lee@alumni.kaist.ac.kr*
*Samsung SDS*
*South Korea*

**Juho Lee**  *juholee@kaist.ac.kr*
*Kim Jaechul Graduate School of AI*
*KAIST, South Korea*

**Hongseok Yang**  *hongseok.yang@kaist.ac.kr*
*School of Computing*
*KAIST, South Korea*

**Reviewed on OpenReview:** *https://openreview.net/forum?id=Sx1khIIi95*

## Abstract

This Supplementary Material is organised as follows. Appendix A presents additional results on global convergence and feature learning when the activation function is the (non-smooth) ReLU function. In particular, Theorem A.1 states conditions for the global convergence of gradient flow in the ReLU case, and is similar to Theorem 5.1 (smooth case) in the main paper. Appendix A.3 discusses some open problems in our framework when dealing with the ReLU activation function. Useful bounds and identities are presented in Appendix B. Appendix C gives a proof of the proposition regarding the structure of the limiting NTG at initialisation while Appendix D provides a secondary proposition regarding the minimum eigenvalue of the NTG at initialisation. Appendix E states and proves secondary lemmas on gradient flow dynamics. Appendix F and G give details of the main proof for global convergence of gradient flow, respectively for the ReLU and smooth case. The proofs are rather short and mostly build on the secondary lemmas and propositions of Appendices D and E. Appendix H gives a detailed proof for global convergence of gradient descent in the smooth case. The proof builds on results of convergence of gradient flow. Appendix I gives proofs of the feature-learning results for the smooth case in Section 7, and Appendix J presents proofs of the corresponding feature-learning results for the ReLU case in Appendix A.2. Appendix K provides additional experiments to those of Section 8, under a smooth activation. Finally, Appendix L provides detailed results on the same experiments as in Section 8, but with the ReLU activation instead of the Swish activation function used in the main paper.

# Contents

## List of Figures

# A  Results for the ReLU activation function

Although we assume a smooth activation function in the main text of the paper (Assumption 3.2), some of the results remain true when we drop this assumption and use the ReLU activation function instead. In this section, we explain these results for ReLU. Throughout the section, we assume a weak derivative $\sigma'(x) = \mathbf{1}_{\{x>0\}}$ of the ReLU activation function $\sigma$.

## A.1  Global convergence under gradient flow

Our global convergence theorem under gradient flow in the main text (Theorem 5.1) has a counterpart for the ReLU case, which is given below. This counterpart says that when we train the network with the ReLU activation, with high probability, the loss decays exponentially fast with respect to $\kappa_n$ and the training time $t$, and the weights $\mathbf{w}_{tj}$ and the NTG matrix respectively change by

$$O\left(\frac{n\lambda_{m,j}^{1/2}}{\kappa_n d^{1/2}}\right) \quad \text{and} \quad O\left(\frac{n^2 \sum_{j=1}^m \lambda_{m,j}^{3/2}}{\kappa_n d^{3/2}} + \frac{n^{3/2}\sqrt{\sum_{j=1}^m \lambda_{m,j}^{3/2}}}{\kappa_n^{1/2} d^{5/4}}\right).$$

**Theorem A.1** (Global convergence, gradient flow, ReLU)**.** *Consider $\delta \in (0,1)$. Let $D_0 = \sqrt{2C^2 + (2/d)}$. Assume Assumptions 3.1 and 3.3, and the use of the ReLU activation function. Also, assume $\gamma > 0$ and*

$$m \geq \max\left(\frac{2^3 n \log \frac{4n}{\delta}}{\kappa_n d}, \frac{2^{25} n^4 D_0^2}{\kappa_n^4 d^3 \gamma^2 \delta^5}, \frac{2^{35} n^6 D_0^2}{\kappa_n^6 d^5 \gamma^2 \delta^5}\right).$$

*Then, with probability at least $1 - \delta$, the following properties hold for all $t \geq 0$:*

*(a) $\mathrm{eig}_{\min}(\widehat{\Theta}_m(\mathbf{X}; \mathbf{W}_t)) \geq \frac{\gamma\kappa_n}{4}$;*
*(b) $L_m(\mathbf{W}_t) \leq e^{-(\gamma\kappa_n t)/2} L_m(\mathbf{W}_0)$;*
*(c) $\|\mathbf{w}_{tj} - \mathbf{w}_{0j}\| \leq \frac{2^3 n D_0}{\kappa_n d^{1/2}\gamma\delta^{1/2}}\sqrt{\lambda_{m,j}}$ for all $j \in [m]$;*
*(d) $\|\widehat{\Theta}_m(\mathbf{X}; \mathbf{W}_t) - \widehat{\Theta}_m(\mathbf{X}; \mathbf{W}_0)\|_2 \leq \left(\frac{2^9 n^2 D_0}{\kappa_n d^{3/2}\gamma\delta^{5/2}} \cdot \sum_{j=1}^m \lambda_{m,j}^{3/2}\right) + \left(\frac{2^6 n^{3/2} D_0^{1/2}}{\kappa_n^{1/2} d^{5/4}\gamma^{1/2}\delta^{5/4}} \cdot \sqrt{\sum_{j=1}^m \lambda_{m,j}^{3/2}}\right).$*

The proof of the theorem is given in Appendix F, and uses Lemmas E.1 to E.3 and Proposition D.1.

The theorem guarantees that whenever $\gamma > 0$, the training error converges to 0 exponentially fast. Also, it implies that the weight change is bounded by a factor $\sqrt{\lambda_{m,j}}$, and the NTG change is bounded by a factor $\sqrt{\sum_{j=1}^m \lambda_{m,j}^{3/2}}$. As we show in Appendix B.2, as $m$ tends to $\infty$,

$$\lambda_{m,j} \to (1-\gamma)\widetilde{\lambda}_j \text{ for every } j \geq 1, \quad \text{and} \quad \sum_{j=1}^m \lambda_{m,j}^{3/2} \to (1-\gamma)^{3/2}\sum_{j=1}^\infty \widetilde{\lambda}_j^{3/2}.$$

Thus, when $\widetilde{\lambda}_j > 0$ (note that we necessarily have $\widetilde{\lambda}_1 > 0$), the upper bound in (c) is vanishing in the infinite-width limit if and only if $\gamma = 1$ (NTK regime); similarly, the upper bound in (d) is vanishing if and only if $\gamma = 1$. In fact, both feature learning and non-uniform feature learning in high-probability versions of Definitions 7.1 and 7.3 occur whenever $\gamma < 1$, as we will show in the next subsection.

## A.2  Results on feature learning

We present feature-learning results for the ReLU activation. The proofs of the theorems in this subsection appear in Appendix J

We start with a result that corresponds to Theorem 7.9 in the smooth-activation case. The result says that if $\gamma < 1$ and the activation function is ReLU, then after the first step of gradient descent, both feature learning and non-uniform feature learning occur in a slightly weaker sense than that of Definitions 7.1 and 7.3 where we have substituted the almost-sure conditions with corresponding high-probability conditions.

**Theorem A.2.** *Suppose that Assumptions 3.1, 3.3, 7.7 and 7.8 hold. Suppose also that $\gamma < 1$ and that the activation function $\sigma$ is ReLU. If $\widetilde{\lambda}_k > 0$, then with probability at least $1 - (1/2)^k$, the following inequalities hold for all $i \in [n]$:*

$$\liminf_{m \to \infty} \frac{\sum_{j=1}^m \lambda_{m,j} \Big( \sigma(Z_j(\mathbf{x}_i; \mathbf{W}_1)) - \sigma(Z_j(\mathbf{x}_i; \mathbf{W}_0)) \Big)^2}{\sum_{j=1}^m \lambda_{m,j} \Big( \sigma(Z_j(\mathbf{x}_i; \mathbf{W}_0)) \Big)^2} > 0 \tag{S.1}$$

$$and \qquad \liminf_{m \to \infty} \frac{\max_{j \in [m]} \lambda_{m,j} \Big( \sigma(Z_j(\mathbf{x}_i; \mathbf{W}_1)) - \sigma(Z_j(\mathbf{x}_i; \mathbf{W}_0)) \Big)^2}{\sum_{j=1}^m \lambda_{m,j} \Big( \sigma(Z_j(\mathbf{x}_i; \mathbf{W}_0)) \Big)^2} > 0 \tag{S.2}$$

As we mentioned already, the proof of Theorem A.2 appears in Appendix J. Here we explain the key steps of the proof. Note that the condition for non-uniform feature learning in Equation (S.2) implies that for feature learning in Equation (S.1). Thus, we focus on proving the former condition. The crux of proving the condition in Equation (S.2) lies in the derivation of the following lower bound:

$$\liminf_{m \to \infty} \left( \max_{j \in [m]} \lambda_{m,j} \Big( \sigma(Z_j(\mathbf{x}_i; \mathbf{W}_1)) - \sigma(Z_j(\mathbf{x}_i; \mathbf{W}_0)) \Big)^2 \right)$$

$$\geq \max_{j \in [k]} \left( \mathbf{1}_{\left\{ \mathbf{w}_{0j}^\top \mathbf{x}_i \geq 0 \right\}} \cdot \min \left\{ \frac{\eta^2 c^2 (1-\gamma)^2 \widetilde{\lambda}_j^2}{d^2}, \frac{(1-\gamma) \widetilde{\lambda}_j (\mathbf{w}_{0j}^\top \mathbf{x}_i)^2}{d} \right\} \right)$$

where $c$ is a positive real-valued continuous random variable that depends only on the outputs $y_1, \ldots, y_n$. In particular, $c$ does not depend on $\mathbf{W}_0$ nor $m$, and moreover $c^2 > 0$ almost surely. The assumptions of the theorem and the properties of $c$ imply that the above lower bound is strictly positive if $\mathbf{w}_{0j}^\top \mathbf{x}_i > 0$ for some $j$, and this latter condition happens with probability at least $1 - (1/2)^k$, which gives the claim of the theorem.

Recall that by definition, $\widetilde{\lambda}_1$ is always positive. Thus, Theorem A.2 implies that the inequalities in Equations (S.1) and (S.2) (which are the conditions for feature learning and non-uniform feature learning stated in Definitions 7.1 and 7.3 excepting the almost-sure condition) hold with probability at least $1/2$ for *any* choice of the node-scaling parameters. Another immediate and perhaps more important consequence of the theorem is that, if *all* the $\widetilde{\lambda}_j$'s are positive, then both feature learning and non-uniform feature learning occur, precisely in the sense of Definitions 7.1 and 7.3. This is because, in that case, the inequalities in Equations (S.1) and (S.2) hold with probability at least $1 - (1/2)^k$ for all $k$ by Theorem A.2, but this implies that both inequalities hold almost surely. The next corollary states this consequence more explicitly.

**Corollary A.3.** *Suppose Assumptions 3.1, 3.3, 7.7 and 7.8 hold. Suppose also that $\gamma < 1$ and that the activation function $\sigma$ is ReLU. Let $i \in [n]$. If $\widetilde{\lambda}_j > 0$ for all $j$, the following inequalities hold almost surely:*

$$\liminf_{m \to \infty} \frac{\sum_{j=1}^m \lambda_{m,j} \Big( \sigma(Z_j(\mathbf{x}_i; \mathbf{W}_1)) - \sigma(Z_j(\mathbf{x}_i; \mathbf{W}_0)) \Big)^2}{\sum_{j=1}^m \lambda_{m,j} \Big( \sigma(Z_j(\mathbf{x}_i; \mathbf{W}_0)) \Big)^2} > 0$$

$$and \qquad \liminf_{m \to \infty} \frac{\max_{j \in [m]} \lambda_{m,j} \Big( \sigma(Z_j(\mathbf{x}_i; \mathbf{W}_1)) - \sigma(Z_j(\mathbf{x}_i; \mathbf{W}_0)) \Big)^2}{\sum_{j=1}^m \lambda_{m,j} \Big( \sigma(Z_j(\mathbf{x}_i; \mathbf{W}_0)) \Big)^2} > 0.$$

Our next result about the ReLU activation function is a counterpart of Theorem 7.10 in the smooth-activation case. It says that for all $j$, if $\gamma < 1$ and $\widetilde{\lambda}_j > 0$, then with probability at least $1/2$, the first step of gradient descent induces a non-zero change in the squared norm of the weight vector $\mathbf{w}_j$ in the infinite-width limit. The result also suggests that the change in the squared norm is proportional to $\widetilde{\lambda}_j$.

**Theorem A.4.** *Suppose Assumptions 3.1, 3.3, 7.7 and 7.8 hold. Suppose also that the activation function $\sigma$ is ReLU. Then, for all $j$, the following holds almost surely:*

$$\liminf_{m\to\infty} \left\| \nabla_{\mathbf{w}_{tj}} L(\mathbf{W}_t)\big|_{t=0} \right\|^2 \geq \frac{(1-\gamma)\widetilde{\lambda}_j}{d} \left| \sum_{i=1}^{n} \sum_{i'=1}^{n} y_i y_{i'} \left( \mathbf{x}_i^\top \mathbf{x}_{i'} \mathbf{1}_{\{\mathbf{w}_{0j}^\top \mathbf{x}_i \geq 0\}} \mathbf{1}_{\{\mathbf{w}_{0j}^\top \mathbf{x}_{i'} \geq 0\}} \right) \right|. \tag{S.3}$$

*In particular, if $\gamma < 1$ and $\widetilde{\lambda}_j > 0$, then with probability at least $1/2$, the above lower bound is positive so that*

$$\liminf_{m\to\infty} \left\| \nabla_{\mathbf{w}_{tj}} L(\mathbf{W}_t)\big|_{t=0} \right\|^2 > 0.$$

## A.3 Discussion

Theorem A.1 is the counterpart of Theorem 5.1 for the global convergence of gradient flow with the ReLU activation function. Despite empirical evidence from Appendix L suggesting that similar convergence results could potentially be applicable to GD in the ReLU context, we have yet to substantiate this with a comprehensive proof. The proof of the global convergence of GD with smooth activation provided in Appendix H relies on a Taylor approximation. This necessitates the activation function $\sigma$ to be twice differentiable. It is worth noting that, in the symmetric NTK case, the global convergence of GD with the ReLU activation has been shown by Du et al. (2019b, Section 4). Their proof, however, critically relies on the fact that the weights remain stationary throughout the iterations of GD, which is not the scenario we are dealing with here when $\gamma > 0$. As such, it remains a compelling open question to determine whether the global convergence of GD can be proven within our specific framework when employing the ReLU activation function.

# B Useful bounds and identities

## B.1 Matrix Chernoff inequalities

The following matrix bounds can be found in (Tropp, 2012).

**Proposition B.1.** *Consider a finite sequence $(X_1, X_2, \ldots, X_p)$ of independent, random, positive semi-definite $n \times n$ matrices with $\operatorname{eig}_{\max}(X_j) \leq R$ almost surely for all $j \in [p]$, for some $R > 0$. Define*

$$\mu_{\min} = \operatorname{eig}_{\min}\left( \sum_{j=1}^{p} \mathbb{E}[X_j] \right) \quad and \quad \mu_{\max} = \operatorname{eig}_{\max}\left( \sum_{j=1}^{p} \mathbb{E}[X_j] \right).$$

*Then, for all $\delta \in [0, 1)$,*

$$\Pr\left( \operatorname{eig}_{\min}\left( \sum_{j=1}^{p} X_j \right) \leq (1-\delta)\mu_{\min} \right) \leq n \left[ \frac{e^{-\delta}}{(1-\delta)^{1-\delta}} \right]^{\mu_{\min}/R} \leq n e^{-\delta^2 \mu_{\min}/(2R)}.$$

*Also, for all $\delta \geq 0$,*

$$\Pr\left( \operatorname{eig}_{\max}\left( \sum_{j=1}^{p} X_j \right) \geq (1+\delta)\mu_{\max} \right) \leq n \left[ \frac{e^{\delta}}{(1+\delta)^{1+\delta}} \right]^{\mu_{\max}/R} \leq n e^{-\delta^2 \mu_{\max}/((2+\delta)R)}.$$

## B.2 Some identities on $(\lambda_{m,j})_{j\in[m]}$

The following proposition summarises a number of useful properties on the scaling parameters defined by (1).

**Proposition B.2.** *For all $m \geq 1$,*

$$\sum_{j=1}^{m} \lambda_{m,j} = 1, \tag{S.4}$$

$$\sqrt{\gamma m} \leq \sum_{j=1}^{m} \sqrt{\lambda_{m,j}} \leq \sqrt{m}. \tag{S.5}$$

*For every $r > 1$, as $m \to \infty$,*

$$\sum_{j=1}^{m} \lambda_{m,j}^r \sim \sum_{j=1}^{m} \left(\lambda_{m,j}^{(2)}\right)^r \to (1-\gamma)^r \sum_{j \geq 1} \widetilde{\lambda}_j^r. \tag{S.6}$$

*Finally, we have, for all $\rho \in (0,1)$,*

$$\lim_{m \to \infty} \sum_{j=\lfloor \rho m \rfloor + 1}^{m} \lambda_{m,j} = \gamma(1-\rho). \tag{S.7}$$

*Proof.* Equation (S.4) follows from the definition of $\lambda_{m,j}$ as shown below:

$$\sum_{j=1}^{m} \lambda_{m,j} = \sum_{j=1}^{m} \left(\frac{\gamma}{m} + (1-\gamma)\frac{\widetilde{\lambda}_j}{\sum_{k=1}^{m} \widetilde{\lambda}_k}\right) = \gamma + (1-\gamma)\sum_{j=1}^{m} \frac{\widetilde{\lambda}_j}{\sum_{k=1}^{m} \widetilde{\lambda}_k} = \gamma + (1-\gamma) = 1.$$

In Equation (S.5), the upper bound follows from Cauchy-Schwarz and Equation (S.4), and the lower bound from the definition of $\lambda_{m,j}$:

$$\sqrt{\gamma m} = \sum_{j=1}^{m} \sqrt{\frac{\gamma}{m}} \leq \sum_{j=1}^{m} \sqrt{\lambda_{m,j}} \leq \sqrt{\sum_{j=1}^{m} \lambda_{m,j}} \sqrt{\sum_{j=1}^{m} 1} = 1 \cdot \sqrt{m}.$$

For Equation (S.6), we note the following bounds on the sum of the $\lambda_{m,j}^r$ for all $r > 1$:

$$\sum_{j=1}^{m} \left(\lambda_{m,j}^{(2)}\right)^r \leq \sum_{j=1}^{m} (\lambda_{m,j})^r \leq \left(\left[\sum_{j=1}^{m} \left(\lambda_{m,j}^{(1)}\right)^r\right]^{1/r} + \left[\sum_{j=1}^{m} \left(\lambda_{m,j}^{(2)}\right)^r\right]^{1/r}\right)^r$$

where the second inequality uses the Minkowski inequality. But as $m \to \infty$, the term $\sum_{j=1}^{m}(\lambda_{m,j}^{(1)})^r = \gamma^r m^{-(r-1)} \to 0$. Furthermore, as $m \to \infty$,

$$\sum_{j=1}^{m} \left(\lambda_{m,j}^{(2)}\right)^r = \frac{(1-\gamma)^r}{\left(\sum_{k=1}^{m} \widetilde{\lambda}_k\right)^r} \sum_{j=1}^{m} \widetilde{\lambda}_j^r \to (1-\gamma)^r \sum_{j \geq 1} \widetilde{\lambda}_j^r$$

because $(\sum_{k \geq 1} \widetilde{\lambda}_k)^r = 1$.

Finally, we prove Equation (S.7). For all $\rho \in (0,1)$, we have

$$\sum_{j=\lfloor \rho m \rfloor + 1}^{m} \lambda_{m,j} = \frac{\gamma(m - \lfloor \rho m \rfloor)}{m} + (1-\gamma)\frac{\sum_{j=\lfloor \rho m \rfloor}^{m} \widetilde{\lambda}_j}{\sum_{j=1}^{m} \widetilde{\lambda}_j}.$$

By sandwiching, $\frac{\gamma(m - \lfloor \rho m \rfloor)}{m} \to \gamma(1-\rho)$. Additionally, the series $\sum_{j=1}^{m} \widetilde{\lambda}_j$ converges to 1. Thus, its tail converges to 0 and $\sum_{j=\lfloor \rho m \rfloor + 1}^{m} \widetilde{\lambda}_j \to 0$. $\qquad \square$

Figure 1 shows the value of $\sum_{j \geq 1} \widetilde{\lambda}_j^2 = \frac{\zeta(2/\alpha)}{\zeta(1/\alpha)^2}$ as a function of $\alpha$, when using Zipf weights Equation (4).

## C   Proof of Proposition 4.3 on the limiting NTG

This proposition holds also under the ReLU activation case. In what follows, we will give a proof that works for both the smooth activation function and ReLU.

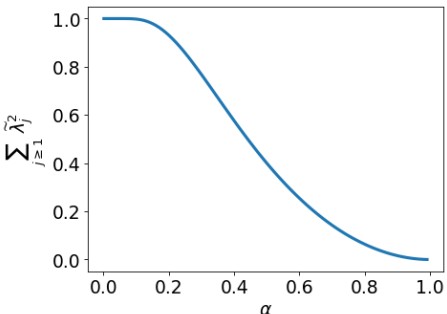

Figure 1: Value of $\sum_{j=1}^{\infty} \widetilde{\lambda}_j^2$ as a function of $\alpha$, where $(\widetilde{\lambda}_j)_{j\geq 1}$ are defined as in Equation (4), As $\alpha \to 1$, it converges to 0, which corresponds to the kernel regime.

It is sufficient to look at the convergence of individual entries of the NTG matrix; that is, to show that, for each pair $1 \leq i, i' \leq n$,

$$
\Theta_m(\mathbf{x}_i, \mathbf{x}_{i'}; \mathbf{W}_0) = \frac{\mathbf{x}_i^\top \mathbf{x}_{i'}}{d} \times \left( \frac{\gamma}{m} \sum_{j=1}^{m} \sigma'(Z_j(\mathbf{x}_i; \mathbf{W}_0))\sigma'(Z_j(\mathbf{x}_{i'}; \mathbf{W}_0)) \right.
$$
$$
\left. + \frac{(1-\gamma)}{\sum_{k=1}^{m} \widetilde{\lambda}_k} \sum_{j=1}^{m} \widetilde{\lambda}_j \sigma'(Z_j(\mathbf{x}_i; \mathbf{W}_0))\sigma'(Z_j(\mathbf{x}_{i'}; \mathbf{W}_0)) \right) \tag{S.8}
$$

tends to

$$
\gamma \Theta^*(\mathbf{x}_i, \mathbf{x}_{i'}) + \frac{(1-\gamma)}{d} \mathbf{x}_i^\top \mathbf{x}_{i'} \sum_{j=1}^{\infty} \widetilde{\lambda}_j \sigma'(Z_j(\mathbf{x}_i; \mathbf{W}_0))\sigma'(Z_j(\mathbf{x}_{i'}; \mathbf{W}_0)) \tag{S.9}
$$

almost surely as $m \to \infty$. Using the fact that $|\sigma'(z)| \leq 1$ and the triangle inequality, the modulus of the difference between the RHS of Equation (S.8) and Equation (S.9) is upper bounded by

$$
\left| \frac{\mathbf{x}_i^\top \mathbf{x}_{i'}}{d} \right| \left( \gamma \left| \left( \frac{1}{m} \sum_{j=1}^{m} \sigma'(Z_j(\mathbf{x}_i; \mathbf{W}_0))\sigma'(Z_j(\mathbf{x}_{i'}; \mathbf{W}_0)) \right) - \mathbb{E}[\sigma'(Z_1(\mathbf{x}_i; \mathbf{W}_0))\sigma'(Z_1(\mathbf{x}_{i'}; \mathbf{W}_0))] \right| \right.
$$
$$
\left. + (1-\gamma) \left[ \left( \frac{1}{\sum_{j=1}^{m} \widetilde{\lambda}_j} - 1 \right) \sum_{j=1}^{m} \widetilde{\lambda}_j + \sum_{j=m+1}^{\infty} \widetilde{\lambda}_j \right] \right)
$$
$$
= \left| \frac{\mathbf{x}_i^\top \mathbf{x}_{i'}}{d} \right| \left( \gamma \left| \left( \frac{1}{m} \sum_{j=1}^{m} \sigma'(Z_j(\mathbf{x}_i; \mathbf{W}_0))\sigma'(Z_j(\mathbf{x}_{i'}; \mathbf{W}_0)) \right) - \mathbb{E}[\sigma'(Z_1(\mathbf{x}_i; \mathbf{W}_0))\sigma'(Z_1(\mathbf{x}_{i'}; \mathbf{W}_0))] \right| \right.
$$
$$
\left. + 2(1-\gamma) \left[ 1 - \sum_{j=1}^{m} \widetilde{\lambda}_j \right] \right)
$$

which tends to 0 almost surely as $m$ tends to infinity using the law of large numbers and the fact that $\sum_{j=1}^{\infty} \widetilde{\lambda}_j = 1$.

## D Secondary Proposition - NTG at initialisation

The following proposition is a corollary of Lemma 4 in (Oymak and Soltanolkotabi, 2020). It holds under both the ReLU and smooth activation cases. A proof is included for completeness.

**Proposition D.1.** *Let $\delta \in (0,1)$. Assume Assumptions 3.1 and 3.3, $\gamma > 0$, and $m \geq \frac{2^3 n \log \frac{n}{\delta}}{\kappa_n d}$. Also, assume that the activation function satisfies Assumption 3.2 or it is ReLU. Then, with probability at least $1 - \delta$,*

$$\mathrm{eig}_{\min}(\widehat{\Theta}_m(\mathbf{X}; \mathbf{W}_0)) \geq \mathrm{eig}_{\min}(\widehat{\Theta}_m^{(1)}(\mathbf{X}; \mathbf{W}_0)) > \frac{\gamma \kappa_n}{2} > 0.$$

*Proof.* We follow here the proof of Lemma 4 in (Oymak and Soltanolkotabi, 2020).

$$\widehat{\Theta}_m(\mathbf{X}; \mathbf{W}) = \frac{1}{d} \sum_{j=1}^{m} \lambda_{m,j} A_j$$

$$= \frac{1}{d} \sum_{j=1}^{m} \lambda_{m,j}^{(1)} A_j + \frac{1}{d} \sum_{j=1}^{m} \lambda_{m,j}^{(2)} A_j$$

where

$$A_j = \mathrm{diag}(\boldsymbol{\sigma}'(\mathbf{X}\mathbf{w}_j / \sqrt{d}))\mathbf{X}\mathbf{X}^\top \mathrm{diag}(\boldsymbol{\sigma}'(\mathbf{X}\mathbf{w}_j / \sqrt{d})).$$

Let $\widehat{\Theta}_m^{(1)}(\mathbf{X}; \mathbf{W}) = \frac{1}{d} \sum_{j=1}^m \lambda_{m,j}^{(1)} A_j = \frac{\gamma}{md} \sum_{j=1}^m A_j$. Note that $\mathrm{eig}_{\min}(\widehat{\Theta}_m(\mathbf{X}; \mathbf{W})) \geq \mathrm{eig}_{\min}(\widehat{\Theta}_m^{(1)}(\mathbf{X}; \mathbf{W}))$ a.s., and

$$\mathbb{E}[\widehat{\Theta}_m^{(1)}(\mathbf{X}; \mathbf{W}_0)] = \gamma \widehat{\Theta}^*(\mathbf{X})$$

where $\widehat{\Theta}^*(\mathbf{X})$ is defined in Equation (10). We have, for all $j \geq 1$,

$$\|A_j\|_2 = \mathrm{eig}_{\max}(A_j) \leq \mathrm{eig}_{\max}(\mathrm{diag}(\boldsymbol{\sigma}'(\mathbf{X}\mathbf{w}_j / \sqrt{d}))^2) \, \mathrm{eig}_{\max}(\mathbf{X}\mathbf{X}^\top) \leq \mathrm{eig}_{\max}(\mathbf{X}\mathbf{X}^\top)$$

$$\leq \mathrm{trace}(\mathbf{X}\mathbf{X}^\top) \leq n. \tag{S.10}$$

At initialisation, $A_1, A_2, \ldots, A_m$ are independent random matrices. Using matrix Chernoff inequalities (see Proposition B.1), we obtain, for all $\epsilon \in [0,1)$,

$$\Pr\left(\mathrm{eig}_{\min}(\widehat{\Theta}_m(\mathbf{X}; \mathbf{W}_0)) \leq (1 - \epsilon)\gamma \kappa_n\right) \leq n e^{-\epsilon^2 m \kappa_n d / (2n)}.$$

Let $\delta \in (0,1)$. Taking $\epsilon = 1/2$, we have that, if $\frac{m \kappa_n d}{2^3 n} \geq \log \frac{n}{\delta}$, then

$$\Pr\left(\mathrm{eig}_{\min}(\widehat{\Theta}_m(\mathbf{X}; \mathbf{W}_0)) \leq \frac{\gamma \kappa_n}{2}\right) \leq \delta.$$

$\square$

# E Secondary Lemmas on gradient flow dynamics

The proof technique used to prove Theorems 5.1 and A.1 is similar to that of (Du et al., 2019b) (NTK scaling). In particular, we provide in this section Lemmas similar to Lemmas 3.2, 3.3 and 3.4 in (Du et al., 2019b), but adapted to our setting. Lemma E.1 is an adaptation of Lemma 3.3. Lemmas E.2 and E.4 are adaptations of Lemma 3.2, respectively for the ReLU and smooth activation cases. Lemmas E.3 and E.5 are adaptations of Lemma 3.4, respectively for the ReLU and smooth activation cases.

## E.1 Lemma on exponential decay of the empirical risk and scaling of the weight changes

The following lemma is an adaptation of Lemma 3.3 of (Du et al., 2019b), and applies to both the ReLU and smooth activation cases. It shows that, if the minimum eigenvalue of the NTG matrix is bounded away from 0, gradient flow converges to a global minimum exponentially fast. Recall that $\mathbf{y} = (y_1, \ldots, y_n)^\top \in \mathbb{R}^n$.

**Lemma E.1.** *Let $t > 0$ and $\zeta > 0$. Assume Assumption 3.1 and $\mathrm{eig}_{\min}(\widehat{\Theta}_m(\mathbf{X}; \mathbf{W}_s)) \geq \frac{\zeta}{2}$ for all $0 \leq s \leq t$. Also, assume that the activation function satisfies Assumption 3.2 or it is ReLU. Then,*

$$L_m(\mathbf{W}_t) \leq e^{-\zeta t} L_m(\mathbf{W}_0),$$

*and for all $j \in [m]$,*

$$\|\mathbf{w}_{tj} - \mathbf{w}_{0j}\| \leq \sqrt{\frac{n\lambda_{m,j}}{d}} \|\mathbf{y} - \mathbf{u}_0\| \frac{2}{\zeta}, \tag{S.11}$$

*where* $\mathbf{u}_0 = (f_m(\mathbf{x}_1; \mathbf{W}_0), \ldots, f_m(\mathbf{x}_n; \mathbf{W}_0))^\top \in \mathbb{R}^n$.

*Proof.* For $0 \leq s \leq t$, write $\mathbf{u}_s = (f_m(\mathbf{x}_1; \mathbf{W}_s), \ldots, f_m(\mathbf{x}_n; \mathbf{W}_s))^\top$. We have

$$\frac{d}{ds}\mathbf{u}_s = \widehat{\Theta}_m(\mathbf{X}; \mathbf{W}_s)(\mathbf{y} - \mathbf{u}_s).$$

It follows that

$$\frac{dL_m(\mathbf{W}_s)}{ds} = -(\mathbf{y} - \mathbf{u}_s)^\top \widehat{\Theta}_m(\mathbf{X}; \mathbf{W}_s)(\mathbf{y} - \mathbf{u}_s) \leq -\frac{\zeta}{2}(\mathbf{y} - \mathbf{u}_s)^\top(\mathbf{y} - \mathbf{u}_s) = -\zeta L_m(\mathbf{W}_s).$$

Using Grönwall's inequality, we obtain

$$L_m(\mathbf{W}_t) \leq e^{-\zeta t} L_m(\mathbf{W}_0).$$

For $0 \leq s \leq t$, using the Cauchy-Schwarz inequality, we get

$$\begin{aligned}
\left\|\frac{d\mathbf{w}_{sj}}{ds}\right\|^2 &= \left\|\sqrt{\lambda_{m,j}}\frac{a_j}{\sqrt{d}} \sum_{i=1}^n \sigma'(Z_{sj}(\mathbf{x}_i))\mathbf{x}_i \cdot (y_i - f_m(\mathbf{x}_i; \mathbf{W}_s))\right\|^2 \\
&= \frac{\lambda_{m,j}}{d} \sum_{k=1}^d \left(\sum_{i=1}^n \sigma'(Z_{sj}(\mathbf{x}_i))x_{ik} \cdot (y_i - f_m(\mathbf{x}_i; \mathbf{W}_s))\right)^2 \\
&\leq \frac{\lambda_{m,j}}{d} \sum_{k=1}^d \left(\sum_{i=1}^n x_{ik}^2\right)\left(\sum_{i=1}^n \sigma'(Z_{sj}(\mathbf{x}_i))^2(y_i - f_m(\mathbf{x}_i; \mathbf{W}_s))^2\right) \\
&= \frac{\lambda_{m,j}}{d} \left(\sum_{i=1}^n \sigma'(Z_{sj}(\mathbf{x}_i))^2(y_i - f_m(\mathbf{x}_i; \mathbf{W}_s))^2\right)\left(\sum_{k=1}^d \sum_{i=1}^n x_{ik}^2\right) \\
&\leq \frac{\lambda_{m,j}}{d} \left(\sum_{i=1}^n (y_i - f_m(\mathbf{x}_i; \mathbf{W}_s))^2\right)\left(\sum_{i=1}^n \sum_{k=1}^d x_{ik}^2\right) \\
&\leq \frac{n\lambda_{m,j}}{d} \|\mathbf{y} - \mathbf{u}_s\|^2 \\
&\leq \frac{n\lambda_{m,j}}{d} \|\mathbf{y} - \mathbf{u}_0\|^2 e^{-\zeta s}.
\end{aligned}$$

Integrating and using Minkowski's integral inequality, we obtain

$$\begin{aligned}
\|\mathbf{w}_{tj} - \mathbf{w}_{0j}\| = \left\|\int_0^t \frac{d}{ds}\mathbf{w}_{sj}ds\right\| &\leq \int_0^t \left\|\frac{d}{ds}\mathbf{w}_{sj}\right\| ds \\
&\leq \sqrt{\frac{n\lambda_{m,j}}{d}} \|\mathbf{y} - \mathbf{u}_0\| \int_0^t e^{-\zeta s/2}ds \\
&\leq \sqrt{\frac{n\lambda_{m,j}}{d}} \|\mathbf{y} - \mathbf{u}_0\| \frac{2}{\zeta}.
\end{aligned}$$

$\square$

From now on, the proofs for the ReLU and smooth-activation cases slightly differ.

### E.2 Lemma bounding the NTK change and minimum eigenvalue - ReLU case

The next lemma and its proof are similar to Lemma 3.2 in (Du et al., 2019b) and its proof. Recall that $0 < \|\mathbf{x}_i\| \leq 1$ for every $i \in [n]$, and the $\mathbf{w}_{0j}$ are iid $\mathcal{N}(0, \mathbf{I}_d)$.

**Lemma E.2.** *Let $\delta \in (0,1)$, and $c_{m,j} > 0$ for every $j \in [m]$. Assume that Assumptions 3.1 and 3.3 holds and the activation function is ReLU. Then, with probability at least $1 - \delta$, the following holds. For every $\mathbf{W} = (\mathbf{w}_1^\top, \ldots, \mathbf{w}_m^\top)^\top$, if it satisfies*

$$\|\mathbf{w}_{0j} - \mathbf{w}_j\| \leq \frac{\delta^2 c_{m,j}}{4} \quad \text{for all } j \in [m],$$

*we have*

$$\left\|\widehat{\Theta}_m^{(s)}(\mathbf{X}; \mathbf{W}) - \widehat{\Theta}_m^{(s)}(\mathbf{X}; \mathbf{W}_0)\right\|_2 \leq \frac{n}{d} \sum_{j=1}^m \lambda_{m,j}^{(k)} c_{m,j} + \frac{2n}{d} \sqrt{\sum_{j=1}^m \lambda_{m,j}^{(k)} c_{m,j}} \quad \text{for all } k \in [2]$$

*and*

$$\text{eig}_{\min}(\widehat{\Theta}_m(\mathbf{X}; \mathbf{W})) \geq \text{eig}_{\min}(\widehat{\Theta}_m^{(1)}(\mathbf{X}; \mathbf{W}_0)) - \left( \frac{n\gamma}{dm} \sum_{j=1}^m c_{m,j} + \frac{2n\gamma}{dm^{1/2}} \sqrt{\sum_{j=1}^m c_{m,j}} \right). \tag{S.12}$$

*Proof.* For $k \in [2]$, let

$$f_m^{(k)}(-; \mathbf{W}) : \mathbb{R}^d \to \mathbb{R}, \qquad f_m^{(k)}(\mathbf{x}; \mathbf{W}) = \sum_{j=1}^m \sqrt{\lambda_{m,j}^{(k)}} a_j \sigma(Z_j(\mathbf{x}; \mathbf{W})).$$

Define $\nabla_{\mathbf{W}} f_m^{(k)}(\mathbf{X}; \mathbf{W})$ to be the $n$-by-$(md)$ matrix whose $i$-th row is the $md$-dimensional row vector $(\nabla_{\mathbf{W}} f_m^{(k)}(\mathbf{x}_i; \mathbf{W}))^\top$.

Note that for all $k \in [2]$,

$$\left\|\widehat{\Theta}_m^{(k)}(\mathbf{X}; \mathbf{W}) - \widehat{\Theta}_m^{(k)}(\mathbf{X}; \mathbf{W}_0)\right\|_2$$
$$= \left\|\nabla_{\mathbf{W}} f_m^{(k)}(\mathbf{X}; \mathbf{W}) \nabla_{\mathbf{W}} f_m^{(k)}(\mathbf{X}; \mathbf{W})^\top - \nabla_{\mathbf{W}} f_m^{(k)}(\mathbf{X}; \mathbf{W}_0) \nabla_{\mathbf{W}} f_m^{(k)}(\mathbf{X}; \mathbf{W}_0)^\top\right\|_2$$
$$\leq \left\|\nabla_{\mathbf{W}} f_m^{(k)}(\mathbf{X}; \mathbf{W}) - \nabla_{\mathbf{W}} f_m^{(k)}(\mathbf{X}; \mathbf{W}_0)\right\|_2^2 \tag{S.13}$$
$$+ 2\left\|\nabla_{\mathbf{W}} f_m^{(k)}(\mathbf{X}; \mathbf{W}_0)\right\|_2 \left\|\nabla_{\mathbf{W}} f_m^{(k)}(\mathbf{X}; \mathbf{W}) - \nabla_{\mathbf{W}} f_m^{(k)}(\mathbf{X}; \mathbf{W}_0)\right\|_2.$$

The justification of the inequality from above is given below (which is an expanded version of the three equations (364-366) in (Bartlett et al., 2021)): for all $n$-by-$(pd)$ matrices $A$ and $B$,

$$\left\|AA^\top - BB^\top\right\|_2 = \left\|\frac{1}{2}(A - B)(A + B)^\top + \frac{1}{2}(A + B)(A - B)^\top\right\|_2$$
$$\leq \frac{1}{2}\left(\left\|(A - B)(A + B)^\top\right\|_2 + \left\|(A + B)(A - B)^\top\right\|_2\right)$$
$$\leq \frac{1}{2}\left(\|A - B\|_2 \times \left\|(A + B)^\top\right\|_2 + \|A + B\|_2 \times \left\|(A - B)^\top\right\|_2\right)$$
$$= \|A - B\|_2 \times \|A + B\|_2$$
$$\leq \|A - B\|_2 \times (\|A - B + B\|_2 + \|B\|_2)$$
$$\leq \|A - B\|_2 \times (\|A - B\|_2 + 2\|B\|_2).$$

Coming back to the inequality in Equation (S.13), we next bound the two terms $\left\|\nabla_{\mathbf{W}} f_m^{(k)}(\mathbf{X}; \mathbf{W}_0)\right\|_2$ and $\left\|\nabla_{\mathbf{W}} f_m^{(k)}(\mathbf{X}; \mathbf{W}) - \nabla_{\mathbf{W}} f_m^{(k)}(\mathbf{X}; \mathbf{W}_0)\right\|_2$ there.

We bound the first term as follows:

$$\left\|\nabla_{\mathbf{W}} f_m^{(k)}(\mathbf{X}; \mathbf{W}_0)\right\|_2^2 \leq \left\|\nabla_{\mathbf{W}} f_m^{(k)}(\mathbf{X}; \mathbf{W}_0)\right\|_F^2 = \sum_{i=1}^n \sum_{j=1}^m \left\|\nabla_{\mathbf{w}_j} f_m^{(k)}(\mathbf{x}_i; \mathbf{W}_0)\right\|^2$$

$$= \sum_{i=1}^n \sum_{j=1}^m \lambda_{m,j}^{(k)} \left|\sigma'(Z_j(\mathbf{x}_i; \mathbf{W}_0))\right|^2 \frac{\|\mathbf{x}_i\|^2}{d}$$

$$\leq \frac{n}{d} \sum_{j=1}^m \lambda_{m,j}^{(k)} \leq \frac{n}{d} \gamma_k \tag{S.14}$$

where $\gamma_1 = \gamma$ and $\gamma_2 = 1 - \gamma$. The second inequality uses the assumption that $|\sigma'(x)| \leq 1$ for all $x \in \mathbb{R}$ and $\|\mathbf{x}_i\| \leq 1$ for all $i \in [n]$. The third inequality follows from the fact that $\sum_{j=1}^m \lambda_{m,j}^{(k)} \leq \sum_{j=1}^m \lambda_{m,j} = 1$.

For the second term, we recall that $Z_j(\mathbf{x}_i; \mathbf{W}) = \frac{1}{\sqrt{d}} \mathbf{w}_j^\top \mathbf{x}_i$. Using this fact, we derive an upper bound for the second term as follows:

$$\left\|\nabla_{\mathbf{W}} f_m^{(k)}(\mathbf{X}; \mathbf{W}) - \nabla_{\mathbf{W}} f_m^{(k)}(\mathbf{X}; \mathbf{W}_0)\right\|_2^2$$

$$\leq \left\|\nabla_{\mathbf{W}} f_m^{(k)}(\mathbf{X}; \mathbf{W}) - \nabla_{\mathbf{W}} f_m^{(k)}(\mathbf{X}; \mathbf{W}_0)\right\|_F^2$$

$$= \sum_{i=1}^n \sum_{j=1}^m \left\|\nabla_{\mathbf{w}_j} f_m^{(k)}(\mathbf{x}_i; \mathbf{W}) - \nabla_{\mathbf{w}_j} f_m^{(k)}(\mathbf{x}_i; \mathbf{W}_0)\right\|^2$$

$$= \sum_{i=1}^n \sum_{j=1}^m \left\|\sqrt{\lambda_{m,j}^{(k)}} a_j \frac{\mathbf{x}_i}{\sqrt{d}} \left[\sigma'(Z_j(\mathbf{x}_i; \mathbf{W})) - \sigma'(Z_j(\mathbf{x}_i; \mathbf{W}_0))\right]\right\|^2$$

$$= \frac{1}{d} \sum_{i=1}^n \sum_{j=1}^m \|\mathbf{x}_i\|^2 \lambda_{m,j}^{(k)} \left|\sigma'(Z_j(\mathbf{x}_i; \mathbf{W})) - \sigma'(Z_j(\mathbf{x}_i; \mathbf{W}_0))\right|^2. \tag{S.15}$$

In the rest of the proof, we will derive a probabilistic bound on the upper bound just obtained, and show the conclusions claimed in the lemma.

For any $\epsilon > 0$, $i \in [n]$, and $j \in [m]$, we define the event

$$A_{i,j}(\epsilon) = \left\{\exists \mathbf{w}_j \text{ s.t. } \|\mathbf{w}_{0j} - \mathbf{w}_j\| \leq \epsilon \text{ and } \sigma'(\mathbf{w}_j^\top \mathbf{x}_i) \neq \sigma'(\mathbf{w}_{0j}^\top \mathbf{x}_i)\right\}.$$

If this event happens, we have $|\mathbf{w}_{0j}^\top \mathbf{x}_i| \leq \epsilon$. To see this, assume that $A_{i,j}(\epsilon)$ holds with $\mathbf{w}_j$ as a witness of the existential quantification, and note that since the norm of $\mathbf{x}_i$ is at most 1,

$$\left|\mathbf{w}_{0j}^\top \mathbf{x}_i - \mathbf{w}_j^\top \mathbf{x}_i\right| \leq \|\mathbf{w}_{0j} - \mathbf{w}_j\| \|\mathbf{x}_i\| \leq \epsilon.$$

If $\mathbf{w}_{0j}^\top \mathbf{x}_i > 0$, then $\mathbf{w}_j^\top \mathbf{x}_i \leq 0$ and thus

$$\mathbf{w}_{0j}^\top \mathbf{x}_i \leq \epsilon + \mathbf{w}_j^\top \mathbf{x}_i < \epsilon.$$

Alternatively, if $\mathbf{w}_{0j}^\top \mathbf{x}_i \leq 0$, then $\mathbf{w}_j^\top \mathbf{x}_i > 0$ and thus

$$-\mathbf{w}_{0j}^\top \mathbf{x}_i \leq \epsilon - \mathbf{w}_j^\top \mathbf{x}_i \leq \epsilon.$$

In both cases, we have the desired $|\mathbf{w}_{0j}^\top \mathbf{x}_i| \leq \epsilon$.

Using the observation that we have just explained and the fact that $\mathbf{w}_{0j}^\top \mathbf{x}_i \sim \mathcal{N}(0, \|\mathbf{x}_i\|^2)$, we obtain, for a random variable $N \sim \mathcal{N}(0, 1)$,

$$
\begin{aligned}
\Pr(A_{i,j}(\epsilon)) \leq \Pr\left(|N| \leq \frac{\epsilon}{\|\mathbf{x}_i\|}\right) &= \operatorname{erf}\left(\frac{\epsilon}{\|\mathbf{x}_i\|\sqrt{2}}\right) \\
&\leq \sqrt{1 - \exp\left(-\left(4\left(\frac{\epsilon}{\|\mathbf{x}_i\|\sqrt{2}}\right)^2\right)/\pi\right)} \\
&\leq \sqrt{\frac{2\epsilon^2}{\|\mathbf{x}_i\|^2\pi}} \leq \frac{\epsilon}{\|\mathbf{x}_i\|},
\end{aligned}
\tag{S.16}
$$

where the second inequality uses $\operatorname{erf}(x) \leq \sqrt{1 - \exp(-(4x^2)/\pi)}$. Let $\Psi(\mathbf{W}_0)$ be the constraint on $\mathbf{W} = (\mathbf{w}_1^\top, \ldots, \mathbf{w}_m^\top)^\top$ defined by

$$
\mathbf{W} \in \Psi(\mathbf{W}_0) \iff \|\mathbf{w}_{0j'} - \mathbf{w}_{j'}\| \leq \frac{\delta^2 c_{m,j'}}{4} \text{ for all } j' \in [m].
$$

Then, for all $k = 1, 2$, we have

$$
\begin{aligned}
\mathbb{E}&\left[\sup_{\mathbf{W} \in \Psi(\mathbf{W}_0)} \left\|\nabla_{\mathbf{W}} f_m^{(k)}(\mathbf{X}; \mathbf{W}) - \nabla_{\mathbf{W}} f_m^{(k)}(\mathbf{X}; \mathbf{W}_0)\right\|_2^2\right] \\
&\leq \frac{1}{d}\sum_{i=1}^n \sum_{j=1}^m \|\mathbf{x}_i\|^2 \lambda_{m,j}^{(k)} \mathbb{E}\left[\sup_{\mathbf{W} \in \Psi(\mathbf{W}_0)} |\sigma'(Z_j(\mathbf{x}_i; \mathbf{W})) - \sigma'(Z_j(\mathbf{x}_i; \mathbf{W}_0))|^2\right] \\
&\leq \frac{1}{d}\sum_{i=1}^n \sum_{j=1}^m \|\mathbf{x}_i\|^2 \lambda_{m,j}^{(k)} \Pr\left(\exists \mathbf{W} \in \Psi(\mathbf{W}_0) \text{ s.t. } \sigma'(Z_j(\mathbf{x}_i; \mathbf{W})) \neq \sigma'(Z_j(\mathbf{x}_i; \mathbf{W}_0))\right) \\
&= \frac{1}{d}\sum_{i=1}^n \sum_{j=1}^m \|\mathbf{x}_i\|^2 \lambda_{m,j}^{(k)} \Pr\left(\exists \mathbf{w}_j \text{ s.t. } \|\mathbf{w}_{0j} - \mathbf{w}_j\| \leq \frac{\delta^2 c_{m,j}}{4} \text{ and } \sigma'(\mathbf{w}_j^\top \mathbf{x}_i) \neq \sigma'(\mathbf{w}_{0j}^\top \mathbf{x}_i)\right) \\
&\leq \frac{1}{d}\sum_{i=1}^n \sum_{j=1}^m \|\mathbf{x}_i\|^2 \lambda_{m,j}^{(k)} \Pr\left(A_{i,j}(\delta^2 c_{m,j}/4)\right) \\
&\leq \frac{(\delta^2/4)}{d}\sum_{i=1}^n \sum_{j=1}^m \|\mathbf{x}_i\| \lambda_{m,j}^{(k)} c_{m,j} \\
&\leq \frac{n(\delta^2/4)}{d}\sum_{j=1}^m \lambda_{m,j}^{(k)} c_{m,j}.
\end{aligned}
$$

The first inequality uses the bound in Equation (S.15), and the fourth inequality uses the inequality derived in Equation (S.16).

We bring together the bound on the expectation just shown and also the bounds proved in Equations (S.13) and (S.14). Recalling that $\gamma_1 = \gamma$ and $\gamma_2 = 1 - \gamma$, we have

$$
\mathbb{E}\left[\sup_{\mathbf{W} \in \Psi(\mathbf{W}_0)} \left\| \widehat{\Theta}_m^{(k)}(\mathbf{X}; \mathbf{W}) - \widehat{\Theta}_m^{(k)}(\mathbf{X}; \mathbf{W}_0) \right\|_2 \right]
$$

$$
\leq \mathbb{E}\left[\sup_{\mathbf{W} \in \Psi(\mathbf{W}_0)} \left\| \nabla_{\mathbf{W}} f_m^{(k)}(\mathbf{X}; \mathbf{W}) - \nabla_{\mathbf{W}} f_m^{(k)}(\mathbf{X}; \mathbf{W}_0) \right\|_2^2 \right]
$$

$$
+ 2\,\mathbb{E}\left[\sup_{\mathbf{W} \in \Psi(\mathbf{W}_0)} \left\| \nabla_{\mathbf{W}} f_m^{(k)}(\mathbf{X}; \mathbf{W}_0) \right\|_2 \left\| \nabla_{\mathbf{W}} f_m^{(k)}(\mathbf{X}; \mathbf{W}) - \nabla_{\mathbf{W}} f_m^{(k)}(\mathbf{X}; \mathbf{W}_0) \right\|_2 \right]
$$

$$
\leq \mathbb{E}\left[\sup_{\mathbf{W} \in \Psi(\mathbf{W}_0)} \left\| \nabla_{\mathbf{W}} f_m^{(k)}(\mathbf{X}; \mathbf{W}) - \nabla_{\mathbf{W}} f_m^{(k)}(\mathbf{X}; \mathbf{W}_0) \right\|_2^2 \right]
$$

$$
+ 2\sqrt{\frac{n}{d}\gamma_k}\,\mathbb{E}\left[\sup_{\mathbf{W} \in \Psi(\mathbf{W}_0)} \left\| \nabla_{\mathbf{W}} f_m^{(k)}(\mathbf{X}; \mathbf{W}) - \nabla_{\mathbf{W}} f_m^{(k)}(\mathbf{X}; \mathbf{W}_0) \right\|_2 \right]
$$

$$
\leq \mathbb{E}\left[\sup_{\mathbf{W} \in \Psi(\mathbf{W}_0)} \left\| \nabla_{\mathbf{W}} f_m^{(k)}(\mathbf{X}; \mathbf{W}) - \nabla_{\mathbf{W}} f_m^{(k)}(\mathbf{X}; \mathbf{W}_0) \right\|_2^2 \right]
$$

$$
+ 2\sqrt{\frac{n}{d}\gamma_k}\,\sqrt{\mathbb{E}\left[\sup_{\mathbf{W} \in \Psi(\mathbf{W}_0)} \left\| \nabla_{\mathbf{W}} f_m^{(k)}(\mathbf{X}; \mathbf{W}) - \nabla_{\mathbf{W}} f_m^{(k)}(\mathbf{X}; \mathbf{W}_0) \right\|_2^2 \right]}
$$

$$
\leq \frac{n(\delta^2/4)}{d} \sum_{j=1}^{m} \lambda_{m,j}^{(k)} c_{m,j} + 2\sqrt{\frac{n}{d}\gamma_k} \sqrt{\frac{n(\delta^2/4)}{d} \sum_{j=1}^{m} \lambda_{m,j}^{(k)} c_{m,j}}
$$

$$
\leq \frac{\delta}{2}\left( \frac{n}{d}\sum_{j=1}^{m} \lambda_{m,j}^{(k)} c_{m,j} + \frac{2n}{d}\sqrt{\gamma_k \sum_{j=1}^{m} \lambda_{m,j}^{(k)} c_{m,j}} \right).
$$

The third inequality uses Jensen's inequality, and the last uses the fact that $\delta/2 \geq (\delta/2)^2$. Hence, for each $k = 1, 2$, by Markov inequality, we have, with probability at least $1 - (\delta/2)$,

$$
\sup_{\mathbf{W} \in \Psi(\mathbf{W}_0)} \left\| \widehat{\Theta}_m^{(k)}(\mathbf{X}; \mathbf{W}) - \widehat{\Theta}_m^{(k)}(\mathbf{X}; \mathbf{W}_0) \right\|_2 \leq \frac{n}{d}\sum_{j=1}^{m} \lambda_{m,j}^{(k)} c_{m,j} + \frac{2n}{d}\sqrt{\gamma_k}\sqrt{\sum_{j=1}^{m} \lambda_{m,j}^{(k)} c_{m,j}}.
$$

By union bound, the conjunction of the above inequalities for the $k = 1$ and $k = 2$ cases holds with probability at least $1 - \delta$.

We prove the last remaining claim using the following lemma.

If $A$ and $B$ are real symmetric matrices, then

$$
\text{eig}_{\min}(A) \geq \text{eig}_{\min}(B) - \|A - B\|_2,
$$

which holds because

$$
\text{eig}_{\min}(A) = \text{eig}_{\min}(B + (A - B)) \geq \text{eig}_{\min}(B) + \text{eig}_{\min}(A - B)
$$
$$
\geq \text{eig}_{\min}(B) - \text{eig}_{\max}(B - A)
$$
$$
\geq \text{eig}_{\min}(B) - \|B - A\|_2 = \text{eig}_{\min}(B) - \|A - B\|_2.
$$

Thus,

$$
\inf_{\mathbf{W} \in \Psi(\mathbf{W}_0)} \left( \mathrm{eig}_{\min}(\widehat{\Theta}_m^{(1)}(\mathbf{X}; \mathbf{W})) \right)
$$

$$
\geq \mathrm{eig}_{\min}(\widehat{\Theta}_m^{(1)}(\mathbf{X}; \mathbf{W}_0)) - \sup_{\mathbf{W} \in \Psi(\mathbf{W}_0)} \left\| \widehat{\Theta}_m^{(1)}(\mathbf{X}; \mathbf{W}) - \widehat{\Theta}_m^{(1)}(\mathbf{X}; \mathbf{W}_0) \right\|_2
$$

$$
\geq \mathrm{eig}_{\min}(\widehat{\Theta}_m^{(1)}(\mathbf{X}; \mathbf{W}_0)) - \left( \frac{n}{d} \sum_{j=1}^{m} \lambda_{m,j}^{(1)} c_{m,j} + \frac{2n}{d} \sqrt{\gamma \sum_{j=1}^{m} \lambda_{m,j}^{(1)} c_{m,j}} \right)
$$

$$
= \mathrm{eig}_{\min}(\widehat{\Theta}_m^{(1)}(\mathbf{X}; \mathbf{W}_0)) - \left( \frac{n\gamma}{dm} \sum_{j=1}^{m} c_{m,j} + \frac{2n\gamma}{dm^{1/2}} \sqrt{\sum_{j=1}^{m} c_{m,j}} \right)
$$

holds with probability at least $1 - \delta$. Equation (S.12) then follows from the fact that for all $\mathbf{W}$, $\mathrm{eig}_{\min}(\widehat{\Theta}_m(\mathbf{X}; \mathbf{W})) \geq \mathrm{eig}_{\min}(\widehat{\Theta}_m^{(1)}(\mathbf{X}; \mathbf{W}))$. □

### E.3 Lemma on a sufficient condition for Theorem A.1 - ReLU case

We now bring together the results from Proposition D.1 and Lemmas E.1 and E.2, and identify a sufficient condition for Theorem A.1, which corresponds to the condition in Lemma 3.4 in (Du et al., 2019b).

**Lemma E.3.** *Consider $\delta \in (0, 1)$. Assume that Assumptions 3.1 and 3.3 hold, the activation function is ReLU, and $c_{m,j} > 0$ for all $j \in [m]$. Also, assume that $\gamma > 0$ and*

$$
m \geq \max \left( \left( \frac{8n \log \frac{4n}{\delta}}{d\kappa_n} \right), \left( \frac{8n}{d\kappa_n} \sum_{j=1}^{m} c_{m,j} \right), \left( \frac{16^2 n^2}{d^2 \kappa_n^2} \sum_{j=1}^{m} c_{m,j} \right) \right).
$$

*Define*

$$
R'_{m,j} = \sqrt{\frac{n\lambda_{m,j}}{d}} \, \|\mathbf{y} - \mathbf{u}_0\| \, \frac{4}{\gamma \kappa_n} \quad and \quad R_{m,j} = \frac{\delta^2 c_{m,j}}{64}.
$$

*If $R'_{m,j} < R_{m,j}$ for all $j \in [m]$ with probability at least $1 - \frac{\delta}{2}$, then on an event with probability at least $1 - \delta$, we have that for all $j \in [m]$, $R'_{m,j} < R_{m,j}$ and the following properties also hold for all $t \geq 0$:*

*(a) $\mathrm{eig}_{\min}(\widehat{\Theta}_m(\mathbf{X}; \mathbf{W}_t)) \geq \frac{\gamma \kappa_n}{4}$;*

*(b) $L_m(\mathbf{W}_t) \leq e^{-(\gamma \kappa_n t)/2} L_m(\mathbf{W}_0)$;*

*(c) $\|\mathbf{w}_{tj} - \mathbf{w}_{0j}\| \leq R'_{m,j}$ for all $j \in [m]$; and*

*(d) $\|\widehat{\Theta}_m(\mathbf{X}; \mathbf{W}_t) - \widehat{\Theta}_m(\mathbf{X}; \mathbf{W}_0)\|_2 \leq \frac{n}{d} \sum_{j=1}^{m} \lambda_{m,j} c_{m,j} + \frac{2\sqrt{2} \cdot n}{d} \sqrt{\sum_{j=1}^{m} \lambda_{m,j} c_{m,j}}$.*

*Proof.* Suppose $R'_{m,j} < R_{m,j}$ for all $j \in [m]$ on some event $A'$ having probability at least $1 - \frac{\delta}{2}$. Also, we would like to instantiate Proposition D.1 and Lemma E.2 with $\delta/4$, so that each of their claims holds with probability at least $1 - \frac{\delta}{4}$. Let $A$ be the intersection of $A'$ with the event that the conjunction of the two claims in Proposition D.1 and Lemma E.2 hold with $\delta/4$. By the union bound, $A$ has probability at least $1 - \delta$. We will show that on the event $A$, the four claimed properties of the lemma hold.

It will be sufficient to show that

$$
\|\mathbf{w}_{sj} - \mathbf{w}_{0j}\| \leq R_{m,j} \quad \text{for all } j \in [m] \text{ and } s \geq 0. \tag{S.17}
$$

To see why doing so is sufficient, pick an arbitrary $t_0 \geq 0$, and assume the above inequality for all $s \geq 0$. Then, by event $A$ and Lemma E.2, for all $0 \leq s \leq t_0$, we have the following upper bound on the change of

the Gram matrix from time 0 to $s$, and the following lower bound on the smallest eigenvalue of $\widehat{\Theta}_m(\mathbf{X}; \mathbf{W}_s)$:

$$
\left\|\widehat{\Theta}_m(\mathbf{X}; \mathbf{W}_s) - \widehat{\Theta}_m(\mathbf{X}; \mathbf{W}_0)\right\|_2 \leq \sum_{k=1}^{2} \left\|\widehat{\Theta}_m^{(k)}(\mathbf{X}; \mathbf{W}_s) - \widehat{\Theta}_m^{(k)}(\mathbf{X}; \mathbf{W}_0)\right\|_2
$$
$$
\leq \sum_{k=1}^{2} \left( \frac{n}{d} \sum_{j=1}^{m} \lambda_{m,j}^{(k)} c_{m,j} + \frac{2n}{d} \sqrt{\sum_{j=1}^{m} \lambda_{m,j}^{(k)} c_{m,j}} \right)
$$
$$
\leq \frac{n}{d} \sum_{j=1}^{m} \lambda_{m,j} c_{m,j} + \frac{2\sqrt{2} \cdot n}{d} \sqrt{\sum_{j=1}^{m} \lambda_{m,j} c_{m,j}}
$$

and

$$
\mathrm{eig}_{\min}(\widehat{\Theta}_m(\mathbf{X}; \mathbf{W}_s)) \geq \mathrm{eig}_{\min}(\widehat{\Theta}_m^{(1)}(\mathbf{X}; \mathbf{W}_0)) - \left( \frac{n\gamma}{dm} \sum_{j=1}^{m} c_{m,j} + \frac{2n\gamma}{dm^{1/2}} \sqrt{\sum_{j=1}^{m} c_{m,j}} \right)
$$
$$
\geq \frac{\gamma\kappa_n}{2} - \frac{\gamma\kappa_n}{4} \cdot \left( \frac{1}{m} \cdot \frac{4n}{d\kappa_n} \sum_{j=1}^{m} c_{m,j} + \frac{1}{m^{1/2}} \cdot \frac{8n}{d\kappa_n} \sqrt{\sum_{j=1}^{m} c_{m,j}} \right)
$$
$$
\geq \frac{\gamma\kappa_n}{2} - \frac{\gamma\kappa_n}{4} = \frac{\gamma\kappa_n}{4}.
$$

We now apply Lemma E.1 with $\zeta$ being set to $\frac{\gamma\kappa_n}{2}$, which gives

$$
L_m(\mathbf{W}_{t_0}) \leq e^{-(\gamma\kappa_n t_0)/2} L_m(\mathbf{W}_0)
$$

and

$$
\|\mathbf{w}_{t_0 j} - \mathbf{w}_{0j}\| \leq \sqrt{\frac{n\lambda_{m,j}}{d}} \|\mathbf{y} - \mathbf{u}_0\| \frac{4}{\gamma\kappa_n} = R'_{m,j} \quad \text{for all } j \in [m].
$$

We have just shown that all the four properties in the lemma hold for $t_0$.

It remains to prove Equation (S.17) under the event $A$ and the assumption that $R'_{m,j} < R_{m,j}$ for all $j \in [m]$ holds on this event. Suppose that Equation (S.17) fails for some $j \in [m]$. Let

$$
t_1 = \inf \left\{ t \mid \|\mathbf{w}_j - \mathbf{w}_{0j}\| > R_{m,j} \text{ for some } j \in [m] \right\}.
$$

Then, by the continuity of $\mathbf{w}_{tj}$ on $t$, we have

$$
\|\mathbf{w}_{sj} - \mathbf{w}_{0j}\| \leq R_{m,j} \quad \text{for all } j \in [m] \text{ and } 0 \leq s \leq t_1
$$

and for some $j_0 \in [m]$,

$$
\|\mathbf{w}_{t_1 j_0} - \mathbf{w}_{0j_0}\| = R_{m,j_0}. \tag{S.18}
$$

Thus, by the argument that we gave in the previous paragraph, we have

$$
\|\mathbf{w}_{t_1 j} - \mathbf{w}_{0j}\| \leq R'_{m,j} \quad \text{for all } j \in [m].
$$

In particular, $\|\mathbf{w}_{t_1 j_0} - \mathbf{w}_{0j_0}\| \leq R'_{m,j_0}$. But this contradicts our assumption $R'_{m,j_0} < R_{m,j_0}$. $\qquad \square$

### E.4 Lemma bounding the NTK change and minimum eigenvalue - Smooth activation case

We now give a version of Lemma E.2 for the smooth activation case (that is, under Assumption 3.2). The proof of this version is similar to the one for Lemma 5 in (Oymak and Soltanolkotabi, 2020), and uses the three equations (364-366) in (Bartlett et al., 2021).

**Lemma E.4.** *Assume that Assumptions 3.1 to 3.3 hold. Let $c_{m,j} > 0$ for every $j \in [m]$. Then, for any fixed $\mathbf{W} = (\mathbf{w}_1^\top, \ldots, \mathbf{w}_m^\top)^\top$, if it satisfies*

$$\|\mathbf{w}_{0j} - \mathbf{w}_j\| \leq \frac{c_{m,j}}{2} \quad \text{for all } j \in [m],$$

*we have*

$$\left\|\widehat{\Theta}_m^{(k)}(\mathbf{X}; \mathbf{W}) - \widehat{\Theta}_m^{(k)}(\mathbf{X}; \mathbf{W}_0)\right\|_2 \leq \frac{nM^2}{4d^2} \sum_{j=1}^{m} \lambda_{m,j}^{(k)} c_{m,j}^2 + \frac{nM}{d^{3/2}} \sqrt{\sum_{j=1}^{m} \lambda_{m,j}^{(k)} c_{m,j}^2} \quad \text{for all } k \in [2]$$

*and*

$$\text{eig}_{\min}(\widehat{\Theta}_m(\mathbf{X}; \mathbf{W})) \geq \text{eig}_{\min}(\widehat{\Theta}_m^{(1)}(\mathbf{X}; \mathbf{W}_0)) - \left(\frac{nM^2\gamma}{4d^2m} \sum_{j=1}^{m} c_{m,j}^2 + \frac{nM\gamma}{d^{3/2}m^{1/2}} \sqrt{\sum_{j=1}^{m} c_{m,j}^2}\right). \tag{S.19}$$

Note that this lemma has a deterministic conclusion, although its original counterpart (Lemma E.2) has a probabilistic one.

*Proof.* The beginning part of the proof is essentially an abbreviated version of the beginning part of the proof of Lemma E.2. This repetition is intended to help the reader by not forcing her or him to look at the proof of Lemma E.2 beforehand.

For $k \in [2]$, let

$$f_m^{(k)}(-; \mathbf{W}) : \mathbb{R}^d \to \mathbb{R}, \qquad f_m^{(k)}(\mathbf{x}; \mathbf{W}) = \sum_{j=1}^{m} \sqrt{\lambda_{m,j}^{(k)}} a_j \sigma(Z_j(\mathbf{x}; \mathbf{W})),$$

and define $\nabla_{\mathbf{W}} f_m^{(k)}(\mathbf{X}; \mathbf{W})$ to be the $n$-by-$(pd)$ matrix whose $i$-th row is the $md$-dimensional row vector $(\nabla_{\mathbf{W}} f_m^{(k)}(\mathbf{x}_i; \mathbf{W}))^\top$.

For all $k \in [2]$, we have

$$\left\|\widehat{\Theta}_m^{(k)}(\mathbf{X}; \mathbf{W}) - \widehat{\Theta}_m^{(k)}(\mathbf{X}; \mathbf{W}_0)\right\|_2$$

$$= \left\|\nabla_{\mathbf{W}} f_m^{(k)}(\mathbf{X}; \mathbf{W}) \nabla_{\mathbf{W}} f_m^{(k)}(\mathbf{X}; \mathbf{W})^\top - \nabla_{\mathbf{W}} f_m^{(k)}(\mathbf{X}; \mathbf{W}_0) \nabla_{\mathbf{W}} f_m^{(k)}(\mathbf{X}; \mathbf{W}_0)^\top\right\|_2$$

$$\leq \left\|\nabla_{\mathbf{W}} f_m^{(k)}(\mathbf{X}; \mathbf{W}) - \nabla_{\mathbf{W}} f_m^{(k)}(\mathbf{X}; \mathbf{W}_0)\right\|_2^2 \tag{S.20}$$

$$+ 2 \left\|\nabla_{\mathbf{W}} f_m^{(k)}(\mathbf{X}; \mathbf{W}_0)\right\|_2 \left\|\nabla_{\mathbf{W}} f_m^{(k)}(\mathbf{X}; \mathbf{W}) - \nabla_{\mathbf{W}} f_m^{(k)}(\mathbf{X}; \mathbf{W}_0)\right\|_2.$$

To see why this inequality holds, see the proof of Lemma E.2. We bound the two terms $\left\|\nabla_{\mathbf{W}} f_m^{(k)}(\mathbf{X}; \mathbf{W}_0)\right\|_2$ and $\left\|\nabla_{\mathbf{W}} f_m^{(k)}(\mathbf{X}; \mathbf{W}) - \nabla_{\mathbf{W}} f_m^{(k)}(\mathbf{X}; \mathbf{W}_0)\right\|_2$ in Equation (S.20). We bound the first term as follows:

$$\left\|\nabla_{\mathbf{W}} f_m^{(k)}(\mathbf{X}; \mathbf{W}_0)\right\|_2^2 \leq \left\|\nabla_{\mathbf{W}} f_m^{(k)}(\mathbf{X}; \mathbf{W}_0)\right\|_F^2 = \sum_{i=1}^{n} \sum_{j=1}^{m} \left\|\nabla_{\mathbf{w}_j} f_m^{(k)}(\mathbf{x}_i; \mathbf{W}_0)\right\|^2$$

$$= \sum_{i=1}^{n} \sum_{j=1}^{m} \lambda_{m,j}^{(k)} |\sigma'(Z_j(\mathbf{x}_i; \mathbf{W}_0))|^2 \frac{\|\mathbf{x}_i\|^2}{d}$$

$$\leq \frac{n}{d} \sum_{j=1}^{m} \lambda_{m,j}^{(k)} \leq \frac{n}{d} \gamma_k$$

where $\gamma_1 = \gamma$ and $\gamma_2 = 1 - \gamma$. The second inequality uses the assumption that $|\sigma'(x)| \leq 1$ for all $x \in \mathbb{R}$ and $\|\mathbf{x}_i\| \leq 1$ for all $i \in [n]$. The third inequality holds because $\sum_{j=1}^m \lambda_{m,j}^{(k)} \leq \sum_{j=1}^m \lambda_{m,j} = 1$. For the second term, we recall that $|\sigma''(x)| \leq M$ and so $\sigma'$ is $M$-Lipschitz, and also that $Z_j(\mathbf{x}_i; \mathbf{W}) = \frac{1}{\sqrt{d}} \mathbf{w}_j^\top \mathbf{x}_i$. Using these facts, we derive an upper bound for the second term as follows:

$$
\left\| \nabla_{\mathbf{W}} f_m^{(k)}(\mathbf{X}; \mathbf{W}) - \nabla_{\mathbf{W}} f_m^{(k)}(\mathbf{X}; \mathbf{W}_0) \right\|_2^2
$$

$$
\leq \left\| \nabla_{\mathbf{W}} f_m^{(k)}(\mathbf{X}; \mathbf{W}) - \nabla_{\mathbf{W}} f_m^{(k)}(\mathbf{X}; \mathbf{W}_0) \right\|_F^2
$$

$$
= \sum_{i=1}^n \sum_{j=1}^m \left\| \nabla_{\mathbf{w}_j} f_m^{(k)}(\mathbf{x}_i; \mathbf{W}) - \nabla_{\mathbf{w}_j} f_m^{(k)}(\mathbf{x}_i; \mathbf{W}_0) \right\|^2
$$

$$
= \sum_{i=1}^n \sum_{j=1}^m \left\| \sqrt{\lambda_{m,j}^{(k)}} a_j \frac{\mathbf{x}_i}{\sqrt{d}} \left[ \sigma'(Z_j(\mathbf{x}_i; \mathbf{W})) - \sigma'(Z_j(\mathbf{x}_i; \mathbf{W}_0)) \right] \right\|^2
$$

$$
= \frac{1}{d} \sum_{i=1}^n \|\mathbf{x}_i\|^2 \sum_{j=1}^m \lambda_{m,j}^{(k)} \left[ \sigma'(Z_j(\mathbf{x}_i; \mathbf{W})) - \sigma'(Z_j(\mathbf{x}_i; \mathbf{W}_0)) \right]^2
$$

$$
\leq \frac{1}{d} \sum_{i=1}^n \sum_{j=1}^m \lambda_{m,j}^{(k)} \left[ \sigma'(Z_j(\mathbf{x}_i; \mathbf{W})) - \sigma'(Z_j(\mathbf{x}_i; \mathbf{W}_0)) \right]^2
$$

$$
\leq \frac{M^2}{d^2} \sum_{i=1}^n \sum_{j=1}^m \lambda_{m,j}^{(k)} \left( (\mathbf{w}_j - \mathbf{w}_{0j})^\top \mathbf{x}_i \right)^2
$$

$$
\leq \frac{nM^2}{d^2} \sum_{j=1}^m \lambda_{m,j}^{(k)} \|\mathbf{w}_j - \mathbf{w}_{0j}\|^2
$$

$$
\leq \frac{nM^2}{4d^2} \sum_{j=1}^m \lambda_{m,j}^{(k)} c_{m,j}^2.
$$

The second to last step uses the Cauchy-Schwartz inequality, and the last step uses our assumption that $\|\mathbf{w}_j - \mathbf{w}_{0j}\| \leq \frac{c_{m,j}}{2}$ for all $j \in [m]$. From the derived bounds on the first and second terms in the last line of Equation (S.20), it follows that

$$
\left\| \widehat{\Theta}_m^{(k)}(\mathbf{X}; \mathbf{W}) - \widehat{\Theta}_m^{(k)}(\mathbf{X}; \mathbf{W}_0) \right\|_2 \leq \frac{nM^2}{4d^2} \sum_{j=1}^m \lambda_{m,j}^{(k)} c_{m,j}^2 + 2\sqrt{\frac{n}{d} \gamma_k} \sqrt{\frac{nM^2}{4d^2} \sum_{j=1}^m \lambda_{m,j}^{(k)} c_{m,j}^2}
$$

$$
= \frac{nM^2}{4d^2} \sum_{j=1}^m \lambda_{m,j}^{(k)} c_{m,j}^2 + \frac{nM}{d^{3/2}} \sqrt{\gamma_k \sum_{j=1}^m \lambda_{m,j}^{(k)} c_{m,j}^2}.
$$

Finally, as noted in the proof of Lemma E.2, we have

$$
\mathrm{eig}_{\min}(\widehat{\Theta}_m(\mathbf{X}; \mathbf{W})) \geq \mathrm{eig}_{\min}(\widehat{\Theta}_m^{(1)}(\mathbf{X}; \mathbf{W}))
$$

$$
\geq \mathrm{eig}_{\min}(\widehat{\Theta}_m^{(1)}(\mathbf{X}; \mathbf{W}_0)) - \left\| \widehat{\Theta}_m^{(1)}(\mathbf{X}; \mathbf{W}) - \widehat{\Theta}_m^{(1)}(\mathbf{X}; \mathbf{W}_0) \right\|_2.
$$

Thus,

$$
\mathrm{eig}_{\min}(\widehat{\Theta}_m(\mathbf{X}; \mathbf{W})) \geq \mathrm{eig}_{\min}(\widehat{\Theta}_m^{(1)}(\mathbf{X}; \mathbf{W}_0)) - \left( \frac{nM^2\gamma}{4d^2m} \sum_{j=1}^m c_{m,j}^2 + \frac{nM\gamma}{d^{3/2}m^{1/2}} \sqrt{\sum_{j=1}^m c_{m,j}^2} \right).
$$

$\square$

### E.5 Lemma on a sufficient condition for Theorem 5.1 - Smooth activation case

We now give a version of Lemma E.3 for the smooth activation case (i.e., under Assumption 3.2). It brings together the results from Proposition D.1 and Lemmas E.1 and E.4, and identifies a sufficient condition for Theorem A.1, which corresponds to the condition in Lemma 3.4 in (Du et al., 2019b).

**Lemma E.5.** *Assume that Assumptions 3.1 to 3.3 hold. Let $\delta \in (0, 1)$, and $c_{m,j} > 0$ for all $j \in [m]$. Assume that $\gamma > 0$ and*

$$m \geq \max \left( \frac{8n \log \frac{2n}{\delta}}{d\kappa_n}, \ \frac{nM^2\delta^2}{8d^2\kappa_n} \sum_{j=1}^{m} c_{m,j}^2, \ \frac{4n^2M^2\delta^2}{d^3\kappa_n^2} \sum_{j=1}^{m} c_{m,j}^2 \right).$$

*For each $j \in [m]$, define*

$$R'_{m,j} = \sqrt{\frac{n\lambda_{m,j}}{d}} \|\mathbf{y} - \mathbf{u}_0\| \frac{4}{\gamma\kappa_n} \quad and \quad R_{m,j} = \frac{\delta c_{m,j}}{8}.$$

*If $R'_{m,j} < R_{m,j}$ for all $j \in [m]$ with probability at least $1 - \frac{\delta}{2}$, then on an event with probability at least $1 - \delta$, we have that for all $j \in [m]$, $R'_{m,j} < R_{m,j}$ and the following properties also hold for all $t \geq 0$:*

(a) $\mathrm{eig}_{\min}(\widehat{\Theta}_m(\mathbf{X}; \mathbf{W}_t)) \geq \frac{\gamma\kappa_n}{4}$;

(b) $L_m(\mathbf{W}_t) \leq e^{-(\gamma\kappa_n t)/2} L_m(\mathbf{W}_0)$;

(c) $\|\mathbf{w}_{tj} - \mathbf{w}_{0j}\| \leq R'_{m,j}$ for all $j \in [m]$; and

(d) $\|\widehat{\Theta}_m(\mathbf{X}; \mathbf{W}_t) - \widehat{\Theta}_m(\mathbf{X}; \mathbf{W}_0)\|_2 \leq \frac{nM^2\delta^2}{8^2 d^2} \sum_{j=1}^{m} \lambda_{m,j} c_{m,j}^2 + \frac{nM\delta}{2^{3/2} d^{3/2}} \sqrt{\sum_{j=1}^{m} \lambda_{m,j} c_{m,j}^2}$.

*Proof.* The proof is very similar to that of Lemma E.3, although the concrete bounds in these proofs differ due to the differences between Lemma E.2 and Lemma E.4.

Suppose $R'_{m,j} < R_{m,j}$ for all $j \in [m]$ on some event $A'$ having probability at least $1 - \frac{\delta}{2}$. Also, we would like to instantiate Proposition D.1 with $\delta/2$, so that its claim holds with probability at least $1 - \frac{\delta}{2}$. Let $A$ be the intersection of $A'$ with the event that the claim in Proposition D.1 holds with $\delta/2$. By the union bound, $A$ has probability at least $1 - \delta$. We will show that on the event $A$, the four claimed properties of the lemma hold.

It will be sufficient to show that

$$\|\mathbf{w}_{sj} - \mathbf{w}_{0j}\| \leq R_{m,j} \quad \text{for all } s \geq 0. \tag{S.21}$$

To see why doing so is sufficient, pick an arbitrary $t_0 \geq 0$, and assume the above inequality for all $s \geq 0$. Then, by the event $A$ and Lemma E.4, for all $0 \leq s \leq t_0$, we have the following upper bound on the change of the Gram matrix from time 0 to $s$, and the following lower bound on the smallest eigenvalue of $\widehat{\Theta}_m(\mathbf{X}; \mathbf{W}_s)$:

$$\left\| \widehat{\Theta}_m(\mathbf{X}; \mathbf{W}_s) - \widehat{\Theta}_m(\mathbf{X}; \mathbf{W}_0) \right\|_2$$
$$\leq \left\| \widehat{\Theta}_m^{(1)}(\mathbf{X}; \mathbf{W}_s) - \widehat{\Theta}_m^{(1)}(\mathbf{X}; \mathbf{W}_0) \right\|_2 + \left\| \widehat{\Theta}_m^{(2)}(\mathbf{X}; \mathbf{W}_s) - \widehat{\Theta}_m^{(2)}(\mathbf{X}; \mathbf{W}_0) \right\|_2$$
$$\leq \frac{nM^2\delta^2}{64d^2} \sum_{j=1}^{m} \lambda_{m,j} c_{m,j}^2 + \frac{nM\delta}{2^{3/2} d^{3/2}} \sqrt{\sum_{j=1}^{m} \lambda_{m,j} c_{m,j}^2}$$

and

$$\text{eig}_{\min}(\widehat{\Theta}_m(\mathbf{X}; \mathbf{W}_s)) \geq \text{eig}_{\min}(\widehat{\Theta}_m^{(1)}(\mathbf{X}; \mathbf{W}_0)) - \left( \frac{nM^2\delta^2\gamma}{64d^2m} \sum_{j=1}^m c_{m,j}^2 + \frac{nM\delta\gamma}{4d^{3/2}m^{1/2}} \sqrt{\sum_{j=1}^m c_{m,j}^2} \right)$$

$$> \frac{\gamma\kappa_n}{2} - \frac{\gamma\kappa_n}{4} \left( \frac{1}{m} \cdot \frac{nM^2\delta^2}{16d^2\kappa_n} \sum_{j=1}^m c_{m,j}^2 + \frac{1}{m^{1/2}} \cdot \frac{nM\delta}{d^{3/2}\kappa_n} \sqrt{\sum_{j=1}^m c_{m,j}^2} \right)$$

$$\geq \frac{\gamma\kappa_n}{2} - \frac{\gamma\kappa_n}{4} \left( \frac{1}{2} + \frac{1}{2} \right) = \frac{\gamma\kappa_n}{4}.$$

We now apply the version of Lemma E.1 for the analytic activation $\sigma$, with $\zeta$ being set to $\frac{\gamma\kappa_n}{2}$. This application gives

$$L_m(\mathbf{W}_{t_0}) \leq e^{-(\gamma\kappa_n t_0)/2} L_m(\mathbf{W}_0)$$

and

$$\|\mathbf{w}_{t_0 j} - \mathbf{w}_{0j}\| \leq \sqrt{\frac{n\lambda_{m,j}}{d}} \|\mathbf{y} - \mathbf{u}_0\| \frac{4}{\gamma\kappa_n} = R'_{m,j} \quad \text{for all } j \in [m].$$

We have just shown that all the four properties in the lemma hold for $t_0$.

It remains to prove Equation (S.21) under the event $A$. Suppose that Equation (S.21) fails for some $j \in [m]$. Let

$$t_1 = \inf \{t \mid \|\mathbf{w}_{tj} - \mathbf{w}_{0j}\| > R_{m,j} \text{ for some } j \in [m]\}.$$

Then, by the continuity of $\mathbf{w}_{tj}$ on $t$, we have

$$\|\mathbf{w}_{sj} - \mathbf{w}_{0j}\| \leq R_{m,j} \quad \text{for all } j \in [m] \text{ and } 0 \leq s \leq t_1$$

and for some $j_0 \in [m]$,

$$\|\mathbf{w}_{t_1 j_0} - \mathbf{w}_{0j_0}\| = R_{m,j_0}. \tag{S.22}$$

Thus, by the argument that we gave in the previous paragraph, we have

$$\|\mathbf{w}_{t_1 j} - \mathbf{w}_{0j}\| \leq R'_{m,j} \quad \text{for all } j \in [m].$$

In particular, $\|\mathbf{w}_{t_1 j_0} - \mathbf{w}_{0j_0}\| \leq R'_{m,j_0}$. But this contradicts our assumption $R'_{m,j_0} < R_{m,j_0}$. □

## F Proof of Theorem A.1 on the global convergence of gradient flow (ReLU case)

The proof of Theorem A.1 essentially follows Lemma E.3, which itself follows from the secondary Proposition D.1 and Lemmas E.1 and E.2, derived in Appendices D and E. Pick $\delta \in (0,1)$. Let

$$D = \sqrt{n^2 \left( C^2 + \frac{1}{d} \right) \frac{2 \cdot 512^2}{\gamma^2 \delta^5 \kappa_n^2 d}}$$

where $C$ is the assumed upper bound on the $|y_i|$'s. Assume $\gamma > 0$ and

$$m \geq \max \left( \left( \frac{8n \log \frac{4n}{\delta}}{\kappa_n d} \right), \left( \frac{8nD}{d\kappa_n} \right)^2, \left( \frac{16^2 n^2 D}{d^2 \kappa_n^2} \right)^2 \right)$$

and set $c_{m,j}$ as follows:

$$c_{m,j} = \sqrt{\lambda_{m,j}} \cdot \sqrt{n^2 \left( C^2 + \frac{1}{d} \right) \frac{2 \cdot 512^2}{\gamma^2 \delta^5 \kappa_n^2 d}} = \sqrt{\lambda_{m,j}} \cdot D.$$

Note that

$$\left(\frac{8n}{d\kappa_n}\sum_{j=1}^{m}c_{m,j}\right)^2 = \left(\frac{8nD}{d\kappa_n}\right)^2 \cdot \left(\sum_{j=1}^{m}\sqrt{\lambda_{m,j}}\right)^2 \le \left(\frac{8nD}{d\kappa_n}\right)^2 \cdot \left(\sum_{j=1}^{m}\lambda_{m,j}\right) \cdot m$$

$$= \left(\frac{8nD}{d\kappa_n}\right)^2 \cdot m \le m^2,$$

and also that

$$\left(\frac{16^2 n^2}{d^2\kappa_n^2}\sum_{j=1}^{m}c_{m,j}\right)^2 = \left(\frac{16^2 n^2 D}{d^2\kappa_n^2}\right)^2 \cdot \left(\sum_{j=1}^{m}\sqrt{\lambda_{m,j}}\right)^2 \le \left(\frac{16^2 n^2 D}{d^2\kappa_n^2}\right)^2 \cdot \left(\sum_{j=1}^{m}\lambda_{m,j}\right) \cdot m$$

$$= \left(\frac{16^2 n^2 D}{d^2\kappa_n^2}\right)^2 \cdot m \le m^2.$$

Thus,

$$m \ge \max\left(\left(\frac{8n\log\frac{4n}{\delta}}{d\kappa_n}\right), \left(\frac{8n}{d\kappa_n}\sum_{j=1}^{m}c_{m,j}\right), \left(\frac{16^2 n^2}{d^2\kappa_n^2}\sum_{j=1}^{m}c_{m,j}\right)\right).$$

As a result, we can now employ Lemma E.3. Thus, if we find an event $A'$ such that the probability of $A'$ is at least $1 - (\delta/2)$ and under $A'$, we have $R'_{m,j} < R_{m,j}$, then the conclusion of Lemma E.3 holds. In particular, we may further calculate conclusions (c) and (d) of Lemma E.3 as

$$\|\mathbf{w}_{tj} - \mathbf{w}_{0j}\| \le R'_{m,j} < R_{m,j} = \frac{\delta^2 c_{m,j}}{64} = \frac{\delta^2}{64} \cdot \sqrt{\lambda_{m,j}} \cdot \sqrt{n^2\left(C^2 + \frac{1}{d}\right)\frac{2\cdot512^2}{\gamma^2\delta^5\kappa_n^2 d}}$$

$$= \frac{8n}{\kappa_n d^{1/2}} \cdot \sqrt{\left(C^2 + \frac{1}{d}\right)\frac{2}{\gamma^2\delta}} \cdot \sqrt{\lambda_{m,j}},$$

and

$$\|\widehat{\Theta}_m(\mathbf{X};\mathbf{W}_t) - \widehat{\Theta}_m(\mathbf{X};\mathbf{W}_0)\|_2 \le \frac{n}{d}\sum_{j=1}^{m}\lambda_{m,j}c_{m,j} + \frac{2\sqrt{2}\cdot n}{d}\sqrt{\sum_{j=1}^{m}\lambda_{m,j}c_{m,j}}$$

$$= \frac{n}{d}\cdot D\cdot\sum_{j=1}^{m}\lambda_{m,j}^{3/2} + \frac{2\sqrt{2}\cdot n}{d}\cdot\sqrt{D}\cdot\sqrt{\sum_{j=1}^{m}\lambda_{m,j}^{3/2}}$$

$$= \frac{n}{d}\cdot\sqrt{n^2\left(C^2 + \frac{1}{d}\right)\frac{2\cdot512^2}{\gamma^2\delta^5\kappa_n^2 d}}\cdot\sum_{j=1}^{m}\lambda_{m,j}^{3/2}$$

$$+ \frac{2\sqrt{2}\cdot n}{d}\cdot\left(n^2\left(C^2 + \frac{1}{d}\right)\frac{2\cdot512^2}{\gamma^2\delta^5\kappa_n^2 d}\right)^{1/4}\cdot\sqrt{\sum_{j=1}^{m}\lambda_{m,j}^{3/2}}$$

$$= \frac{512n^2}{\kappa_n d^{3/2}}\cdot\sqrt{\left(C^2 + \frac{1}{d}\right)\frac{2}{\gamma^2\delta^5}}\cdot\sum_{j=1}^{m}\lambda_{m,j}^{3/2}$$

$$+ \frac{64n^{3/2}}{\kappa_n^{1/2}d^{5/4}}\cdot\left(\left(C^2 + \frac{1}{d}\right)\frac{2}{\gamma^2\delta^5}\right)^{1/4}\cdot\sqrt{\sum_{j=1}^{m}\lambda_{m,j}^{3/2}}.$$

It remains to find such an event $A'$. Start by noting that

$$\mathbb{E}[\|\mathbf{y} - \mathbf{u}_0\|^2] = \sum_{i=1}^{n} \left( y_i^2 - 2y_i \mathbb{E}[f_m(\mathbf{x}_i; \mathbf{W}_0)] + \mathbb{E}[f_m(\mathbf{x}_i; \mathbf{W}_0)^2] \right)$$

$$= \sum_{i=1}^{n} \left( y_i^2 - 2y_i \cdot 0 + \mathbb{E}\left[ \frac{1}{d} \sum_{j=1}^{m} \lambda_{m,j} (\mathbf{w}_j^\top \mathbf{x}_i)^2 \mathbf{1}_{\{\mathbf{w}_j^\top \mathbf{x}_i \geq 0\}} \right] \right)$$

$$= \sum_{i=1}^{n} \left( y_i^2 + \frac{1}{d} \sum_{j=1}^{m} \lambda_{m,j} \mathbb{E}\left[ (\mathbf{w}_j^\top \mathbf{x}_i)^2 \mathbf{1}_{\{\mathbf{w}_j^\top \mathbf{x}_i \geq 0\}} \right] \right)$$

$$\leq n \left( C^2 + \frac{1}{d} \right).$$

Thus, by Markov inequality, with probability at least $1 - (\delta/2)$,

$$\|\mathbf{y} - \mathbf{u}_0\|^2 < n \left( C^2 + \frac{1}{d} \right) \frac{2}{\delta}.$$

Let $A'$ be the corresponding event for the above inequality. Then, under $A'$, we have

$$R'_{m,j} = \sqrt{\frac{n \lambda_{m,j}}{d}} \, \|\mathbf{y} - \mathbf{u}_0\| \, \frac{4}{\gamma \kappa_n}$$

$$< \sqrt{\frac{n \lambda_{m,j}}{d}} \cdot \sqrt{n \left( C^2 + \frac{1}{d} \right) \frac{2}{\delta}} \cdot \frac{4}{\gamma \kappa_n}$$

$$= \sqrt{\lambda_{m,j}} \cdot \sqrt{n^2 \left( C^2 + \frac{1}{d} \right) \frac{2 \cdot 4^2}{\gamma^2 \delta \kappa_n^2 d}}$$

$$= \frac{\delta^2 c_{m,j}}{128} < \frac{\delta^2 c_{m,j}}{64} = R_{m,j}.$$

Thus, $A'$ is the desired event.

## G  Proof of Theorem 5.1 on the global convergence of gradient flow (smooth case)

The proof of the theorem is similar to that of Theorem A.1. It derives from Lemma E.5, which itself follows from the secondary Proposition D.1 and Lemmas E.1 and E.4, derived in Appendices D and E. Recall that

$$C_1 = \sup_{c \in (0,1]} \mathbb{E}[\sigma(cz)^2]$$

where the expectation is taken over the real-valued random variable $z$ with the distribution $\mathcal{N}(0, 1/d)$. To see that $C_1$ is finite, note that since $|\sigma'(x)| \leq 1$ for all $x \in \mathbb{R}$, we have

$$|\sigma(cz) - \sigma(0)| \leq |cz| \text{ for all } c \in (0,1].$$

Thus, for every $c \in (0,1]$,

$$\sigma(0) - |cz| \leq \sigma(cz) \leq \sigma(0) + |cz|,$$

which implies that

$$\mathbb{E}[\sigma(cz)^2] \leq \sigma(0)^2 + 2|\sigma(0)| \cdot |c| \cdot \mathbb{E}[|z|] + c^2 \mathbb{E}[z^2]$$

$$\leq \sigma(0)^2 + 2|\sigma(0)| \cdot \mathbb{E}[|z|] + \mathbb{E}[z^2].$$

As a result, $\mathbb{E}[\sigma(cz)^2]$ is bounded, so $C_1$ is finite.

Pick $\delta \in (0, 1)$. Assume $\gamma > 0$ and

$$m \geq \max\left(\left(\frac{8n}{\kappa_n d} \cdot \log \frac{2n}{\delta}\right), \left(\frac{2^{10} n^3 M^2}{\kappa_n^3 d^3} \cdot \frac{C^2 + C_1}{\gamma^2 \delta}\right), \left(\frac{2^{15} n^4 M^2}{\kappa_n^4 d^4} \cdot \frac{C^2 + C_1}{\gamma^2 \delta}\right)\right)$$

and instantiate Lemma E.5 using the below $c_{m,j}$:

$$c_{m,j} = \sqrt{\lambda_{m,j}} \cdot \sqrt{n^2 \left(C^2 + C_1\right) \frac{2 \cdot 64^2}{\gamma^2 \delta^3 \kappa_n^2 d}}$$

where $C$ is the assumed upper bound on the $|y_i|$'s. Note that

$$\frac{nM^2\delta^2}{8d^2\kappa_n} \sum_{j=1}^{m} c_{m,j}^2 = \frac{nM^2\delta^2}{8d^2\kappa_n} \sum_{j=1}^{m} \left(\lambda_{m,j} \cdot n^2 \left(C^2 + C_1\right) \frac{2 \cdot 64^2}{\gamma^2 \delta^3 \kappa_n^2 d}\right)$$

$$= \frac{nM^2\delta^2}{8d^2\kappa_n} \cdot n^2 \left(C^2 + C_1\right) \frac{2 \cdot 64^2}{\gamma^2 \delta^3 \kappa_n^2 d} \cdot \sum_{j=1}^{m} \lambda_{m,j}$$

$$= \frac{2^{10} n^3 M^2}{\kappa_n^3 d^3} \times \frac{C^2 + C_1}{\gamma^2 \delta}$$

and

$$\frac{4n^2 M^2 \delta^2}{d^3 \kappa_n^2} \sum_{j=1}^{m} c_{m,j}^2 = \frac{4n^2 M^2 \delta^2}{d^3 \kappa_n^2} \sum_{j=1}^{m} \left(\lambda_{m,j} \cdot n^2 \left(C^2 + C_1\right) \frac{2 \cdot 64^2}{\gamma^2 \delta^3 \kappa_n^2 d}\right)$$

$$= \frac{4n^2 M^2 \delta^2}{d^3 \kappa_n^2} \cdot n^2 \left(C^2 + C_1\right) \frac{2 \cdot 64^2}{\gamma^2 \delta^3 \kappa_n^2 d} \cdot \sum_{j=1}^{m} \lambda_{m,j}$$

$$= \frac{2^{15} n^4 M^2}{\kappa_n^4 d^4} \times \frac{C^2 + C_1}{\gamma^2 \delta}.$$

Thus,

$$m \geq \max\left(\frac{8n \log \frac{2n}{\delta}}{d\kappa_n}, \frac{nM^2\delta^2}{8d^2\kappa_n} \sum_{j=1}^{m} c_{m,j}^2, \frac{4n^2 M^2 \delta^2}{d^3 \kappa_n^2} \sum_{j=1}^{m} c_{m,j}^2\right).$$

This allows us to employ Lemma E.5. Hence, it is sufficient to find an event $A'$ such that the probability of $A'$ is at least $1 - (\delta/2)$ and under $A'$, we have $R'_{m,j} < R_{m,j}$. The desired conclusion then follows from the conclusion of Lemma E.5, and the below calculations: if $\|\mathbf{w}_{tj} - \mathbf{w}_{0j}\| \leq R'_{m,j}$ and $R'_{m,j} < R_{m,j}$, then

$$\|\mathbf{w}_{tj} - \mathbf{w}_{0j}\| < R_{m,j} = \frac{\delta c_{m,j}}{8}$$

$$= \frac{\delta}{8} \cdot \sqrt{\lambda_{m,j}} \cdot \sqrt{n^2 \left(C^2 + C_1\right) \frac{2 \cdot 64^2}{\gamma^2 \delta^3 \kappa_n^2 d}}$$

$$= \sqrt{\lambda_{m,j}} \times \frac{n}{\kappa_n d^{1/2}} \sqrt{\frac{128(C^2 + C_1)}{\gamma^2 \delta}},$$

and the upper bound on $\|\widehat{\Theta}_m(\mathbf{X}; \mathbf{W}_t) - \widehat{\Theta}_m(\mathbf{X}; \mathbf{W}_0)\|_2$ in the conclusion of Lemma E.5 can be rewritten to

$$
\|\widehat{\Theta}_m(\mathbf{X}; \mathbf{W}_t) - \widehat{\Theta}_m(\mathbf{X}; \mathbf{W}_0)\|_2
$$

$$
\leq \frac{nM^2\delta^2}{8^2 d^2} \sum_{j=1}^m \lambda_{m,j} c_{m,j}^2 + \frac{nM\delta}{2^{3/2} d^{3/2}} \sqrt{\sum_{j=1}^m \lambda_{m,j} c_{m,j}^2}
$$

$$
= \frac{nM^2\delta^2}{4^3 d^2} \sum_{j=1}^m \lambda_{m,j} \left( \lambda_{m,j} n^2 \frac{(C^2 + C_1) 2 \cdot 64^2}{\gamma^2 \delta^3 \kappa_n^2 d} \right)
$$

$$
+ \frac{nM\delta}{2^{3/2} d^{3/2}} \sqrt{\sum_{j=1}^m \lambda_{m,j} \left( \lambda_{m,j} n^2 \frac{(C^2 + C_1) 2 \cdot 64^2}{\gamma^2 \delta^3 \kappa_n^2 d} \right)}
$$

$$
= \left( \frac{n^3 M^2}{\kappa_n^2 d^3} \sum_{j=1}^m \lambda_{m,j}^2 \frac{2^7(C^2 + C_1)}{\gamma^2 \delta} \right) + \frac{n^2 M}{\kappa_n d^2} \sqrt{\sum_{j=1}^m \lambda_{m,j}^2 \frac{2^{10}(C^2 + C_1)}{\gamma^2 \delta}}.
$$

Note that

$$
\mathbb{E}[\|\mathbf{y} - \mathbf{u}_0\|^2] = \sum_{i=1}^n \left( y_i^2 - 2y_i \mathbb{E}[f_m(\mathbf{x}_i; \mathbf{W}_0)] + \mathbb{E}[f_m(\mathbf{x}_i; \mathbf{W}_0)^2] \right)
$$

$$
= \sum_{i=1}^n \left( y_i^2 - 2y_i \cdot 0 + \mathbb{E}\left[ \sum_{j=1}^m \lambda_{m,j} \sigma(Z_j(\mathbf{x}_i; \mathbf{W}_0))^2 \right] \right)
$$

$$
= \sum_{i=1}^n \left( y_i^2 + \sum_{j=1}^m \lambda_{m,j} \mathbb{E}\left[ \sigma(Z_j(\mathbf{x}_i; \mathbf{W}_0))^2 \right] \right)
$$

$$
\leq n\left( C^2 + C_1 \right).
$$

Thus, by Markov inequality, with probability at least $1 - (\delta/2)$,

$$
\|\mathbf{y} - \mathbf{u}_0\|^2 < n\left( C^2 + C_1 \right) \frac{2}{\delta}.
$$

Let $A'$ be the corresponding event for the above inequality. Then, under $A'$, we have

$$
R'_{m,j} = \sqrt{\frac{n\lambda_{m,j}}{d}} \|\mathbf{y} - \mathbf{u}_0\| \frac{4}{\gamma \kappa_n}
$$

$$
< \sqrt{\frac{n\lambda_{m,j}}{d}} \cdot \sqrt{n\left( C^2 + C_1 \right) \frac{2}{\delta}} \cdot \frac{4}{\gamma \kappa_n}
$$

$$
= \sqrt{\lambda_{m,j}} \cdot \sqrt{n^2\left( C^2 + C_1 \right) \frac{2 \cdot 4^2}{\gamma^2 \delta \kappa_n^2 d}}
$$

$$
= \frac{\delta c_{m,j}}{16} < \frac{\delta c_{m,j}}{8} = R_{m,j}.
$$

Thus, $A'$ is the desired event.

## H Proof of Theorem 6.1 on the global convergence of gradient descent (smooth activation)

Our convergence proof follows the structure of the convergence proof of (Du et al., 2019a, Theorem 5.1) with necessary modifications, which in particular account for the changing weights and Gram matrices in our setup.

### H.1 Sketch of the proof

The proof is by induction on the number of gradient-update steps $s$. Here is a sketch of the proof for the inductive case. We start by decomposing the error at step $s + 1$:

$$
\begin{aligned}
\|\mathbf{y} - \mathbf{u}_{s+1}\|^2 &= \|(\mathbf{y} - \mathbf{u}_s) - (\mathbf{u}_{s+1} - \mathbf{u}_s)\|^2 \\
&= \|\mathbf{y} - \mathbf{u}_s\|^2 - 2(\mathbf{y} - \mathbf{u}_s)^\top(\mathbf{u}_{s+1} - \mathbf{u}_s) + \|\mathbf{u}_{s+1} - \mathbf{u}_s\|^2 \\
&= \|\mathbf{y} - \mathbf{u}_s\|^2 - 2(\mathbf{y} - \mathbf{u}_s)^\top \mathbf{I}_1 - 2(\mathbf{y} - \mathbf{u}_s)^\top \mathbf{I}_2 + \|\mathbf{u}_{s+1} - \mathbf{u}_s\|^2,
\end{aligned} \tag{S.23}
$$

where $\mathbf{I}_1 = \eta \widehat{\Theta}_m(\mathbf{X}; \mathbf{W}_s)(\mathbf{y} - \mathbf{u}_s)$ and $\mathbf{I}_2 = (\mathbf{u}_{s+1} - \mathbf{u}_s - \mathbf{I}_1)$. We can then show that with high probability, both the third and the fourth terms in Equation (S.23) are $O(\eta^2)\|\mathbf{y} - \mathbf{u}_s\|^2$, so that the sum of these terms can be bounded from above by $(\eta\gamma\kappa_n/4)\|\mathbf{y} - \mathbf{u}_s\|^2$ if $\eta$ is sufficiently small. On the other hand, the second term can be bounded using the minimum eigenvalue of the positive definite Gram matrix:

$$
\begin{aligned}
-2(\mathbf{y} - \mathbf{u}_s)^\top \mathbf{I}_1 &= \left( -2\eta(\mathbf{y} - \mathbf{u}_s)^\top \widehat{\Theta}_m(\mathbf{X}; \mathbf{W}_s)(\mathbf{y} - \mathbf{u}_s) \right) \\
&\leq -2\eta \operatorname{eig}_{\min}(\widehat{\Theta}_m(\mathbf{X}; \mathbf{W}_s))\|\mathbf{y} - \mathbf{u}_s\|^2.
\end{aligned}
$$

We will show that if the network is large enough, with high probability, $-2\eta \operatorname{eig}_{\min}(\widehat{\Theta}_m(\mathbf{X}; \mathbf{W}_s))$ in the above upper bound is at most $-3\eta\gamma\kappa_n/4$. Putting all these together gives the required bound: with high probability,

$$
\begin{aligned}
\|\mathbf{y} - \mathbf{u}_{s+1}\|^2 &\leq \|\mathbf{y} - \mathbf{u}_s\|^2 - 2(\mathbf{y} - \mathbf{u}_s)^\top \mathbf{I}_1 - 2(\mathbf{y} - \mathbf{u}_s)^\top \mathbf{I}_2 + \|\mathbf{u}_{s+1} - \mathbf{u}_s\|^2 \\
&\leq \|\mathbf{y} - \mathbf{u}_s\|^2 - \frac{3\eta\gamma\kappa_n}{4}\|\mathbf{y} - \mathbf{u}_s\|^2 + \frac{\eta\gamma\kappa_n}{4}\|\mathbf{y} - \mathbf{u}_s\|^2 \\
&\leq \left( 1 - \frac{\eta\gamma\kappa_n}{2} \right) \|\mathbf{y} - \mathbf{u}_s\|^2 \\
&\leq \left( 1 - \frac{\eta\gamma\kappa_n}{2} \right)^{s+1} \|\mathbf{y} - \mathbf{u}_0\|^2.
\end{aligned}
$$

The step of upper-bounding $-2\eta \operatorname{eig}_{\min}(\widehat{\Theta}_m(\mathbf{X}; \mathbf{W}_s))$ by $-3\eta\gamma\kappa_n/4$ is where we have to account for the changing weights and Gram matrix, and this is where the difference between our proof and that of (Du et al., 2019a) lies.

As mentioned already, the Gram matrix $\widehat{\Theta}_m(\mathbf{X}; \mathbf{W}_s)$ changes during gradient descent even when the network is very wide, but we will show that despite these changes, its minimum eigenvalue remains lower-bounded by $3\gamma\kappa_n/8$ with high probability. This can be done using the decomposition $\widehat{\Theta}_m = \widehat{\Theta}_m^{(1)} + \widehat{\Theta}_m^{(2)}$ in Equation (15) from our proof sketch of the global convergence of gradient flow. At a high level, the reasoning goes like this. The induction hypothesis implies that the weight change $\|\mathbf{w}_{sj} - \mathbf{w}_{0j}\|$ is $O(\sqrt{\lambda_{m,j}})$, which is small enough to guarantee that $\widehat{\Theta}_m^{(1)}(\mathbf{X}; \mathbf{W}_{s'})$ remains almost constant during training for a large network. This, in turn, implies that the minimum eigenvalue of $\widehat{\Theta}_m^{(1)}(\mathbf{X}; \mathbf{W}_s)$ is lower-bounded by $3\gamma\kappa_n/8$ with high probability. Since $\operatorname{eig}_{\min}(\widehat{\Theta}_m(\mathbf{X}; \mathbf{W}_s)) \geq \operatorname{eig}_{\min}(\widehat{\Theta}_m^{(1)}(\mathbf{X}; \mathbf{W}_s))$, we get the desired upper bound.

### H.2 Two key lemmas

Before proving the theorem, we show two useful facts. Let $\mathbf{u}(\mathbf{W})$ be the $n$-dimensional vector

$$
(f_m(\mathbf{x}_1; \mathbf{W}), \dots, f_m(\mathbf{x}_n; \mathbf{W}))^\top
$$

which consists of the network outputs on the training inputs under the parameters $\mathbf{W}$. Note that for each gradient-update step $s \in \mathbb{N} \cup \{0\}$, the vector $\mathbf{u}(\mathbf{W}_s)$ is equal to $\mathbf{u}_s$, the notation that we have been using in the main text of the paper. We also define $\mathbf{u}'(\mathbf{W})$ to be the following $n$-by-$m$ matrix:

$$
\mathbf{u}'(\mathbf{W}) = \frac{\partial \mathbf{u}}{\partial \mathbf{W}}.
$$

For each $s \in \mathbb{N} \cup \{0\}$, let $\widehat{\Theta}_m(s) = \widehat{\Theta}_m(\mathbf{X}; \mathbf{W}_s)$ and

$$\widetilde{\mathbf{u}}_{s+1} = \left(\mathbf{u}_s - \eta \frac{d\mathbf{u}_t}{dt}\Big|_{\mathbf{W}_t = \mathbf{W}_s}\right) = \left(\mathbf{u}_s - \eta \widehat{\Theta}_m(s)(\mathbf{u}_s - \mathbf{y})\right)$$

be the Euler discretisation of the gradient flow of the output. Here $\eta > 0$ is the learning rate.

**Lemma H.1.** *For all $\mathbf{W}$ and $j \in [m]$,*

$$\left\|\frac{\partial L_m(\mathbf{W})}{\partial \mathbf{w}_j}\right\| \leq \frac{\sqrt{\lambda_{m,j} n}}{\sqrt{d}} \|\mathbf{y} - \mathbf{u}(\mathbf{W})\|.$$

*Proof.*

$$
\begin{aligned}
\left\|\frac{\partial L_m(\mathbf{W})}{\partial \mathbf{w}_j}\right\| &= \left\|\sum_{i=1}^n (u(\mathbf{W})_i - y_i) \times \sqrt{\lambda_{m,j}} a_j \times \sigma'\left(\frac{\mathbf{w}_j^\top \mathbf{x}_i}{\sqrt{d}}\right) \times \frac{\mathbf{x}_i}{\sqrt{d}}\right\| \\
&\leq \sum_{i=1}^n \left\|(u(\mathbf{W})_i - y_i) \times \sqrt{\lambda_{m,j}} a_j \times \sigma'\left(\frac{\mathbf{w}_j^\top \mathbf{x}_i}{\sqrt{d}}\right) \times \frac{\mathbf{x}_i}{\sqrt{d}}\right\| \\
&\leq \frac{\sqrt{\lambda_{m,j}}}{\sqrt{d}} \times \sum_{i=1}^n |u(\mathbf{W})_i - y_i| \\
&\leq \frac{\sqrt{\lambda_{m,j} n}}{\sqrt{d}} \|\mathbf{y} - \mathbf{u}(\mathbf{W})\|.
\end{aligned}
$$

$\square$

The next lemma gives an upper bound on $\|\mathbf{y} - \mathbf{u}_{s+1}\|$. As we will show shortly, this upper bound will play a crucial role in the proof of Theorem 6.1.

**Lemma H.2.** *Assume Assumptions 3.1 to 3.3. Then, for all $s \in \mathbb{N} \cup \{0\}$, we have*

$$\|\mathbf{y} - \mathbf{u}_{s+1}\|^2 \leq \left(1 - 2\eta \operatorname{eig}_{\min}(\widehat{\Theta}_m(s)) + \frac{2\eta^2 M n^{3/2}}{d^2}\|\mathbf{y} - \mathbf{u}_s\| + \frac{\eta^2 n^2}{d^2}\right) \times \|\mathbf{y} - \mathbf{u}_s\|^2. \tag{S.24}$$

*Proof.* Write

$$\mathbf{u}_{s+1} - \mathbf{u}_s = \underbrace{\widetilde{\mathbf{u}}_{s+1} - \mathbf{u}_s}_{\mathbf{I}_1} + \underbrace{\mathbf{u}_{s+1} - \widetilde{\mathbf{u}}_{s+1}}_{\mathbf{I}_2}.$$

Then, we have

$$
\begin{aligned}
\|\mathbf{y} - \mathbf{u}_{s+1}\|^2 &= \|(\mathbf{y} - \mathbf{u}_s) - (\mathbf{u}_{s+1} - \mathbf{u}_s)\|^2 \\
&= \|\mathbf{y} - \mathbf{u}_s\|^2 - 2(\mathbf{y} - \mathbf{u}_s)^\top (\mathbf{u}_{s+1} - \mathbf{u}_s) + \|\mathbf{u}_{s+1} - \mathbf{u}_s\|^2 \\
&= \|\mathbf{y} - \mathbf{u}_s\|^2 - 2(\mathbf{y} - \mathbf{u}_s)^\top \mathbf{I}_1 - 2(\mathbf{y} - \mathbf{u}_s)^\top \mathbf{I}_2 + \|\mathbf{u}_{s+1} - \mathbf{u}_s\|^2.
\end{aligned}
$$

Since the Gram matrix $\widehat{\Theta}_m(s)$ is positive definite and $\eta > 0$, we have

$$
\begin{aligned}
(\mathbf{y} - \mathbf{u}_s)^\top \mathbf{I}_1 = (\mathbf{y} - \mathbf{u}_s)^\top (\widetilde{\mathbf{u}}_{s+1} - \mathbf{u}_s) &= \eta (\mathbf{y} - \mathbf{u}_s)^\top \widehat{\Theta}_m(s)(\mathbf{y} - \mathbf{u}_s) \\
&\geq \eta \operatorname{eig}_{\min}(\widehat{\Theta}_m(s)) \|\mathbf{y} - \mathbf{u}_s\|^2.
\end{aligned}
$$

We now get a bound on $\mathbf{I}_2$. Note that $\widehat{\Theta}_m(s) = \mathbf{u}_s'(\mathbf{u}_s')^\top$ where $\mathbf{u}_s' = \mathbf{u}'(\mathbf{W}_s) = \frac{\partial \mathbf{u}}{\partial \mathbf{W}}\big|_{\mathbf{W} = \mathbf{W}_s}$. Let

$$L_m'(\mathbf{W}) = \frac{\partial L_m(\mathbf{W})}{\partial \mathbf{W}} = \sum_{i=1}^n (u(\mathbf{W})_i - y_i) u'(\mathbf{W})_i = \mathbf{u}'(\mathbf{W})^\top (\mathbf{u}(\mathbf{W}) - \mathbf{y})$$

and

$$L'_m(s) = L'_m(\mathbf{W}_s).$$

Then,

$$
\begin{aligned}
\mathbf{I}_2 &= \mathbf{u}_{s+1} - \mathbf{u}_s + \eta \mathbf{u}'_s(\mathbf{u}'_s)^\top(\mathbf{u}_s - \mathbf{y}) \\
&= \left( -\int_{r=0}^{\eta} \left( \mathbf{u}'(\mathbf{W}_s - rL'_m(s)) \right) L'_m(s)\, dr \right) + \eta \mathbf{u}'_s(\mathbf{u}'_s)^\top(\mathbf{u}_s - \mathbf{y}) \\
&= \int_{r=0}^{\eta} \left( \mathbf{u}'_s - \mathbf{u}'(\mathbf{W}_s - rL'_m(s)) \right) L'_m(s)\, dr.
\end{aligned}
$$

Also,

$$\|L'_m(s)\| = \left\| \sum_{i=1}^{n}(y_i - u_{si})u'_{si} \right\| \le \sum_{i=1}^{n}|y_i - u_{si}|\, \|u'_{si}\|$$

and

$$\|u'_{si}\|^2 = \sum_{j=1}^{m}\lambda_{m,j}a_j^2 \left( \sigma'\left( \frac{\mathbf{w}_{sj}^\top \mathbf{x}_i}{\sqrt{d}} \right) \right)^2 \frac{\|\mathbf{x}_i\|^2}{d} \le \frac{1}{d},$$

since $\sum_j \lambda_{m,j} = 1$, $a_j \in \{-1, +1\}$, $\sigma'$ is 1-Lipschitz, and $\|\mathbf{x}_i\| \le 1$. Hence, by Cauchy-Schwarz,

$$\|L'_m(s)\| \le \frac{1}{\sqrt{d}}\sum_{i=1}^{n}|y_i - u_{si}| \le \frac{\sqrt{n}}{\sqrt{d}}\|\mathbf{y} - \mathbf{u}_s\|.$$

Let $\mathbf{W}_{(s,r)} = \mathbf{W}_s - rL'_m(s)$. For $j \in [m]$, write $\mathbf{w}_{(s,r)j}$ for the part of $\mathbf{W}_{(s,r)}$ going to the $j$-th node. Then, for all $i \in [n]$,

$$
\begin{aligned}
\left\| u'_{si} - u'(\mathbf{W}_{(s,r)})_i \right\|^2 &= \sum_{j=1}^{m}\lambda_{m,j}a_j^2 \left( \sigma'\left( \frac{\mathbf{w}_{sj}^\top \mathbf{x}_i}{\sqrt{d}} \right) - \sigma'\left( \frac{\mathbf{w}_{(s,r)j}^\top \mathbf{x}_i}{\sqrt{d}} \right) \right)^2 \frac{\|\mathbf{x}_i\|^2}{d} \\
&\le M^2 \sum_{j=1}^{m}\lambda_{m,j}a_j^2 \left( \left(\mathbf{w}_{sj} - \mathbf{w}_{(s,r)j}\right)^\top \mathbf{x}_i \right)^2 \frac{\|\mathbf{x}_i\|^2}{d^2} \\
&\le \frac{M^2}{d^2}\sum_{j=1}^{m}\lambda_{m,j}\left\| \mathbf{w}_{sj} - \mathbf{w}_{(s,r)j} \right\|^2 \\
&\le \frac{M^2}{d^2}\left\| \mathbf{W}_s - \mathbf{W}_{(s,r)} \right\|^2.
\end{aligned}
$$

The first inequality follows from the $M$-Lipschitz continuity of $\sigma'$, and the next inequality from $a_j \in \{-1, 1\}$, $\|\mathbf{x}_i\| \le 1$, and Cauchy-Schwartz. The last inequality uses the fact that $\sum_j \lambda_{m,j} = 1$. Finally, for all $0 \le r \le \eta$,

$$\left\| \mathbf{W}_s - \mathbf{W}_{(s,r)} \right\| = r\|L'_m(s)\| \le \eta \frac{\sqrt{n}}{\sqrt{d}}\|\mathbf{y} - \mathbf{u}_s\|.$$

Thus,

$$
\begin{aligned}
\|\mathbf{I}_2\|^2 &= \sum_{i=1}^n \left( \int_{r=0}^\eta \left( u'_{si} - u'(\mathbf{W}_{(s,r)})_i \right)^\top L'_m(s)\, dr \right)^2 \\
&\leq \sum_{i=1}^n \left( \int_{r=0}^\eta \left| \left( u'_{si} - u'(\mathbf{W}_{(s,r)})_i \right)^\top L'_m(s) \right| dr \right)^2 \\
&\leq \sum_{i=1}^n \left( \int_{r=0}^\eta \|u'_{si} - u'(\mathbf{W}_{(s,r)})_i\| \times \|L'_m(s)\|\, dr \right)^2 \\
&\leq \sum_{i=1}^n \left( \int_{r=0}^\eta \frac{\eta M \sqrt{n}}{d^{3/2}} \|\mathbf{y} - \mathbf{u}_s\| \times \frac{\sqrt{n}}{\sqrt{d}} \|\mathbf{y} - \mathbf{u}_s\|\, dr \right)^2 \\
&= \frac{\eta^4 M^2 n^3}{d^4} \|\mathbf{y} - \mathbf{u}_s\|^4 \\
&= \left( \frac{\eta^2 M n^{3/2}}{d^2} \|\mathbf{y} - \mathbf{u}_s\|^2 \right)^2.
\end{aligned}
$$

As the upper bound depends quadratically on $\eta$, we can choose it small enough for gradient descent to converge, as we will show in the proof of Theorem 6.1 in the next subsection.

Recall that $\|\mathbf{y} - \mathbf{u}_{s+1}\|^2$ can be expressed as the sum of four terms:

$$
\|\mathbf{y} - \mathbf{u}_{s+1}\|^2 = \|\mathbf{y} - \mathbf{u}_s\|^2 - 2(\mathbf{y} - \mathbf{u}_s)^\top \mathbf{I}_1 - 2(\mathbf{y} - \mathbf{u}_s)^\top \mathbf{I}_2 + \|\mathbf{u}_{s+1} - \mathbf{u}_s\|^2. \tag{S.25}
$$

Thus far, we have bounded the second and third terms on the RHS of Equation (S.25):

$$
\begin{aligned}
-2(\mathbf{y} - \mathbf{u}_s)^\top \mathbf{I}_1 &\leq -2\eta\, \mathrm{eig}_{\min}(\widehat{\Theta}_m(s)) \|\mathbf{y} - \mathbf{u}_s\|^2, \\
-2(\mathbf{y} - \mathbf{u}_s)^\top \mathbf{I}_2 &\leq 2\|\mathbf{y} - \mathbf{u}_s\|\|\mathbf{I}_2\| \leq \left( \frac{2\eta^2 M n^{3/2}}{d^2} \|\mathbf{y} - \mathbf{u}_s\|^3 \right).
\end{aligned}
$$

These bounds lead to the first three terms in the claimed upper bound of Equation (S.24). It remains to get an appropriate upper bound of the fourth term on the RHS of Equation (S.25).

Using the bound on the derivative of the loss in Lemma H.1, we complete the proof:

$$
\begin{aligned}
\|\mathbf{u}_{s+1} - \mathbf{u}_s\|^2 &= \sum_{i=1}^{n} (u_{(s+1)i} - u_{si})^2 \\
&= \sum_{i=1}^{n} \left( \sum_{j=1}^{m} \sqrt{\lambda_{m,j}} a_j \left( \sigma \left( \frac{\mathbf{w}_{(s+1)j}^{\top} \mathbf{x}_i}{\sqrt{d}} \right) - \sigma \left( \frac{\mathbf{w}_{sj}^{\top} \mathbf{x}_i}{\sqrt{d}} \right) \right) \right)^2 \\
&\leq \sum_{i=1}^{n} \left( \sum_{j=1}^{m} \sqrt{\lambda_{m,j}} a_j \left| \sigma \left( \frac{\mathbf{w}_{(s+1)j}^{\top} \mathbf{x}_i}{\sqrt{d}} \right) - \sigma \left( \frac{\mathbf{w}_{sj}^{\top} \mathbf{x}_i}{\sqrt{d}} \right) \right| \right)^2 \\
&\leq \sum_{i=1}^{n} \left( \sum_{j=1}^{m} \sqrt{\lambda_{m,j}} a_j \left| \frac{\mathbf{w}_{(s+1)j}^{\top} \mathbf{x}_i}{\sqrt{d}} - \frac{\mathbf{w}_{sj}^{\top} \mathbf{x}_i}{\sqrt{d}} \right| \right)^2 \\
&\leq \sum_{i=1}^{n} \left( \sum_{j=1}^{m} \frac{\sqrt{\lambda_{m,j}} a_j}{\sqrt{d}} \|\mathbf{w}_{(s+1)j} - \mathbf{w}_{sj}\| \|\mathbf{x}_i\| \right)^2 \\
&\leq \left( \sum_{i=1}^{n} \|\mathbf{x}_i\|^2 \right) \times \left( \sum_{j=1}^{m} \frac{\sqrt{\lambda_{m,j}} a_j}{\sqrt{d}} \times \|\mathbf{w}_{(s+1)j} - \mathbf{w}_{sj}\| \right)^2 \\
&\leq n \times \left( \sum_{j=1}^{m} \frac{\sqrt{\lambda_{m,j}} a_j}{\sqrt{d}} \times \left\| \eta \frac{\partial L_m(\mathbf{W}_s)}{\partial \mathbf{w}_{sj}} \right\| \right)^2 \\
&\leq n \times \left( \sum_{j=1}^{m} \frac{\sqrt{\lambda_{m,j}} a_j}{\sqrt{d}} \times \frac{\eta \sqrt{\lambda_{m,j}} n}{\sqrt{d}} \|\mathbf{y} - \mathbf{u}_s\| \right)^2 \\
&\leq \frac{\eta^2 n^2}{d^2} \|\mathbf{y} - \mathbf{u}_s\|^2 \left( \sum_{j=1}^{m} \lambda_{m,j} \right)^2 \\
&= \frac{\eta^2 n^2}{d^2} \|\mathbf{y} - \mathbf{u}_s\|^2.
\end{aligned}
$$

$\square$

## H.3  Proof of Theorem 6.1

Using the lemmas we have just shown, we will prove global convergence of gradient descent. Recall the assumed bound $C$ on $|y_i|$ for every $i \geq 1$ in Assumption 3.1, and also

$$
C_1 = \sup_{c \in (0,1]} \mathbb{E}[\sigma(cz)^2]
$$

where the expectation is taken over the real-valued random variable $z$ distributed as $\mathcal{N}(0, 1/d)$. As shown in Appendix G, $C_1$ is finite.

By the argument in Appendix G again, there exists an event $E_1$ such that $E_1$ happens with probability at least $1 - (\delta/2)$ and conditioned on $E_1$, we have

$$
\|\mathbf{y} - \mathbf{u}_0\| < \sqrt{n(C^2 + C_1)\frac{2}{\delta}}. \tag{S.26}
$$

Meanwhile, by Proposition D.1, there is an event $E_2$ such that $E_2$ happens with probability at least $1 - (\delta/2)$ and conditioned on $E_2$, we have

$$
\text{eig}_{\min}(\widehat{\Theta}_m(0)) > \frac{\gamma \kappa_n}{2}. \tag{S.27}
$$

Let $E_3$ be the event that is the conjunction of $E_1$ and $E_2$. This event happens with probability at least $1 - \delta$, and under this event, Equations (S.26) and (S.27) both hold.

Condition on $E_3$. We prove the inequality in Equation (16) by induction on $s$. The base case of $s = 0$ is immediate. To prove the inductive case, assume that $s \geq 1$, and that the inequality in Equation (16) holds for all $s' = 0, 1, \ldots, s - 1$.

Let $\alpha = \eta\gamma\kappa_n/2$ and $\beta = (1 - \alpha)^{1/2}$ and

$$c_{m,j} = \frac{\eta n}{1 - \beta}\sqrt{\frac{8\lambda_{m,j}(C^2 + C_1)}{\delta d}}.$$

Then,

$$\sum_{j=1}^{m} c_{m,j}^2 = \left(\frac{\eta^2 n^2}{(1 - \beta)^2}\frac{8(C^2 + C_1)}{\delta d}\sum_{j=1}^{m}\lambda_{m,j}\right) = \left(\frac{\eta^2 n^2}{(1 - \beta)^2}\frac{8(C^2 + C_1)}{\delta d}\right).$$

Note that for all $j \in [m]$,

$$\begin{aligned}
\|\mathbf{w}_{sj} - \mathbf{w}_{0j}\| &\leq \sum_{s'=0}^{s-1}\|\mathbf{w}_{(s'+1)j} - \mathbf{w}_{s'j}\| \\
&\leq \sum_{s'=0}^{s-1}\eta\left\|\frac{\partial L_m(\mathbf{W}_{s'})}{\partial\mathbf{w}_{s'j}}\right\| \\
&\leq \sum_{s'=0}^{s-1}\eta\sqrt{\frac{\lambda_{m,j}n}{d}}\|\mathbf{y} - \mathbf{u}_{s'}\| \\
&\leq \eta\sqrt{\frac{\lambda_{m,j}n}{d}}\sum_{s'=0}^{s-1}(1 - \alpha)^{s'/2}\|\mathbf{y} - \mathbf{u}_0\| \\
&\leq \frac{\eta}{1 - \beta}\sqrt{\frac{\lambda_{m,j}n}{d}}\|\mathbf{y} - \mathbf{u}_0\| \\
&\leq \frac{\eta}{1 - \beta}\sqrt{\frac{\lambda_{m,j}n}{d}}\sqrt{n(C^2 + C_1)\frac{2}{\delta}} \\
&= \frac{1}{2}\times\frac{\eta n}{1 - \beta}\sqrt{\frac{8\lambda_{m,j}(C^2 + C_1)}{\delta d}} = \frac{c_{m,j}}{2}
\end{aligned}$$

where the third inequality uses the bound shown in Lemma H.1, the fourth inequality follows from the induction hypothesis, and the sixth inequality uses the bound in (S.26). Thus, by Lemma E.4 with $c_{m,j}$ from above and the lower bound on the minimum eigenvalue in Equation (S.27), we have

$$\begin{aligned}
&\mathrm{eig}_{\min}(\widehat{\Theta}_m(s)) \\
&\geq \mathrm{eig}_{\min}(\widehat{\Theta}_m^{(1)}(\mathbf{X}; \mathbf{W}_0)) - \left(\frac{nM^2\gamma}{4d^2m}\sum_{j=1}^{m}c_{m,j}^2 + \frac{nM\gamma}{d^{3/2}m^{1/2}}\sqrt{\sum_{j=1}^{m}c_{m,j}^2}\right) \\
&= \frac{\gamma\kappa_n}{2} - \left(\frac{nM^2\gamma}{4d^2m}\left(\frac{\eta^2 n^2}{(1 - \beta)^2}\frac{8(C^2 + C_1)}{\delta d}\right) + \frac{nM\gamma}{d^{3/2}m^{1/2}}\sqrt{\frac{\eta^2 n^2}{(1 - \beta)^2}\frac{8(C^2 + C_1)}{\delta d}}\right) \\
&= \frac{\gamma\kappa_n}{2} - \left(\frac{2\eta^2 n^3 M^2\gamma(C^2 + C_1)}{d^3m(1 - \beta)^2\delta} + \frac{\sqrt{8}\eta n^2 M\gamma(C^2 + C_1)^{1/2}}{d^2m^{1/2}(1 - \beta)\delta^{1/2}}\right).
\end{aligned}$$

Meanwhile, by Lemma H.2, the induction hypothesis, and Equation (S.26),

$$\|\mathbf{y} - \mathbf{u}_{s+1}\|^2$$

$$\leq \left(1 - 2\eta \operatorname{eig}_{\min}(\widehat{\Theta}_m(s)) + \frac{2\eta^2 M n^{3/2}}{d^2}\|\mathbf{y} - \mathbf{u}_s\| + \frac{\eta^2 n^2}{d^2}\right)\|\mathbf{y} - \mathbf{u}_s\|^2$$

$$\leq \left(1 - 2\eta \operatorname{eig}_{\min}(\widehat{\Theta}_m(s)) + \frac{2\eta^2 M n^{3/2}}{d^2}(1-\alpha)^{s/2}\|\mathbf{y} - \mathbf{u}_0\| + \frac{\eta^2 n^2}{d^2}\right)\|\mathbf{y} - \mathbf{u}_s\|^2$$

$$\leq \left(1 - 2\eta \operatorname{eig}_{\min}(\widehat{\Theta}_m(s)) + \frac{2\eta^2 M n^{3/2}}{d^2}(1-\alpha)^{s/2}\sqrt{n(C^2 + C_1)\frac{2}{\delta}} + \frac{\eta^2 n^2}{d^2}\right)\|\mathbf{y} - \mathbf{u}_s\|^2.$$

Thus, we can complete the proof of this inductive case if we show that

$$\left(2\eta \operatorname{eig}_{\min}(\widehat{\Theta}_m(s)) - \frac{2\eta^2 M n^{3/2}}{d^2}(1-\alpha)^{s/2}\sqrt{n(C^2 + C_1)\frac{2}{\delta}} - \frac{\eta^2 n^2}{d^2}\right) \geq \frac{\eta\gamma\kappa_n}{2}$$

which is equivalent to

$$\operatorname{eig}_{\min}(\widehat{\Theta}_m(s)) \geq \left(\frac{\eta M n^{3/2}}{d^2}(1-\alpha)^{s/2}\sqrt{n(C^2 + C_1)\frac{2}{\delta}} + \frac{\eta n^2}{2d^2} + \frac{\gamma\kappa_n}{4}\right).$$

We will show this sufficient condition by proving the following stronger inequality (stronger because of the lower bound on $\operatorname{eig}_{\min}(\widehat{\Theta}_m(s))$ that we have derived above):

$$\frac{\gamma\kappa_n}{2} - \left(\frac{2\eta^2 n^3 M^2 \gamma(C^2 + C_1)}{d^3 m(1-\beta)^2\delta} + \frac{\sqrt{8}\eta n^2 M\gamma(C^2 + C_1)^{1/2}}{d^2 m^{1/2}(1-\beta)\delta^{1/2}}\right)$$

$$\geq \left(\frac{\eta M n^{3/2}}{d^2}(1-\alpha)^{s/2}\sqrt{n(C^2 + C_1)\frac{2}{\delta}} + \frac{\eta n^2}{2d^2} + \frac{\gamma\kappa_n}{4}\right),$$

which is equivalent to

$$\frac{\gamma\kappa_n}{4} \geq \left(\frac{2\eta^2 n^3 M^2 \gamma(C^2 + C_1)}{d^3 m(1-\beta)^2\delta} + \frac{\sqrt{8}\eta n^2 M\gamma(C^2 + C_1)^{1/2}}{d^2 m^{1/2}(1-\beta)\delta^{1/2}}\right.$$

$$\left. + \frac{\eta M n^{3/2}}{d^2}(1-\alpha)^{s/2}\sqrt{n(C^2 + C_1)\frac{2}{\delta}} + \frac{\eta n^2}{2d^2}\right).$$

But the four summands on the RHS of the above inequality are at most $\gamma\kappa_n/16$ by the assumed upper bound on $\eta$, the assumed lower bound on $m$, and the fact that $(1-\alpha) \leq 1$. Thus, the inequality from above holds, as desired.

# I Proofs of the results of Section 7 on feature learning (smooth case)

## I.1 Proofs of Section 7.2 (linear activation)

### I.1.1 Proof of Theorem 7.4

Consider a linear activation $\sigma(z) = z$. The model is therefore defined as

$$f_m(\mathbf{x}; \mathbf{W}) = \frac{1}{\sqrt{d}}\sum_{j=1}^{m}\sqrt{\lambda_{m,j}}a_j\mathbf{w}_j^\top\mathbf{x}.$$

The objective function in Equation (5) can be written as

$$L_m(\mathbf{W}) = \frac{1}{2}\|\mathbf{y} - \mathbf{A}\mathbf{W}\|^2 \tag{S.28}$$

where $\mathbf{y} = (y_1, \ldots, y_n)^\top$ and $\mathbf{A}$ is the $n \times md$ matrix defined by

$$\mathbf{A} = \frac{1}{\sqrt{d}} \begin{pmatrix} \sqrt{\lambda_{m,1}} a_1 \mathbf{x}_1^\top & \cdots & \sqrt{\lambda_{m,m}} a_m \mathbf{x}_1^\top \\ \vdots & & \vdots \\ \sqrt{\lambda_{m,1}} a_1 \mathbf{x}_n^\top & \cdots & \sqrt{\lambda_{m,m}} a_m \mathbf{x}_n^\top \end{pmatrix} = \frac{1}{\sqrt{d}} (\mathbf{B} \otimes \mathbf{X}),$$

where $\otimes$ denotes the Kronecker product and $\mathbf{B} = (\sqrt{\lambda_{m,1}} a_1 \ldots \sqrt{\lambda_{m,m}} a_m) \in \mathbb{R}^{1 \times m}$. We sometimes view $\mathbf{B}$ as a row vector and write $\mathbf{B}^\top$ to mean the corresponding $m$-dimensional (column) vector. Let

$$\mathbf{X} = \mathbf{U}\mathbf{D}\mathbf{V}^\top$$

be a reduced SVD of the data matrix $\mathbf{X}$, where $\mathbf{U}$ is a $n \times k$ matrix with orthonormal columns, $\mathbf{D}$ is a diagonal $k \times k$ matrix, $\mathbf{V}$ is a $d \times k$ matrix with orthonormal columns, and $k \leq \min(n, d)$ is the rank of $\mathbf{X}$. Define

$$\mathbf{V}' = \frac{1}{\sqrt{\sum_{j=1}^m \lambda_{m,j}}} (\mathbf{B}^\top \otimes \mathbf{V}) \in \mathbb{R}^{md \times k}.$$

Note that $\mathbf{V}'$ has orthonormal columns as

$$(\mathbf{V}')^\top \mathbf{V}' = \frac{1}{\sum_{j=1}^m \lambda_{m,j}} \sum_{j=1}^m \lambda_{m,j} a_j^2 (\mathbf{V}^\top \mathbf{V}) = I_k.$$

Therefore,

$$\mathbf{A} = \mathbf{U} \left( \frac{\sqrt{\sum_{j=1}^m \lambda_{m,j}}}{\sqrt{d}} \mathbf{D} \right) (\mathbf{V}')^\top$$

is the reduced SVD of $\mathbf{A}$, and

$$\mathbf{A}\mathbf{A}^\top = \frac{\sum_{j=1}^m \lambda_{m,j}}{d} \mathbf{X}\mathbf{X}^\top = \frac{\sum_{j=1}^m \lambda_{m,j}}{d} \mathbf{U}\mathbf{D}^2\mathbf{U}^\top.$$

If $k < md$, let $\mathbf{V}'_\perp$ be a matrix in $\mathbb{R}^{md \times (md-k)}$ that makes the $md \times md$ matrix $(\mathbf{V}', \mathbf{V}'_\perp)$ orthonormal; otherwise, let $\mathbf{V}'_\perp$ be the $md$ dimensional zero vector.

The solution of Equation (S.28) under gradient flow or gradient descent with the initialisation $\mathbf{W}_0$ is given by

$$\mathbf{W}_\infty = \mathbf{A}^\dagger \mathbf{y} + \mathbf{V}'_\perp (\mathbf{V}'_\perp)^\top \mathbf{W}_0 = \frac{\sqrt{d}}{\sqrt{\sum_j \lambda_{m,j}}} \mathbf{V}'\mathbf{D}^{-1}\mathbf{U}^\top \mathbf{y} + \mathbf{V}'_\perp (\mathbf{V}'_\perp)^\top \mathbf{W}_0$$

where $(-)^\dagger$ is the Moore-Penrose inverse operator. Also,

$$\mathbf{W}_0 = \mathbf{V}'(\mathbf{V}')^\top \mathbf{W}_0 + \mathbf{V}'_\perp (\mathbf{V}'_\perp)^\top \mathbf{W}_0.$$

From these facts, we can derive a formula that describes the changes in weights during the training based on gradient flow or gradient descent:

$$\mathbf{W}_\infty - \mathbf{W}_0 = \frac{\sqrt{d}}{\sqrt{\sum_j \lambda_{m,j}}} \left( \mathbf{V}'\mathbf{D}^{-1}\mathbf{U}^\top \mathbf{y} \right) + (\mathbf{V}'_\perp (\mathbf{V}'_\perp)^\top \mathbf{W}_0) - \mathbf{W}_0$$

$$= \left( \mathbf{B}^\top \otimes \frac{\sqrt{d}}{\sum_j \lambda_{m,j}} \left( \mathbf{V}\mathbf{D}^{-1}\mathbf{U}^\top \mathbf{y} \right) \right) - (\mathbf{V}'(\mathbf{V}')^\top \mathbf{W}_0)$$

$$= \frac{1}{\sum_j \lambda_{m,j}} \left( \mathbf{B}^\top \otimes \left( \boldsymbol{\beta}_\infty - \mathbf{V}\mathbf{V}^\top \boldsymbol{\beta}_0 \right) \right)$$

where $\boldsymbol{\beta}_0 = \sum_{j=1}^m \sqrt{\lambda_{m,j}} a_j \mathbf{w}_{0j}$ and

$$\boldsymbol{\beta}_\infty = \sqrt{d}\mathbf{X}^\dagger\mathbf{y} = \sqrt{d}\left(\mathbf{V}\mathbf{D}^{-1}\mathbf{U}^\top\mathbf{y}\right)$$

is the minimum-norm minimiser of $\frac{1}{2}\|\mathbf{y} - \frac{1}{\sqrt{d}}\mathbf{X}\boldsymbol{\beta}\|^2$. It follows that

$$\mathbf{w}_{\infty j} - \mathbf{w}_{0j} = \frac{\sqrt{\lambda_{m,j}}}{\sum_k \lambda_{m,k}} a_j (\boldsymbol{\beta}_\infty - \mathbf{V}\mathbf{V}^\top\boldsymbol{\beta}_0).$$

### I.1.2 Proof of Theorem 7.5

First note that, under the scaling (1),

$$\sum_{k=1}^m \lambda_{m,k} \left(\sigma(Z_k(\mathbf{x}; \mathbf{W}_0))\right)^2 \to \frac{\gamma}{d}\mathbb{E}\left[\left(\mathbf{x}^\top\mathbf{w}_{01}\right)^2\right] + \frac{(1-\gamma)}{d}\sum_{k=1}^\infty \widetilde{\lambda}_k \left(\mathbf{x}^\top\mathbf{w}_{0k}\right)^2$$

almost surely as $m \to \infty$; hence the denominator in (17) is of order 1. Similarly, under the mean-field scaling,

$$m \times \sum_{k=1}^m \frac{1}{m^2} \left(\sigma(Z_k(\mathbf{x}; \mathbf{W}_0))\right)^2 \to \frac{1}{d}\mathbb{E}\left[\left(\mathbf{x}^\top\mathbf{w}_{01}\right)^2\right]$$

almost surely as $m \to \infty$; hence the denominator in (17) is of order $1/m$. For the numerator, from Equation (19), we have

$$\sum_{j=1}^m \lambda_{m,j} \left(\sigma(Z_j(\mathbf{x}; \mathbf{W}_\infty)) - \sigma(Z_j(\mathbf{x}; \mathbf{W}_0))\right)^2 = \frac{1}{d}\frac{\sum_{j=1}^m \lambda_{m,j}^2}{(\sum_{k=1}^m \lambda_{m,k})^2}\left(\mathbf{x}^\top(\boldsymbol{\beta}_\infty - \mathbf{V}\mathbf{V}^\top\boldsymbol{\beta}_0)\right)^2.$$

Under the scaling (1), $\frac{\sum_{j=1}^m \lambda_{m,j}^2}{(\sum_k \lambda_{m,k})^2} = \sum_{j=1}^m \lambda_{m,j}^2 \to (1-\gamma)^2 \sum_{j\geq 1} \widetilde{\lambda}_j^2$. Hence feature learning occurs if and only if $\gamma < 1$. Under the mean-field scaling, $\frac{\sum_{j=1}^m \lambda_{m,j}^2}{(\sum_k \lambda_{m,k})^2} = 1/m$. Hence feature learning occurs. Additionally, as the $(\lambda_{m,j})_{j\geq 1}$ are ordered, we have

$$\max_{j=1,\dots,m} \lambda_{m,j} \left(\sigma(Z_j(\mathbf{x}; \mathbf{W}_\infty)) - \sigma(Z_j(\mathbf{x}; \mathbf{W}_0))\right)^2 = \max_{j=1,\dots,m} \frac{1}{d}\frac{\lambda_{m,j}^2}{(\sum_k \lambda_{m,k})^2}\left(\mathbf{x}^\top(\boldsymbol{\beta}_\infty - \mathbf{V}\mathbf{V}^\top\boldsymbol{\beta}_0)\right)^2$$

$$= \frac{\lambda_{m,1}^2}{d(\sum_k \lambda_{m,k})^2}\left(\mathbf{x}^\top(\boldsymbol{\beta}_\infty - \mathbf{V}\mathbf{V}^\top\boldsymbol{\beta}_0)\right)^2.$$

Under the scaling (1), $\lambda_{m,1}^2 \to (1-\gamma)\widetilde{\lambda}_1$, with $\widetilde{\lambda}_1 > 0$, hence non-uniform feature learning occurs if and only if $\gamma < 1$. Under mean-field scaling, $\lambda_{m,1}^2/(\sum_k \lambda_{m,k})^2 = 1/m^2 = o(1/m)$, hence non-uniform feature learning does not occur. Additionally, by Equation (19) in Theorem 7.4, we have

$$\sqrt{\lambda_{m,j}} a_j \mathbf{w}_{\infty j} = \sqrt{\lambda_{m,j}} a_j \mathbf{w}_{0j} + \frac{\lambda_{m,j}}{\sum_k \lambda_{m,k}}(\boldsymbol{\beta}_\infty - \mathbf{V}\mathbf{V}^\top\boldsymbol{\beta}_0)$$

$$= \frac{\lambda_{m,j}}{\sum_k \lambda_{m,k}}\left(\boldsymbol{\beta}_\infty - \mathbf{V}\mathbf{V}^\top\sum_{k\neq j} \sqrt{\lambda_{m,k}} a_k \mathbf{w}_{0k}\right) + \sqrt{\lambda_{m,j}}\left(I_d - \frac{\lambda_{m,j}}{\sum_k \lambda_{m,k}}\lambda_{m,j}\mathbf{V}\mathbf{V}^\top\right) a_j \mathbf{w}_{0j}.$$

The right-hand side is the sum of two independent Gaussian random vectors, and is therefore a Gaussian random vector, with mean $\frac{\lambda_{m,j}}{\sum_k \lambda_{m,k}}\boldsymbol{\beta}_\infty$ and covariance matrix $\frac{\lambda_{m,j}^2}{(\sum_k \lambda_{m,k})^2}(\sum_k \lambda_{m,k} - \lambda_{m,j})(\mathbf{V}\mathbf{V}^\top)^2 + \lambda_{m,j}\left(I_d - \frac{\lambda_{m,j}}{\sum_k \lambda_{m,k}}\mathbf{V}\mathbf{V}^\top\right)^2 = \lambda_{m,j}(I_d - \frac{\lambda_{m,j}}{\sum_k \lambda_{m,k}}\mathbf{V}\mathbf{V}^\top)$. The distributional convergence in Equation (21) then follows from Slutsky's theorem.

### I.1.3  Proof of Proposition 7.6

Using Markov and Cauchy-Schwarz inequalities,

$$\Pr\left(\left|\widetilde{f}_{m,\rho}(\mathbf{x};\mathbf{W}_\infty) - f_m(\mathbf{x};\mathbf{W}_\infty)\right| > \varepsilon\right) \le \frac{\|\mathbf{x}\|}{\varepsilon\sqrt{d}} \times \mathbb{E}\left[\left\|\sum_{j>\lfloor\rho m\rfloor} \sqrt{\lambda_{m,j}}\, a_j \mathbf{w}_{\infty j}\right\|\right].$$

Meanwhile, we have

$$\left\|\sum_{j>\lfloor\rho m\rfloor} \sqrt{\lambda_{m,j}}\, a_j \mathbf{w}_{\infty j}\right\|$$

$$\le \left\|\sum_{j>\lfloor\rho m\rfloor} \lambda_{m,j}\left(\boldsymbol{\beta}_\infty - \mathbf{V}\mathbf{V}^\top \sum_{k\ne j} \sqrt{\lambda_{m,k}}\, a_k \mathbf{w}_{0k}\right)\right\| + \left\|\sum_{j>\lfloor\rho m\rfloor} \sqrt{\lambda_{m,j}}\left(I_d - \lambda_{m,j}\mathbf{V}\mathbf{V}^\top\right) a_j \mathbf{w}_{0j}\right\|$$

$$\le \sum_{j>\lfloor\rho m\rfloor} \lambda_{m,j}\left(\|\boldsymbol{\beta}_\infty\| + \left\|\mathbf{V}\mathbf{V}^\top \sum_{k\ne j} \sqrt{\lambda_{m,k}}\, a_k \mathbf{w}_{0k}\right\|\right) + \left\|\sum_{j>\lfloor\rho m\rfloor} \sqrt{\lambda_{m,j}}\left(I_d - \lambda_{m,j}\mathbf{V}\mathbf{V}^\top\right) a_j \mathbf{w}_{0j}\right\|.$$

Also,

$$\mathbb{E}\left[\left\|\mathbf{V}\mathbf{V}^\top \sum_{k\ne j} \sqrt{\lambda_{m,k}}\, a_k \mathbf{w}_{0k}\right\|\right] \le \sqrt{\mathbb{E}\left[\left\|\mathbf{V}\mathbf{V}^\top \sum_{k\ne j} \sqrt{\lambda_{m,k}}\, a_k \mathbf{w}_{0k}\right\|^2\right]}$$

$$= \sqrt{(1 - \lambda_{m,j})\operatorname{trace}(\mathbf{V}\mathbf{V}^\top)}$$

$$\le \sqrt{d},$$

and

$$\mathbb{E}\left[\left\|\sum_{j>\lfloor\rho m\rfloor} \sqrt{\lambda_{m,j}}\left(I_d - \lambda_{m,j}\mathbf{V}\mathbf{V}^\top\right) a_j \mathbf{w}_{0j}\right\|\right] \le \sqrt{\mathbb{E}\left[\left\|\sum_{j>\lfloor\rho m\rfloor} \sqrt{\lambda_{m,j}}\left(I_d - \lambda_{m,j}\mathbf{V}\mathbf{V}^\top\right) a_j \mathbf{w}_{0j}\right\|^2\right]}$$

$$= \sqrt{\sum_{j>\lfloor\rho m\rfloor} \lambda_{m,j}\operatorname{trace}\left(\left(I_d - \lambda_{m,j}\mathbf{V}\mathbf{V}^\top\right)^2\right)}$$

$$= \sqrt{\sum_{j>\lfloor\rho m\rfloor} \lambda_{m,j}\operatorname{trace}\left(I_d - 2\lambda_{m,j}\mathbf{V}\mathbf{V}^\top + \lambda_{m,j}^2\mathbf{V}\mathbf{V}^\top\right)}$$

$$\le \sqrt{\sum_{j>\lfloor\rho m\rfloor} \lambda_{m,j} \times d \times (1 - \lambda_{m,j})^2}$$

$$\le \sqrt{d\sum_{j>\lfloor\rho m\rfloor} \lambda_{m,j}}.$$

By combining the above inequalities, we obtain the desired result:

$$\Pr\left(\left|\widetilde{f}_{m,\rho}(\mathbf{x};\mathbf{W}_\infty) - f_m(\mathbf{x};\mathbf{W}_\infty)\right| > \varepsilon\right) \le \frac{\|\mathbf{x}\|}{\varepsilon\sqrt{d}}\left(\left(\|\boldsymbol{\beta}_\infty\| + \sqrt{d}\right)\left(\sum_{j>\lfloor\rho m\rfloor} \lambda_{m,j}\right) + \sqrt{d\sum_{j>\lfloor\rho m\rfloor} \lambda_{m,j}}\right).$$

### I.2 Proofs of Section 7.3 (nonlinear activation)

### I.2.1 Proof of Theorem 7.9

Our proof of Theorem 7.9 relies on the following observation on the linear combinations of continuous independent real-valued random variables.

**Lemma I.1.** *Let $z_1, \dots, z_n$ be independent continuous real-valued random variables. Let $\mathcal{B} \subset \mathbb{R}$ be a finite subset of the real numbers such that $\mathcal{B} \neq \{0\}$. Then, almost surely,*

$$\min_{\mathbf{b} \in \mathcal{B}^n \setminus \{0, \dots, 0\}} \left| \sum_{i=1}^n b_i z_i \right| > 0.$$

*Proof.* Denote $\mathcal{S} = \mathcal{B}^n \setminus \{0, \dots, 0\}$. For any $\mathbf{b} = (b_1, \dots, b_n) \in \mathcal{S}$, $\sum_{i=1}^n b_i z_i$, so that $\Pr(\sum_{i=1}^n b_i z_i = 0) = 0$. Hence, since $\mathcal{S}$ is finite,

$$\Pr\left( \min_{\mathbf{b} \in \mathcal{S}} \left| \sum_{i=1}^n b_i z_i \right| = 0 \right) = \Pr\left( \bigcup_{\mathbf{b} \in \mathcal{S}} \left\{ \sum_{i=1}^n b_i z_i = 0 \right\} \right)$$

$$\leq \sum_{\mathbf{b} \in \mathcal{S}} \Pr\left( \sum_{i=1}^n b_i z_i = 0 \right)$$

$$= 0.$$

$\square$

The proof also uses the our globally-made standard assumption that for every random variable $Z \sim \mathcal{N}(0, s^2)$ for some $s > 0$, the expectation $\mathbb{E}[\sigma(Z)^2]$ is finite and greater than 0.

**Proof of Theorem 7.9.** Since non-uniform feature learning implies feature learning, we prove the former only. We start by showing that the denominator $\sum_{j=1}^m \lambda_{m,j}(\sigma(Z_j(\mathbf{x}_i; \mathbf{W}_0)))^2$ in the condition for non-uniform feature learning converges to a positive finite value almost surely as $m$ tends to $\infty$. To see this, note

$$\lim_{m \to \infty} \sum_{j=1}^m \lambda_{m,j}(\sigma(Z_j(\mathbf{x}_i; \mathbf{W}_0)))^2 = \lim_{m \to \infty} \sum_{j=1}^m \left( \gamma \cdot \frac{1}{m} + (1 - \gamma) \cdot \frac{\widetilde{\lambda}_j}{\sum_{j'=1}^m \widetilde{\lambda}_{j'}} \right) \sigma(Z_j(\mathbf{x}_i; \mathbf{W}_0))^2$$

$$= \left( \gamma \cdot \lim_{m \to \infty} \sum_{j=1}^m \frac{1}{m} \sigma(Z_j(\mathbf{x}_i; \mathbf{W}_0))^2 \right) + \left( (1 - \gamma) \cdot \frac{\lim_{m \to \infty} \sum_{j=1}^m \widetilde{\lambda}_j \sigma(Z_j(\mathbf{x}_i; \mathbf{W}_0))^2}{\lim_{m \to \infty} \sum_{j'=1}^m \widetilde{\lambda}_{j'}} \right)$$

$$= \gamma \cdot \mathbb{E}_{Z \sim \mathcal{N}(0, \|\mathbf{x}_i\|^2/d)} \left[ \sigma(Z)^2 \right] + (1 - \gamma) \cdot \sum_{j=1}^{\infty} \widetilde{\lambda}_j \sigma(Z_j(\mathbf{x}_i; \mathbf{W}_0))^2.$$

The expectation in the first summand is positive and finite by our globally-made assumption on the activation function $\sigma$. Also, the infinite sum in the second summand is positive almost surely because it is greater than $\widetilde{\lambda}_1 \sigma(Z_1(\mathbf{x}_i; \mathbf{W}_0))^2$ but $\widetilde{\lambda}_1 \sigma(Z_1(\mathbf{x}_i; \mathbf{W}_0))^2$ is almost surely positive; $\widetilde{\lambda}_1 > 0$ and $\sigma(Z_1(\mathbf{x}_i; \mathbf{W}_0))$ is almost surely non-zero due to the injectivity of $\sigma$ and the continuity of the random variable $Z_1(\mathbf{x}; \mathbf{W}_0)$. Furthermore, the sum is almost surely finite as well, because its expectation is $\mathbb{E}_{Z \sim \mathcal{N}(0, \|\mathbf{x}_i\|^2/d)}[\sigma(Z)^2]$ which is finite by our globally-made assumption on the activation function $\sigma$. Thus, the limit of the denominator is positive and finite almost surely.

Since the denominator in the condition of non-uniform feature learning converges to a positive finite value almost surely, the condition holds if

$$\liminf_{m \to \infty} \left( \max_{j \in [m]} \lambda_{m,j} \left( \sigma(Z_j(\mathbf{x}_i; \mathbf{W}_1)) - \sigma(Z_j(\mathbf{x}_i; \mathbf{W}_0)) \right)^2 \right) > 0 \quad \text{almost surely.} \tag{S.29}$$

Note that the limit here is not redundant since $\mathbf{W}_1$ depends on $m$. The new condition in Equation (S.29) can be simplified further. It holds whenever

$$\liminf_{m\to\infty} \left( Z_1(\mathbf{x}_i; \mathbf{W}_1) - Z_1(\mathbf{x}_i; \mathbf{W}_0) \right)^2 > 0. \tag{S.30}$$

To see this, note that by the assumption of the theorem and the inverse function theorem, $\sigma^{-1}$ is a well-defined continuous function and also that $Z_1(\mathbf{x}_i; \mathbf{W}_0)$ does not depend on $m$. As a result, the inequality in Equation (S.30) implies

$$\liminf_{m\to\infty} \left( \sigma(Z_1(\mathbf{x}_i; \mathbf{W}_1)) - \sigma(Z_1(\mathbf{x}_i; \mathbf{W}_0)) \right)^2 > 0, \tag{S.31}$$

because otherwise some subsequence of $(\sigma(Z_1(\mathbf{x}_i; \mathbf{W}_1)))_m$ would converge to $\sigma(Z_1(\mathbf{x}_i; \mathbf{W}_0))$ as $m$ tends to $\infty$, but then by the continuity of $\sigma^{-1}$, the corresponding subsequence of $(Z_1(\mathbf{x}_i; \mathbf{W}_1))_m$ would converge to $Z_1(\mathbf{x}_i; \mathbf{W}_0)$, which contradicts Equation (S.30). Now using Equation (S.31), the assumption $\gamma > 0$, and the fact that $\widetilde{\lambda}_1 > 0$, we can prove the condition in Equation (S.29) as follows:

$$\liminf_{m\to\infty} \max_{j\in[m]} \lambda_{m,j}\left(\sigma(Z_j(\mathbf{x}_i; \mathbf{W}_1)) - \sigma(Z_j(\mathbf{x}_i; \mathbf{W}_0))\right)^2 \geq \liminf_{m\to\infty} \lambda_{m,1}\left(\sigma(Z_1(\mathbf{x}_i; \mathbf{W}_1)) - \sigma(Z_1(\mathbf{x}_i; \mathbf{W}_0))\right)^2$$

$$\geq \liminf_{m\to\infty}(1-\gamma)\widetilde{\lambda}_1\left(\sigma(Z_1(\mathbf{x}_i; \mathbf{W}_1)) - \sigma(Z_1(\mathbf{x}_i; \mathbf{W}_0))\right)^2$$

$$> 0.$$

We now show that Equation (S.30) holds almost surely. Note that

$$\left( Z_1(\mathbf{x}_i; \mathbf{W}_1) - Z_1(\mathbf{x}_i; \mathbf{W}_0) \right)^2 = \left( \frac{\mathbf{w}_{11}^\top \mathbf{x}_i}{\sqrt{d}} - \frac{\mathbf{w}_{01}^\top \mathbf{x}_i}{\sqrt{d}} \right)^2$$

$$= \frac{1}{d}\left( \eta \left( \nabla_{\mathbf{w}_{tj}} L(\mathbf{W}_t)\big|_{t=0} \right)^\top \mathbf{x}_i \right)^2$$

$$= \frac{\eta^2}{d}\left( \sum_{i'=1}^{n} y_{i'}\sqrt{\lambda_{m,1}}a_1\sigma'\left(\frac{\mathbf{w}_{0j}^\top \mathbf{x}_{i'}}{\sqrt{d}}\right)\frac{\mathbf{x}_{i'}^\top \mathbf{x}_i}{\sqrt{d}} \right)^2$$

$$= \frac{\eta^2\lambda_{m,1}}{d^2}\left( \sum_{i'=1}^{n} y_{i'}\left(\sigma'\left(\frac{\mathbf{w}_{0j}^\top \mathbf{x}_{i'}}{\sqrt{d}}\right)\mathbf{x}_{i'}^\top \mathbf{x}_i\right) \right)^2$$

$$\geq \frac{\eta^2(1-\gamma)\widetilde{\lambda}_1}{d^2}\left( \sum_{i'=1}^{n} y_{i'}\left(\sigma'\left(\frac{\mathbf{w}_{0j}^\top \mathbf{x}_{i'}}{\sqrt{d}}\right)\mathbf{x}_{i'}^\top \mathbf{x}_i\right) \right)^2.$$

Since the lower bound from above does not depend on $m$, we have

$$\liminf_{m\to\infty} \left( Z_1(\mathbf{x}_i; \mathbf{W}_1) - Z_1(\mathbf{x}_i; \mathbf{W}_0) \right)^2 \geq \frac{\eta^2(1-\gamma)\widetilde{\lambda}_1}{d^2}\left( \sum_{i'=1}^{n} y_{i'}\left(\sigma'\left(\frac{\mathbf{w}_{0j}^\top \mathbf{x}_{i'}}{\sqrt{d}}\right)\mathbf{x}_{i'}^\top \mathbf{x}_i\right) \right)^2.$$

Since $\eta^2(1-\gamma)\widetilde{\lambda}_1/d^2$ is positive, this lower bound is positive almost surely whenever the summation inside the square is positive almost surely.

Conditioning on $\mathbf{w}_{0j}$ and noting that $\sigma'\left(\frac{\mathbf{w}_{0j}^\top \mathbf{x}_i}{\sqrt{d}}\right)\|\mathbf{x}_i\|^2 > 0$, we have by Lemma I.1 that almost surely

$$\sum_{i'=1}^{n} y_{i'}\left(\sigma'\left(\frac{\mathbf{w}_{0j}^\top \mathbf{x}_{i'}}{\sqrt{d}}\right)\mathbf{x}_{i'}^\top \mathbf{x}_i\right) > 0.$$

We may use this lemma since the $y_{i'}$'s are independent from $\mathbf{w}_{0j}$ and so their distributions are unaffected by the conditioning. Now note that this almost-sure positivity of the summation holds regardless of which value the conditioned $\mathbf{w}_{0j}$ takes. Thus, the summation is positive almost surely without the conditioning. This completes the proof.

### I.2.2 Proof of Theorem 7.10

The proof of the theorem uses the following lemma on quadratic combinations of continuous independent random variables.

**Lemma I.2.** *Let $z_1, \dots, z_n$ be continuous independent real-valued random variables. Let $B$ be an $n$-by-$n$ real-valued matrix such that $B_{ii} \neq 0$ for some $i \in [n]$. Then, almost surely,*

$$\left| \sum_{i=1}^{n} \sum_{i'=1}^{n} z_i z_{i'} B_{ii'} \right| > 0.$$

*Proof.* Let $i \in [n]$ such that $B_{ii} \neq 0$. Then, when viewed as a polynomial on $z_i$,

$$\sum_{i=1}^{n} \sum_{i'=1}^{n} z_i z_{i'} B_{ii'}$$

is a quadratic polynomial with a non-zero coefficient for the term $z_i^2$. As a result, the zero set of this polynomial on $z_i$ has measure zero with respect to Lebesgue measure, that is, the Lebesgue measure of the set

$$\left\{ z_i \ \middle| \ \sum_{i=1}^{n} \sum_{i'=1}^{n} z_i z_{i'} B_{ii'} = 0 \right\} \subseteq \mathbb{R}$$

is zero (because the zero set of any analytic function has zero Lebesgue measure). Furthermore, $z_i$ is a continuous random variable, and so we have

$$\mathbb{E}\left[ \mathbf{1}_{\left\{ \sum_{i=1}^{n} \sum_{i'=1}^{n} z_i z_{i'} B_{ii'} = 0 \right\}} \ \middle| \ \{ z_{i'} \mid i' \in [n], i' \neq i \} \right] = 0.$$

As a result,

$$\Pr\left( \sum_{i=1}^{n} \sum_{i'=1}^{n} z_i z_{i'} B_{ii'} = 0 \right) = \mathbb{E}\left[ \mathbf{1}_{\left\{ \sum_{i=1}^{n} \sum_{i'=1}^{n} z_i z_{i'} B_{ii'} = 0 \right\}} \right]$$

$$= \mathbb{E}\left[ \mathbb{E}\left[ \mathbf{1}_{\left\{ \sum_{i=1}^{n} \sum_{i'=1}^{n} z_i z_{i'} B_{ii'} = 0 \right\}} \ \middle| \ \{ z_{i'} \mid i' \in [n], i' \neq i \} \right] \right]$$

$$= \mathbb{E}[0] = 0.$$

This proves the claim of the lemma. $\qquad\square$

**Proof of Theorem 7.10.** We first compute a lower bound of the squared norm of the gradient, which does not depend on $m$.

$$\left\| \nabla_{\mathbf{w}_{tj}} L(\mathbf{W}_t) \big|_{t=0} \right\|^2 = \left\| \sum_{i=1}^{n} y_i \sqrt{\lambda_{m,j}} a_j \sigma'\left( \frac{\mathbf{w}_{0j}^\top \mathbf{x}_i}{\sqrt{d}} \right) \frac{\mathbf{x}_i}{\sqrt{d}} \right\|^2$$

$$= \frac{\lambda_{m,j}}{d} \left\| \sum_{i=1}^{n} y_i \sigma'\left( \frac{\mathbf{w}_{0j}^\top \mathbf{x}_i}{\sqrt{d}} \right) \mathbf{x}_i \right\|^2$$

$$\geq \frac{(1-\gamma)\widetilde{\lambda}_j}{d} \left\| \sum_{i=1}^{n} y_i \sigma'\left( \frac{\mathbf{w}_{0j}^\top \mathbf{x}_i}{\sqrt{d}} \right) \mathbf{x}_i \right\|^2$$

$$= \frac{(1-\gamma)\widetilde{\lambda}_j}{d} \left| \sum_{k=1}^{d} \sum_{i=1}^{n} \sum_{i'=1}^{n} y_i y_{i'} \sigma'\left( \frac{\mathbf{w}_{0j}^\top \mathbf{x}_i}{\sqrt{d}} \right) \sigma'\left( \frac{\mathbf{w}_{0j}^\top \mathbf{x}_{i'}}{\sqrt{d}} \right) x_{ik} x_{i'k} \right|$$

$$= \frac{(1-\gamma)\widetilde{\lambda}_j}{d} \left| \sum_{i=1}^{n} \sum_{i'=1}^{n} y_i y_{i'} \left( \mathbf{x}_i^\top \mathbf{x}_{i'} \sigma'\left( \frac{\mathbf{w}_{0j}^\top \mathbf{x}_i}{\sqrt{d}} \right) \sigma'\left( \frac{\mathbf{w}_{0j}^\top \mathbf{x}_{i'}}{\sqrt{d}} \right) \right) \right|.$$

Thus,

$$\liminf_{m\to\infty} \left\| \nabla_{\mathbf{w}_{tj}} L(\mathbf{W}_t) \big|_{t=0} \right\|^2 \geq \frac{(1-\gamma)\widetilde{\lambda}_j}{d} \left| \sum_{i=1}^n \sum_{i'=1}^n y_i y_{i'} \left( \mathbf{x}_i^\top \mathbf{x}_{i'} \sigma' \left( \frac{\mathbf{w}_{0j}^\top \mathbf{x}_i}{\sqrt{d}} \right) \sigma' \left( \frac{\mathbf{w}_{0j}^\top \mathbf{x}_{i'}}{\sqrt{d}} \right) \right) \right|.$$

But by assumption, $((1-\gamma)\widetilde{\lambda}_j)/d$ is positive. The other factor in the lower bound is also positive with probability one. To see this, note that the $y_i$'s in the factor are continuous independent random variables, independent also from $\mathbf{w}_{0j}$, and

$$\|\mathbf{x}_i\|^2 \sigma' \left( \frac{\mathbf{w}_{0j}^\top \mathbf{x}_i}{\sqrt{d}} \right)^2 > 0 \quad \text{for all } i \in [n],$$

due to Assumption 3.1 and the assumption that $\sigma' > 0$. As a result, conditioned on $\mathbf{w}_{0j}$, by Lemma I.2, the factor is positive almost surely with respect to the conditional distributions of the $y_i$'s, which are the same as the original unconditional distributions of them due to the independence of the $y_i$'s with respect to $\mathbf{w}_{0j}$. Since this positivity holds regardless of which value $\mathbf{w}_{0j}$ takes, it also holds without the conditioning on $\mathbf{w}_{0j}$. This completes the proof.

## J  Proofs of the results of Appendix A.2 on feature learning (ReLU case)

### J.1  Proof of Theorem A.2

Our proof relies on a few lemmas.

**Lemma J.1.** *Assume Assumption 7.7. If the activation function $\sigma$ is ReLU, we have*

$$\nabla_{\mathbf{w}_{tj}} L(\mathbf{W}_t) \big|_{t=0} = \left( -\sqrt{\frac{\lambda_{m,j}}{d}} a_j \sum_{i=1}^m y_i \sigma'(Z_j(\mathbf{x}_i; \mathbf{W}_0)) \mathbf{x}_i \right) = \left( -\sqrt{\frac{\lambda_{m,j}}{d}} a_j \sum_{i=1}^m \mathbf{1}_{\{\mathbf{w}_{0j}^\top \mathbf{x}_i \geq 0\}} y_i \mathbf{x}_i \right)$$

*Proof.* The lemma follows from a straightforward calculation using the fact that $f_m(\mathbf{x}; \mathbf{W}_0) = 0$ for all $\mathbf{x} \in \mathbb{R}^d$. □

**Lemma J.2.** *Assume Assumptions 3.1, 7.7 and 7.8. Then, we have that for all $m$, $j \in [m]$, and $i \in [n]$,*

$$\lambda_{m,j} \left( \sigma(Z_j(\mathbf{x}_i; \mathbf{W}_1)) - \sigma(Z_j(\mathbf{x}_i; \mathbf{W}_0)) \right)^2 \geq \mathbf{1}_{\{\mathbf{w}_{0j}^\top \mathbf{x}_i \geq 0\}} \cdot \min \left\{ \frac{\eta^2 c^2 (1-\gamma)^2 \widetilde{\lambda}_j^2}{d^2}, \frac{(1-\gamma)\widetilde{\lambda}_j (\mathbf{w}_{0j}^\top \mathbf{x}_i)^2}{d} \right\}$$

*where $c$ depends only on the inputs/outputs (in particular, not depending on $m$) and is almost surely strictly positive (almost surely, with respect to the input/output).*

*Proof.* Using Lemma J.1, we can compute

$$\begin{aligned}
&\sigma(Z_j(\mathbf{x}_i; \mathbf{W}_1)) - \sigma(Z_j(\mathbf{x}_i; \mathbf{W}_0)) \\
&\quad = \sigma(Z_j(\mathbf{x}_i; \mathbf{W}_1)) - \sigma(Z_j(\mathbf{x}_i; \mathbf{W}_0)) \\
&\quad = \sigma \left( \frac{1}{\sqrt{d}} \left( \mathbf{w}_{0j} + \eta \sqrt{\frac{\lambda_{m,j}}{d}} a_j \sum_{i'=1}^n \mathbf{1}_{\{\mathbf{w}_{0j}^\top \mathbf{x}_{i'} \geq 0\}} y_{i'} \mathbf{x}_{i'} \right)^\top \mathbf{x}_i \right) - \sigma \left( \frac{\mathbf{w}_{0j}^\top \mathbf{x}_i}{\sqrt{d}} \right) \\
&\quad = \sigma \left( \frac{\mathbf{w}_{0j}^\top \mathbf{x}_i}{\sqrt{d}} + \eta \frac{\sqrt{\lambda_{m,j}}}{d} a_j \sum_{i'=1}^n \left( \mathbf{1}_{\{\mathbf{w}_{0j}^\top \mathbf{x}_{i'} \geq 0\}} \mathbf{x}_{i'}^\top \mathbf{x}_i \right) y_{i'} \right) - \sigma \left( \frac{\mathbf{w}_{0j}^\top \mathbf{x}_i}{\sqrt{d}} \right).
\end{aligned}$$

Denoting $\delta_i = ((\eta\sqrt{\lambda_{m,j}}a_j)/d)\sum_{i'=1}^n (\mathbf{1}_{\{\mathbf{w}_{0j}^\top \mathbf{x}_{i'} \geq 0\}}\mathbf{x}_{i'}^\top \mathbf{x}_i)y_{i'}$, we have that

$$|\sigma(Z_j(\mathbf{x}_i; \mathbf{W}_1)) - \sigma(Z_j(\mathbf{x}_i; \mathbf{W}_0))| = \left|\sigma\left(\frac{\mathbf{w}_{0j}^\top \mathbf{x}_i}{\sqrt{d}} + \delta_i\right) - \sigma\left(\frac{\mathbf{w}_{0j}^\top \mathbf{x}_i}{\sqrt{d}}\right)\right|$$

$$\geq \mathbf{1}_{\{\mathbf{w}_{0j}^\top \mathbf{x}_i \geq 0\}}\left|\sigma\left(\frac{\mathbf{w}_{0j}^\top \mathbf{x}_i}{\sqrt{d}} + \delta_i\right) - \sigma\left(\frac{\mathbf{w}_{0j}^\top \mathbf{x}_i}{\sqrt{d}}\right)\right|$$

$$\geq \mathbf{1}_{\{\mathbf{w}_{0j}^\top \mathbf{x}_i \geq 0\}}\left(\mathbf{1}_{\{\mathbf{w}_{0j}^\top \mathbf{x}_i + \sqrt{d}\delta_i \geq 0\}}|\delta_i| + \mathbf{1}_{\{\mathbf{w}_{0j}^\top \mathbf{x}_i + \sqrt{d}\delta_i < 0\}}\left|\frac{\mathbf{w}_{0j}^\top \mathbf{x}_i}{\sqrt{d}}\right|\right).$$

We then get

$$\lambda_{m,j}\left(\sigma(Z_j(\mathbf{x}_i; \mathbf{W}_1)) - \sigma(Z_j(\mathbf{x}_i; \mathbf{W}_0))\right)^2 \geq \mathbf{1}_{\{\mathbf{w}_{0j}^\top \mathbf{x}_i \geq 0\}}\lambda_{m,j}\min\left\{\delta_i^2, \frac{(\mathbf{w}_{0j}^\top \mathbf{x}_i)^2}{d}.\right\}$$

Now, notice that

$$\mathbf{1}_{\{\mathbf{w}_{0j}^\top \mathbf{x}_i \geq 0\}}\delta_i^2 = \frac{\eta^2}{d^2}\lambda_{m,j}\mathbf{1}_{\{\mathbf{w}_{0j}^\top \mathbf{x}_i \geq 0\}}\left(\sum_{i'=1}^n \left(\mathbf{1}_{\{\mathbf{w}_{0j}^\top \mathbf{x}_{i'} \geq 0\}}\mathbf{x}_{i'}^\top \mathbf{x}_i\right)y_{i'}\right)^2$$

$$= \frac{\eta^2}{d^2}\lambda_{m,j}\mathbf{1}_{\{\mathbf{w}_{0j}^\top \mathbf{x}_i \geq 0\}}\left(\|\mathbf{x}_i\|^2 y_i + \sum_{i'\in\{1,...,n\}\setminus\{i\}}\left(\mathbf{1}_{\{\mathbf{w}_{0j}^\top \mathbf{x}_{i'} \geq 0\}}\mathbf{x}_{i'}^\top \mathbf{x}_i\right)y_{i'}\right)^2$$

$$\geq \frac{\eta^2}{d^2}\lambda_{m,j}\mathbf{1}_{\{\mathbf{w}_{0j}^\top \mathbf{x}_i \geq 0\}}\left(\min_{\mathbf{b}\in\mathcal{S}}\left|\sum_{i'=1}^n b_{i'}y_{i'}\right|\right)^2$$

where $\mathcal{S} = \mathcal{B}^n \setminus \{0,...,0\}$ and $\mathcal{B}$ is given by

$$\mathcal{B} = \left\{u \cdot v \mid u \in \{0,1\}, v \in \left\{\mathbf{x}_1^\top \mathbf{x}_i, \ldots, \mathbf{x}_n^\top \mathbf{x}_i\right\}\right\}$$

Note that the $y_i$'s are continuous and independent random variables by Assumption 7.8. Thus, by Lemma I.1, with probability one,

$$c = \min_{\mathbf{b}\in\mathcal{S}}\left|\sum_{i'=1}^n b_{i'}y_{i'}\right| > 0.$$

Putting everything together, we finally get

$$\lambda_{m,j}\left(\sigma(Z_j(\mathbf{x}_i; \mathbf{W}_1)) - \sigma(Z_j(\mathbf{x}_i; \mathbf{W}_0))\right)^2 \geq \mathbf{1}_{\{\mathbf{w}_{0j}^\top \mathbf{x}_i \geq 0\}} \cdot \min\left\{\frac{\eta^2 c^2 \lambda_{m,j}^2}{d^2}, \frac{\lambda_{m,j}(\mathbf{w}_{0j}^\top \mathbf{x}_i)^2}{d}\right\}$$

$$\geq \mathbf{1}_{\{\mathbf{w}_{0j}^\top \mathbf{x}_i \geq 0\}} \cdot \min\left\{\frac{\eta^2 c^2 (1-\gamma)^2 \widetilde{\lambda}_j^2}{d^2}, \frac{(1-\gamma)\widetilde{\lambda}_j(\mathbf{w}_{0j}^\top \mathbf{x}_i)^2}{d}\right\},$$

which concludes the proof. $\qquad\square$

**Proof of Theorem A.2.** Note that the condition for non-uniform feature learning in Equation (S.2) implies that for feature learning in Equation (S.1). Thus, we will prove only the former condition.

By our setup, we have that $\lambda_{m,1} \geq \ldots \geq \lambda_{m,k} \geq (1-\gamma)\widetilde{\lambda}_k > 0$ for all $m$. Also, by Lemma J.2, we have that for $1 \leq j \leq k$,

$$\lambda_{m,j}\left(\sigma(Z_j(\mathbf{x}_i; \mathbf{W}_1)) - \sigma(Z_j(\mathbf{x}_i; \mathbf{W}_0))\right)^2 \geq \mathbf{1}_{\{\mathbf{w}_{0j}^\top \mathbf{x}_i \geq 0\}} \cdot \min\left\{\frac{\eta^2 c^2 (1-\gamma)^2 \widetilde{\lambda}_j^2}{d^2}, \frac{(1-\gamma)\widetilde{\lambda}_j(\mathbf{w}_{0j}^\top \mathbf{x}_i)^2}{d}\right\}.$$

Thus, for all $m \geq k$,

$$\max_{j \in [m]} \left( \lambda_{m,j} \Big( \sigma(Z_j(\mathbf{x}_i; \mathbf{W}_1)) - \sigma(Z_j(\mathbf{x}_i; \mathbf{W}_0)) \Big)^2 \right)$$

$$\geq \max_{j \in [k]} \left( \mathbf{1}_{\left\{ \mathbf{w}_{0j}^\top \mathbf{x}_i \geq 0 \right\}} \cdot \min \left\{ \frac{\eta^2 c^2 (1 - \gamma)^2 \widetilde{\lambda}_j^2}{d^2}, \frac{(1 - \gamma) \widetilde{\lambda}_j (\mathbf{w}_{0j}^\top \mathbf{x}_i)^2}{d} \right\} \right),$$

which implies

$$\liminf_{m \to \infty} \left( \max_{j \in [m]} \left( \lambda_{m,j} \Big( \sigma(Z_j(\mathbf{x}_i; \mathbf{W}_1)) - \sigma(Z_j(\mathbf{x}_i; \mathbf{W}_0)) \Big)^2 \right) \right)$$

$$\geq \max_{j \in [k]} \left( \mathbf{1}_{\left\{ \mathbf{w}_{0j}^\top \mathbf{x}_i \geq 0 \right\}} \cdot \min \left\{ \frac{\eta^2 c^2 (1 - \gamma)^2 \widetilde{\lambda}_j^2}{d^2}, \frac{(1 - \gamma) \widetilde{\lambda}_j (\mathbf{w}_{0j}^\top \mathbf{x}_i)^2}{d} \right\} \right). \quad \text{(S.32)}$$

We will show that for all $\delta \in (0, 1/2)$, with probability at least $1 - (1/2 + \delta)^k$, the lower bound in Equation (S.32) is positive and

$$0 < \sum_{j=1}^{\infty} \lambda_{m,j} \sigma(Z_j(\mathbf{x}_i; \mathbf{W}_0))^2 < \infty. \quad \text{(S.33)}$$

This will prove the claim of the theorem.

Pick $\delta \in (0, 1/2)$. Let $E$ be the event $\{c > 0\}$. Then, $\Pr(E) = 1$ by Lemma I.1. Note that the first argument of the minimum in the lower bound of Equation (S.32) is positive on the event $E$. Let $\epsilon > 0$ be a positive constant such that

$$\Pr(\mathbf{w}_{0j}^\top \mathbf{x}_i \geq \epsilon) \geq \frac{1}{2} - \delta \text{ for all } j \leq k, \quad \text{(S.34)}$$

which is possible since each $\mathbf{w}_{0j}^\top \mathbf{x}_i$ is a centred normal random variable with variance $\|\mathbf{x}_i\|^2 > 0$. Define $E'_\delta$ be the event $\bigcup_{j=1}^{k} \{\mathbf{w}_{0j}^\top \mathbf{x}_i \geq \epsilon\}$. Then, since $\mathbf{w}_{01}^\top \mathbf{x}_i, \ldots, \mathbf{w}_{0k}^\top \mathbf{x}_i$ are independent and the lower bound in Equation (S.34) holds, we have

$$\Pr(E \cap E'_\delta) \geq 1 - (1/2 + \delta)^k.$$

Now condition on $E \cap E'_\delta$. Then, there exists some $j \leq k$ such that $\mathbf{w}_{0j}^\top \mathbf{x}_i \geq \epsilon$. Thus, the lower bound in Equation (S.32) is positive as shown below:

$$\max_{j' \in [k]} \left( \mathbf{1}_{\left\{ \mathbf{w}_{0j'}^\top \mathbf{x}_i \geq 0 \right\}} \cdot \min \left\{ \frac{\eta^2 c^2 (1 - \gamma)^2 \widetilde{\lambda}_{j'}^2}{d^2}, \frac{(1 - \gamma) \widetilde{\lambda}_{j'} (\mathbf{w}_{0j'}^\top \mathbf{x}_i)^2}{d} \right\} \right)$$

$$\geq \mathbf{1}_{\left\{ \mathbf{w}_{0j}^\top \mathbf{x}_i \geq 0 \right\}} \cdot \min \left\{ \frac{\eta^2 c^2 (1 - \gamma)^2 \widetilde{\lambda}_j^2}{d^2}, \frac{(1 - \gamma) \widetilde{\lambda}_j (\mathbf{w}_{0j}^\top \mathbf{x}_i)^2}{d} \right\}$$

$$\geq \min \left\{ \frac{\eta^2 c^2 (1 - \gamma)^2 \widetilde{\lambda}_j^2}{d^2}, \frac{(1 - \gamma) \widetilde{\lambda}_j \epsilon^2}{d} \right\}$$

$$> 0.$$

Thus, with probability at least $1 - (1/2 + \delta)^k$, we have

$$\liminf_{m \to \infty} \left( \max_{j \in [m]} \lambda_{m,j} \Big( \sigma(Z_j(\mathbf{x}_i; \mathbf{W}_1)) - \sigma(Z_j(\mathbf{x}_i; \mathbf{W}_0)) \Big)^2 \right) > 0.$$

Also, under the same conditioning, we have

$$\sum_{j'=1}^{\infty} \lambda_{m,j'} \cdot \sigma(Z_{j'}(\mathbf{x}_i; \mathbf{W}_0))^2 \geq \lambda_{m,j} \cdot \sigma(Z_j(\mathbf{x}_i; \mathbf{W}_0))^2$$

$$= \frac{\lambda_{m,j}}{d} \cdot \mathbf{1}_{\{\mathbf{w}_j^\top \mathbf{x}_i \geq 0\}} \cdot (\mathbf{w}_j^\top \mathbf{x}_i)^2$$

$$\geq \frac{(1-\gamma)\widetilde{\lambda}_j}{d} \cdot \mathbf{1}_{\{\mathbf{w}_j^\top \mathbf{x}_i \geq 0\}} \cdot (\mathbf{w}_j^\top \mathbf{x}_i)^2$$

$$\geq \frac{(1-\gamma)\widetilde{\lambda}_j}{d} \cdot \epsilon^2$$

$$> 0.$$

Furthermore, without any conditioning, we have

$$\sum_{j'=1}^{\infty} \lambda_{m,j'} \cdot \sigma(Z_{j'}(\mathbf{x}_i; \mathbf{W}_0))^2 < \infty$$

almost surely, because again without any conditioning, the usual expectation of the right-hand side of the above inequality is finite as shown below:

$$\mathbb{E}\left[\sum_{j'=1}^{\infty} \lambda_{m,j'} \cdot \sigma(Z_{j'}(\mathbf{x}_i; \mathbf{W}_0))^2\right] = \sum_{j'=1}^{\infty} \lambda_{m,j'} \cdot \mathbb{E}\left[\sigma(Z_{j'}(\mathbf{x}_i; \mathbf{W}_0))^2\right] = \sum_{j'=1}^{\infty} \lambda_{m,j'} \cdot \frac{\|\mathbf{x}_i\|^2}{2d}$$

$$= \frac{\|\mathbf{x}_i\|^2}{2d} < \infty.$$

Thus, Equation (S.33) holds with probability at least $1 - (1/2 + \delta)^k$. This completes the proof of the theorem.

### J.2 Proof of Theorem A.4

We first compute a lower bound for the squared norm of the gradient, which does not depend on $m$.

$$\left\|\nabla_{\mathbf{w}_{tj}} L(\mathbf{W}_t)\big|_{t=0}\right\|^2 = \left\|\sum_{i=1}^{n} y_i \sqrt{\lambda_{m,j}} a_j \sigma'\left(\frac{\mathbf{w}_{0j}^\top \mathbf{x}_i}{\sqrt{d}}\right) \frac{\mathbf{x}_i}{\sqrt{d}}\right\|^2$$

$$= \frac{\lambda_{m,j}}{d} \left\|\sum_{i=1}^{n} y_i \sigma'\left(\frac{\mathbf{w}_{0j}^\top \mathbf{x}_i}{\sqrt{d}}\right) \mathbf{x}_i\right\|^2$$

$$\geq \frac{(1-\gamma)\widetilde{\lambda}_j}{d} \left\|\sum_{i=1}^{n} y_i \sigma'\left(\frac{\mathbf{w}_{0j}^\top \mathbf{x}_i}{\sqrt{d}}\right) \mathbf{x}_i\right\|^2$$

$$= \frac{(1-\gamma)\widetilde{\lambda}_j}{d} \left|\sum_{k=1}^{d}\sum_{i=1}^{n}\sum_{i'=1}^{n} y_i y_{i'} \sigma'\left(\frac{\mathbf{w}_{0j}^\top \mathbf{x}_i}{\sqrt{d}}\right) \sigma'\left(\frac{\mathbf{w}_{0j}^\top \mathbf{x}_{i'}}{\sqrt{d}}\right) x_{ik} x_{i'k}\right|$$

$$= \frac{(1-\gamma)\widetilde{\lambda}_j}{d} \left|\sum_{i=1}^{n}\sum_{i'=1}^{n} y_i y_{i'} \left(\mathbf{x}_i^\top \mathbf{x}_{i'} \sigma'\left(\frac{\mathbf{w}_{0j}^\top \mathbf{x}_i}{\sqrt{d}}\right) \sigma'\left(\frac{\mathbf{w}_{0j}^\top \mathbf{x}_{i'}}{\sqrt{d}}\right)\right)\right|$$

$$= \frac{(1-\gamma)\widetilde{\lambda}_j}{d} \left|\sum_{i=1}^{n}\sum_{i'=1}^{n} y_i y_{i'} \left(\mathbf{x}_i^\top \mathbf{x}_{i'} \mathbf{1}_{\{\mathbf{w}_{0j}^\top \mathbf{x}_i \geq 0\}} \mathbf{1}_{\{\mathbf{w}_{0j}^\top \mathbf{x}_{i'} \geq 0\}}\right)\right|.$$

Thus,

$$\liminf_{m\to\infty} \left\|\nabla_{\mathbf{w}_{tj}} L(\mathbf{W}_t)\big|_{t=0}\right\|^2 \geq \frac{(1-\gamma)\widetilde{\lambda}_j}{d} \left|\sum_{i=1}^{n}\sum_{i'=1}^{n} y_i y_{i'} \left(\mathbf{x}_i^\top \mathbf{x}_{i'} \mathbf{1}_{\{\mathbf{w}_{0j}^\top \mathbf{x}_i \geq 0\}} \mathbf{1}_{\{\mathbf{w}_{0j}^\top \mathbf{x}_{i'} \geq 0\}}\right)\right|.$$

But by assumption, the factor $((1-\gamma)\widetilde{\lambda}_j)/d$ in the lower bound is always positive. The claim of the theorem follows from the property that the other factor in the lower bound is also positive with probability at least $1/2$. In the rest of the proof, we will show why this is so.

Note that

$$\|\mathbf{x}_i\|^2 \mathbf{1}_{\left\{\mathbf{w}_{0j}^\top \mathbf{x}_i \geq 0\right\}} > 0.$$

if and only if $\mathbf{w}_{0j}^\top \mathbf{x}_i \geq 0$.

Condition on $\mathbf{w}_{0j}$ and recall that the $y_i$'s are continuous independent real-valued random variables which are also independent from $\mathbf{w}_{0j}$, thus their distributions are unaffected by the conditioning. If $\mathbf{w}_{0j}^\top \mathbf{x}_i \geq 0$, the inequality

$$\left| \sum_{i=1}^n \sum_{i'=1}^n y_i y_{i'} \left( \mathbf{x}_i^\top \mathbf{x}_{i'} \mathbf{1}_{\{\mathbf{w}_{0j}^\top \mathbf{x}_i \geq 0\}} \mathbf{1}_{\{\mathbf{w}_{0j}^\top \mathbf{x}_{i'} \geq 0\}} \right) \right| > 0$$

holds almost surely. Since $\mathbf{w}_{0j}^\top \mathbf{x}_i$ holds with probability $1/2$, the above inequality holds unconditionally with probability at least $1/2$, as desired.

## K   Additional experimental results (smooth activation)

We provide here additional results for the experiments described in Section 8.

### K.1   Regression

In Figures 2, 3, 4 and 5 we respectively provide the detailed results for the datasets `concrete`, `energy`, `airfoil` and `plant`.

### K.2   Classification

We provide in Figure 6 detailed results for the MNIST dataset, and in Figure 7 results for the CIFAR–10 dataset. In Figure 8, we provide further details on the individual impact of the parameter $\gamma \in [0, 1]$. Recall that the smaller the value of $\gamma$, the more asymmetry is introduced, where $\gamma = 1$ recovers the iid model. We can see from the experiments that pruning performance is improved as $\gamma$ becomes smaller.

## L   Experimental results for the ReLU activation function

We provide here additional experimental results, as in Appendix K, but with a different activation function. The experimental setting is the same as described in Section 8, except that the swish activation function is replaced by the ReLU function. Although our theory does not cover the convergence of GD with the ReLU, the experimental results obtained in this section are quantitatively similar to those obtained with the swish function.

### L.1   Regression

In Figures 9, 10, 11 and 12 we respectively provide detailed results for the datasets `concrete`, `energy`, `airfoil` and `plant`.

### L.2   Classification

We provide in Figures 13, 14 and 15 detailed results for respectively the MNIST, CIFAR10 and CIFAR100 experiments.

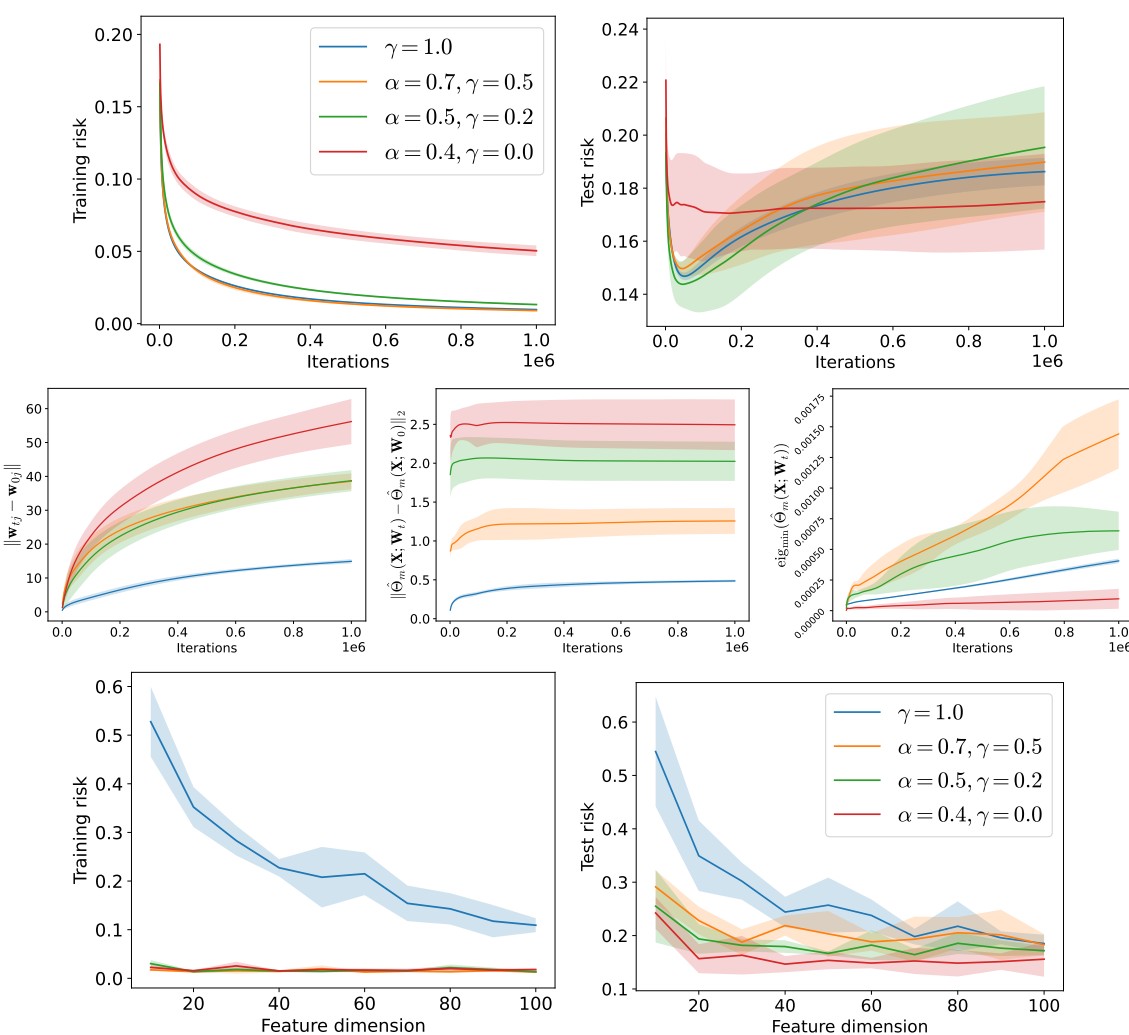

Figure 2: Results for the `concrete` dataset (swish). From left to right and top to bottom, 1) training risks, 2) test risks, 3) differences in weight norms $\|\mathbf{w}_{tj} - \mathbf{w}_{0j}\|$ with $j$'s being the neurons having the maximum difference at the end of the training, 4) difference in NTG matrices, 5) minimum NTG eigenvalues, 6) training risks for transfer learning, and 7) test risks for transfer learning.

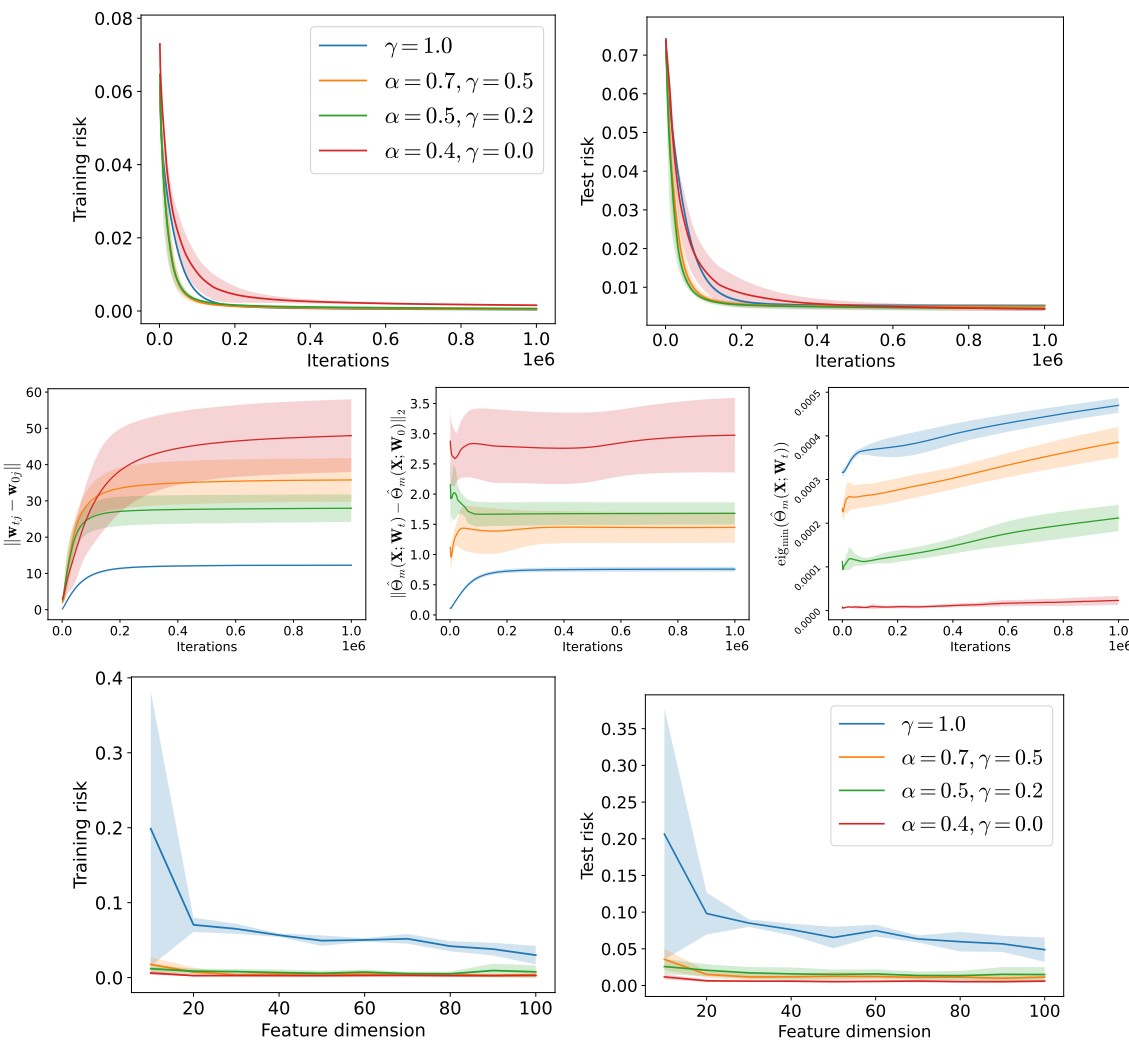

Figure 3: Results for the `energy` dataset (swish). From left to right and top to bottom, 1) training risks, 2) test risks, 3) differences in weight norms $\|\mathbf{w}_{tj} - \mathbf{w}_{0j}\|$ with $j$'s being the neurons having the maximum difference at the end of the training, 4) difference in NTG matrices, 5) minimum NTG eigenvalues, 6) training risks for transfer learning, and 7) test risks for transfer learning.

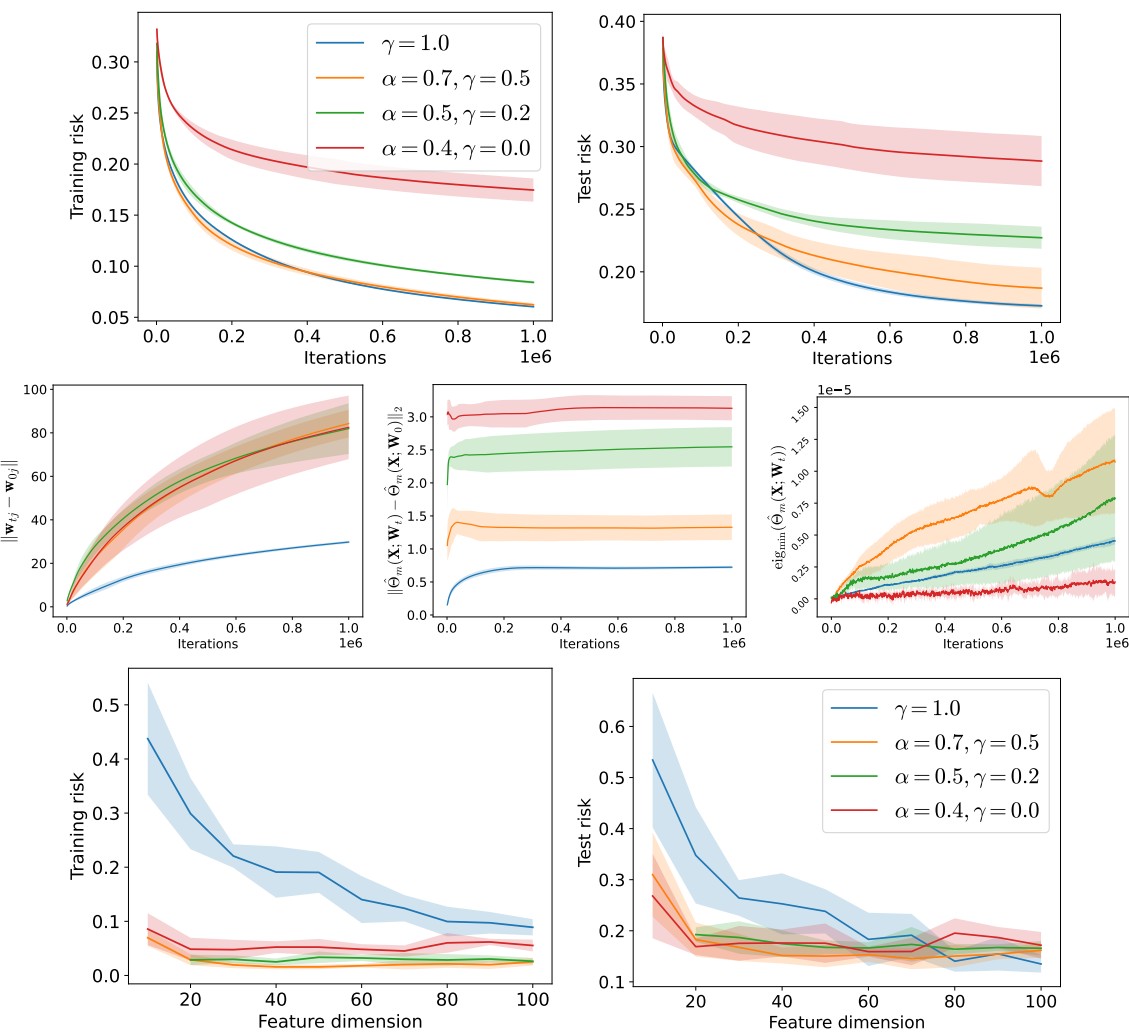

Figure 4: Results for the `airfoil` dataset (swish). From left to right and top to bottom, 1) training risks, 2) test risks, 3) differences in weight norms $\|\mathbf{w}_{tj} - \mathbf{w}_{0j}\|$ with $j$'s being the neurons having the maximum difference at the end of the training, 4) difference in NTG matrices, 5) minimum NTG eigenvalues, 6) training risks for transfer learning, and 7) test risks for transfer learning.

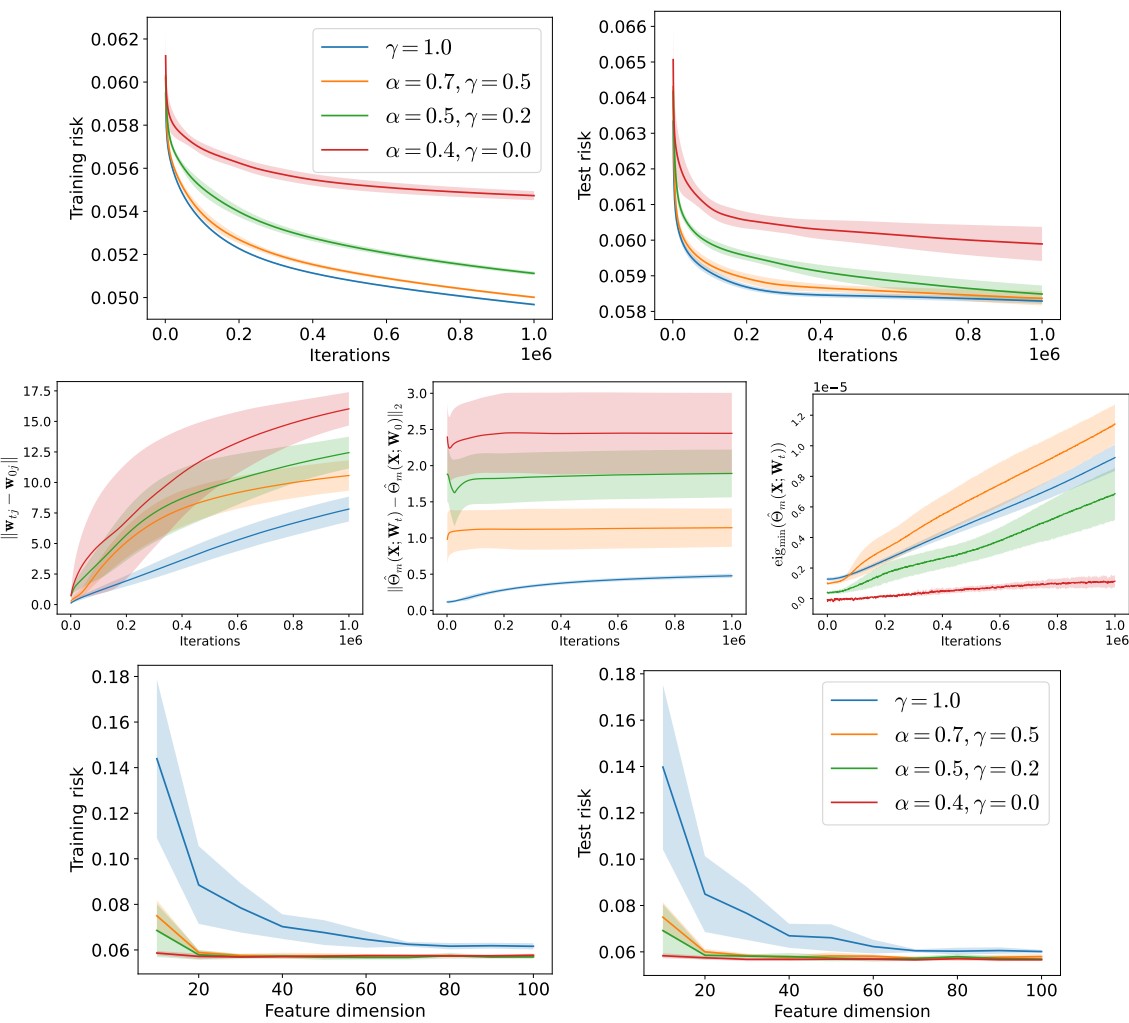

Figure 5: Results for the `plant` dataset (swish). From left to right and top to bottom, 1) training risks, 2) test risks, 3) differences in weight norms $\|\mathbf{w}_{tj} - \mathbf{w}_{0j}\|$ with $j$'s being the neurons having the maximum difference at the end of the training, 4) difference in NTG matrices, 5) minimum NTG eigenvalues, 6) training risks for transfer learning, and 7) test risks for transfer learning.

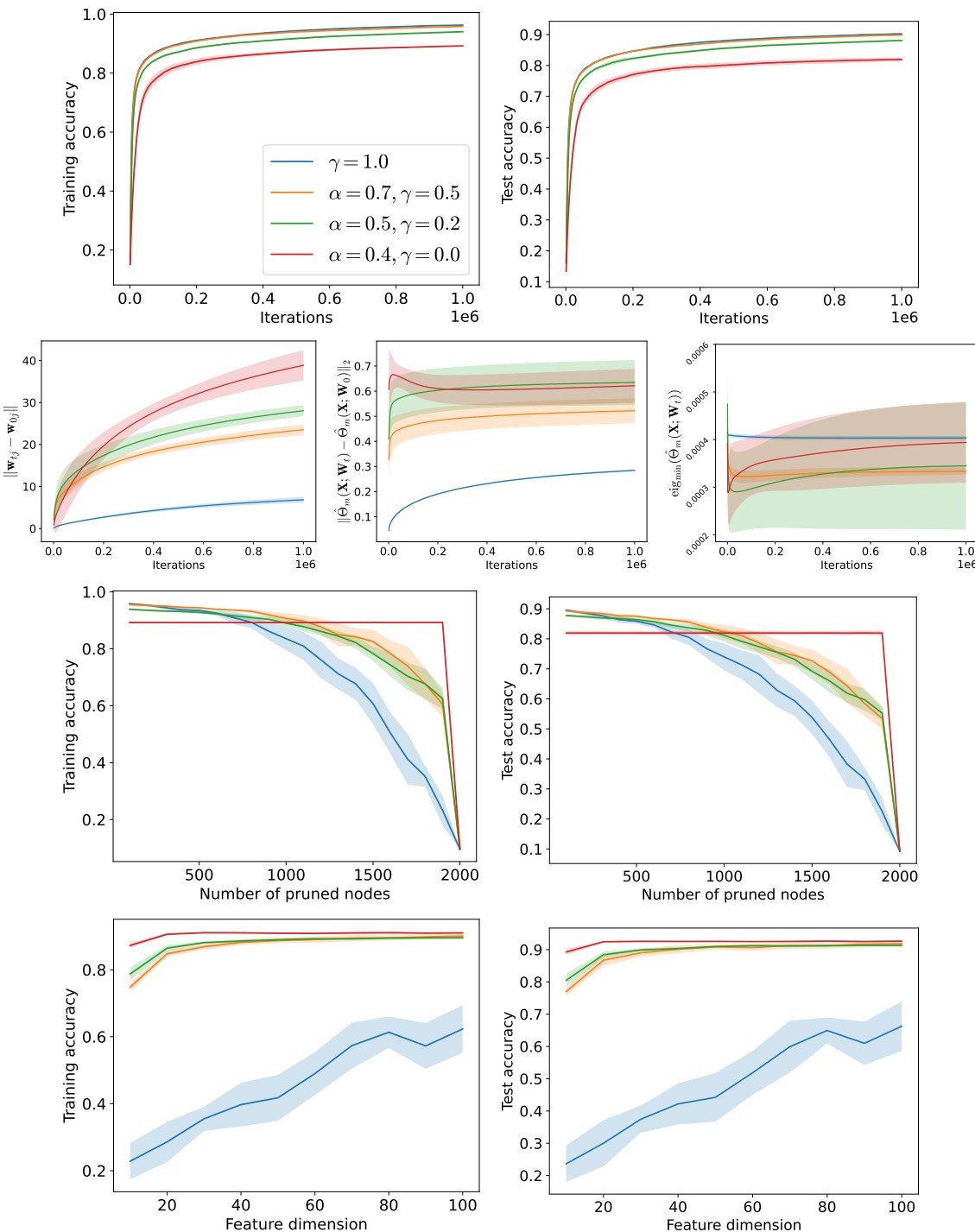

Figure 6: Results for the `MNIST` dataset (swish). From left to right and top to bottom, 1) training risks, 2) test risks, 3) differences in weight norms $\|\mathbf{w}_{tj} - \mathbf{w}_{0j}\|$ with $j$'s being the neurons having the maximum difference at the end of the training, 4) difference in NTG matrices, 5) minimum NTG eigenvalues, 6) training accuracies for pruning, 7) test accuracies for pruning, 8) training accuracies for transfer learning, and 9) test accuracies for transfer learning.

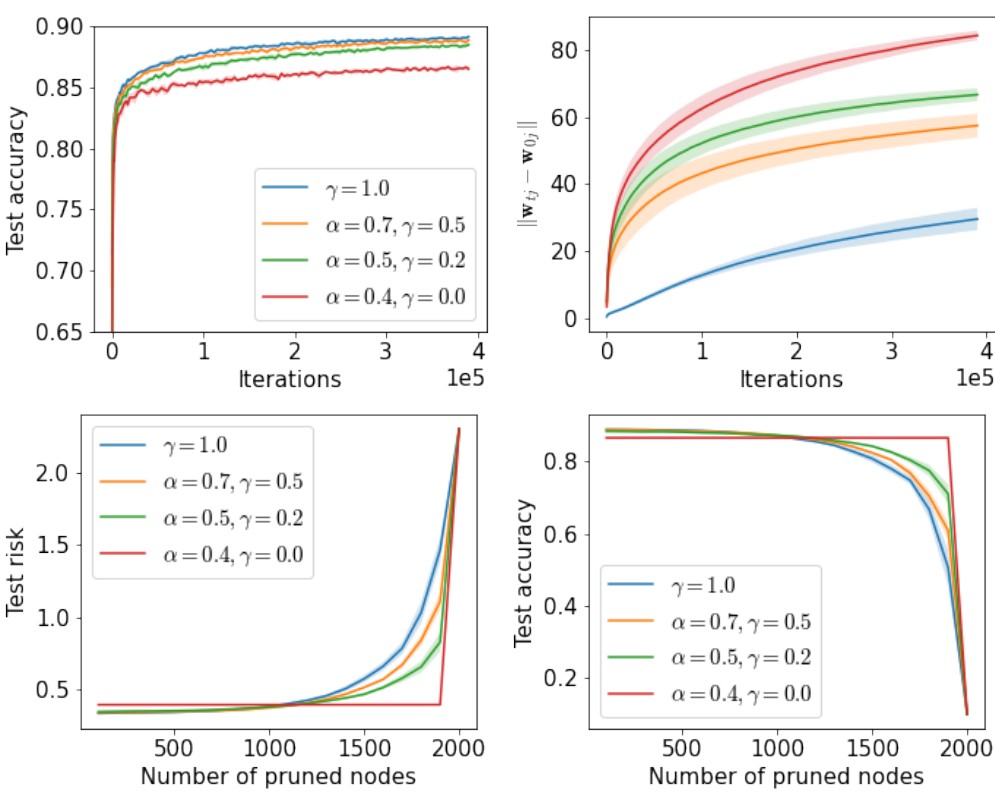

Figure 7: Results for the `CIFAR-10` dataset (swish). From left to right and top to bottom, 1) test accuracies through training, 2) differences in weight norms $\|\mathbf{w}_{tj} - \mathbf{w}_{0j}\|$ with $j$'s being the neurons having the maximum difference at the end of the training, 3) test risks of the pruned models, and 4) test accuracies of the pruned models.

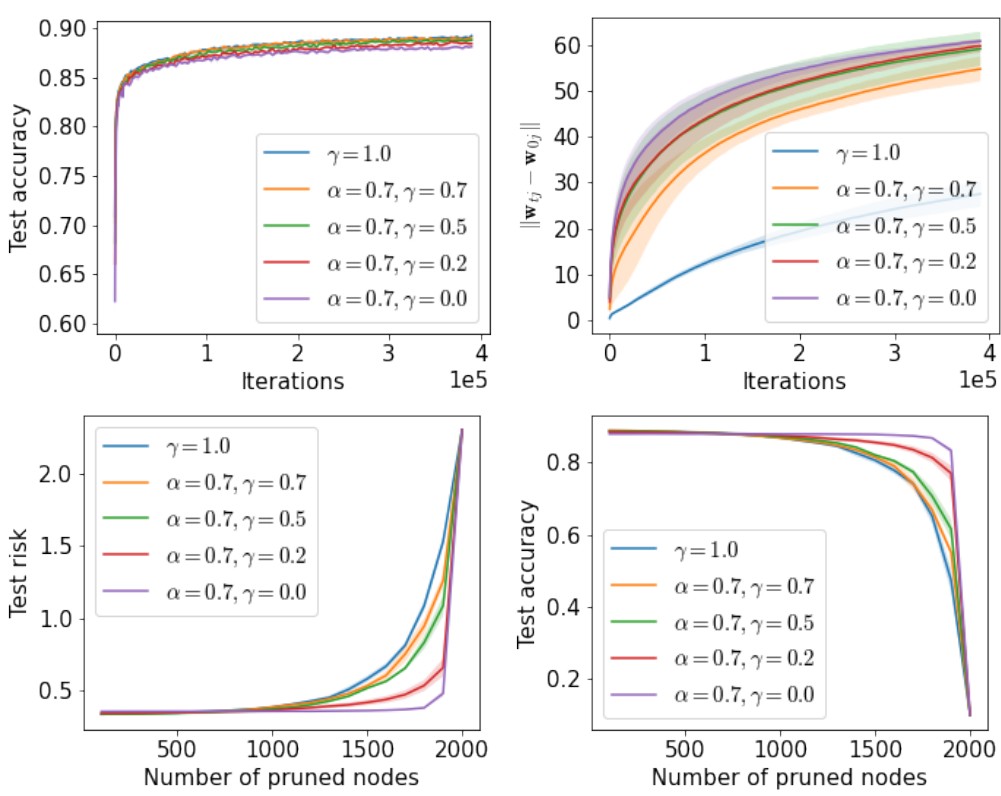

Figure 8: Results for the `CIFAR-10` dataset (swish). Impact of the parameter $\gamma$. From left to right and top to bottom, 1) test accuracies through training, 2) differences in weight norms $\|\mathbf{w}_{tj} - \mathbf{w}_{0j}\|$ with $j$'s being the neurons having the maximum difference at the end of the training, 3) test risks of the pruned models, and 4) test accuracies of the pruned models.

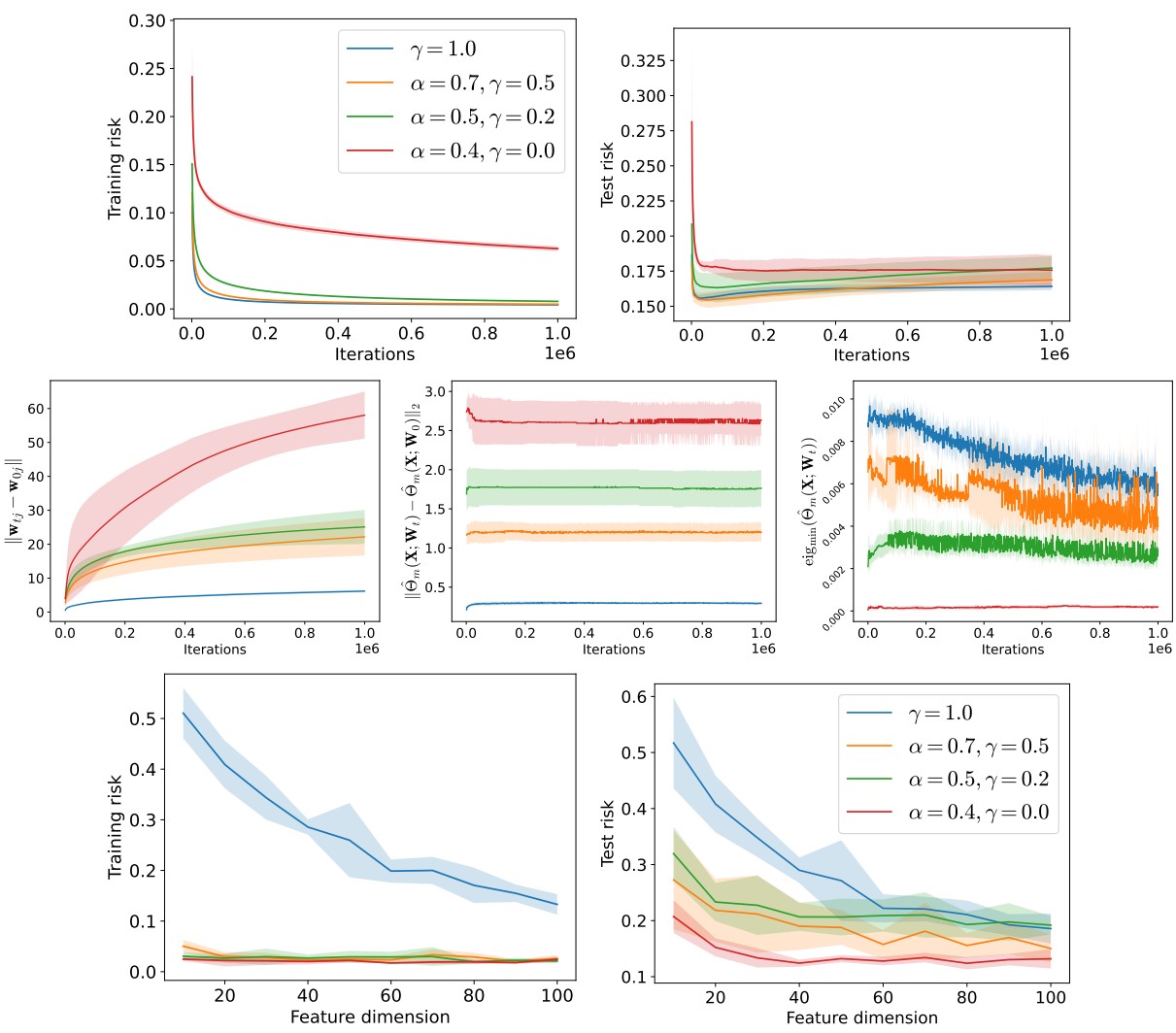

Figure 9: Results for the `concrete` dataset (ReLU). From left to right and top to bottom, 1) training risks, 2) test risks, 3) differences in weight norms $\|\mathbf{w}_{tj} - \mathbf{w}_{0j}\|$ with $j$'s being the neurons having the maximum difference at the end of the training, 4) difference in NTG matrices, 5) minimum NTG eigenvalues, 6) training risks for transfer learning, and 7) test risks for transfer learning.

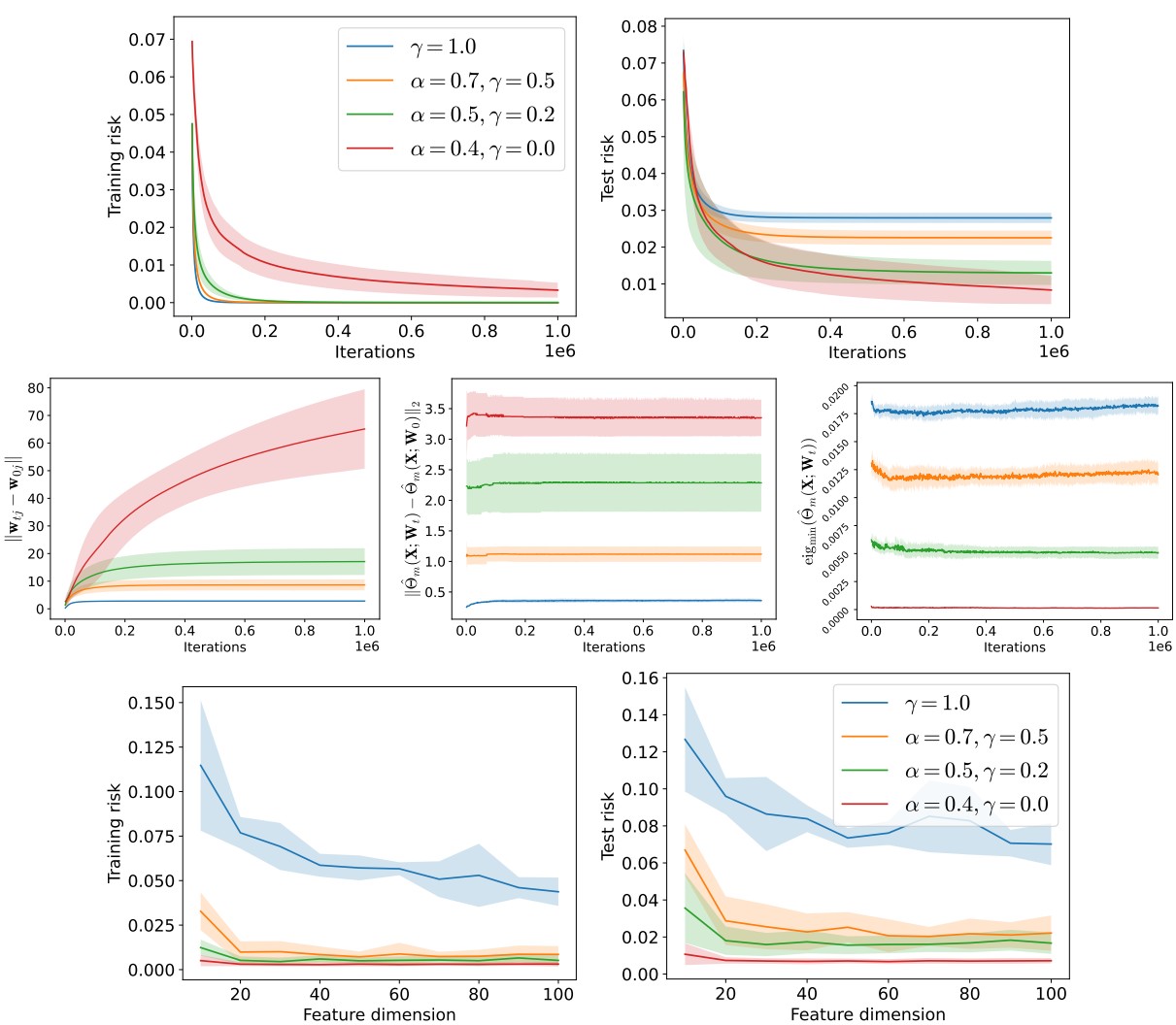

Figure 10: Results for the **energy** dataset (ReLU). From left to right and top to bottom, 1) training risks, 2) test risks, 3) differences in weight norms $\|\mathbf{w}_{tj} - \mathbf{w}_{0j}\|$ with $j$'s being the neurons having the maximum difference at the end of the training, 4) difference in NTG matrices, 5) minimum NTG eigenvalues, 6) training risks for transfer learning, and 7) test risks for transfer learning.

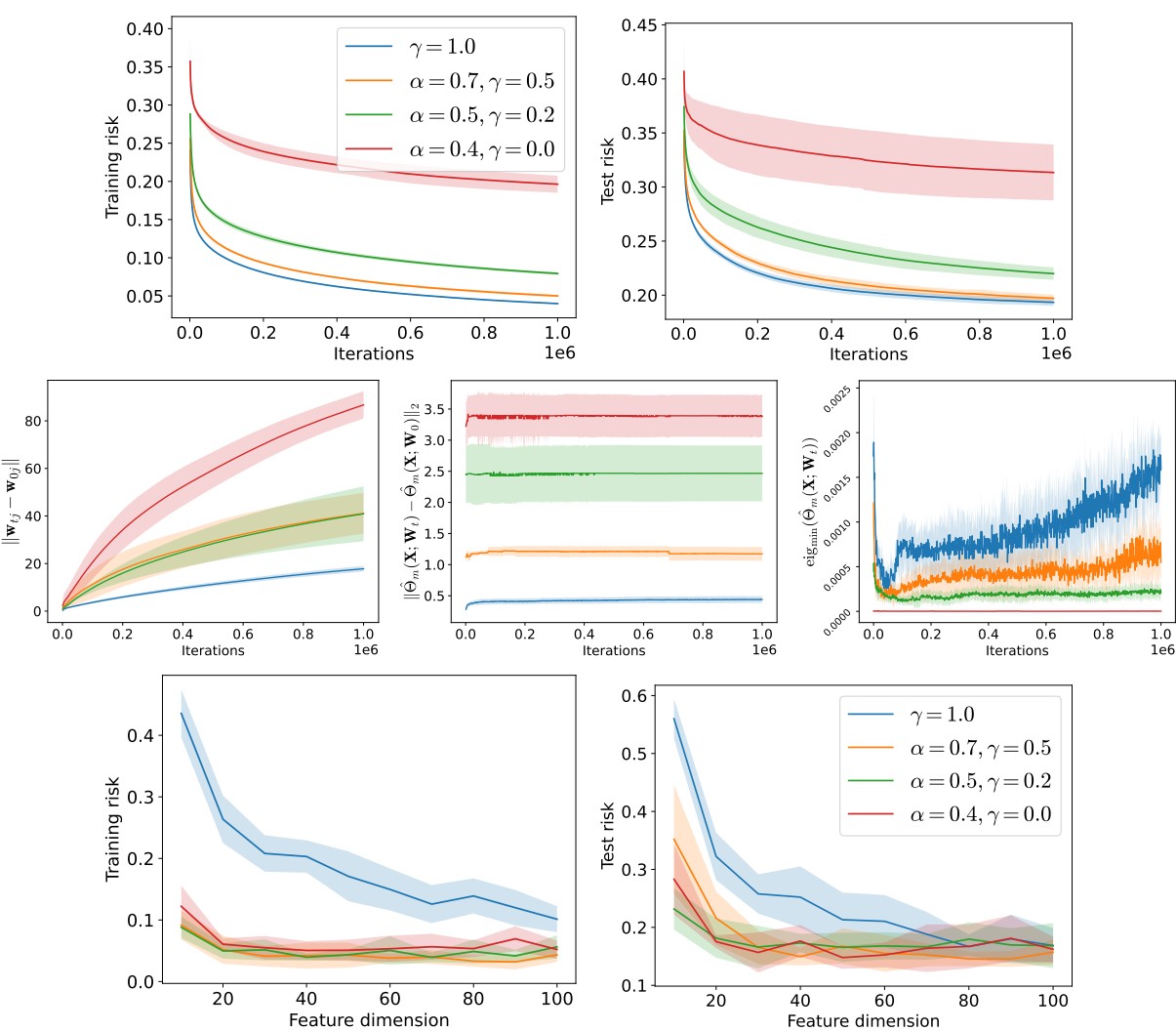

Figure 11: Results for the `airfoil` dataset (ReLU). From left to right and top to bottom, 1) training risks, 2) test risks, 3) differences in weight norms $\|\mathbf{w}_{tj} - \mathbf{w}_{0j}\|$ with $j$'s being the neurons having the maximum difference at the end of the training, 4) difference in NTG matrices, 5) minimum NTG eigenvalues, 6) training risks for transfer learning, and 7) test risks for transfer learning.

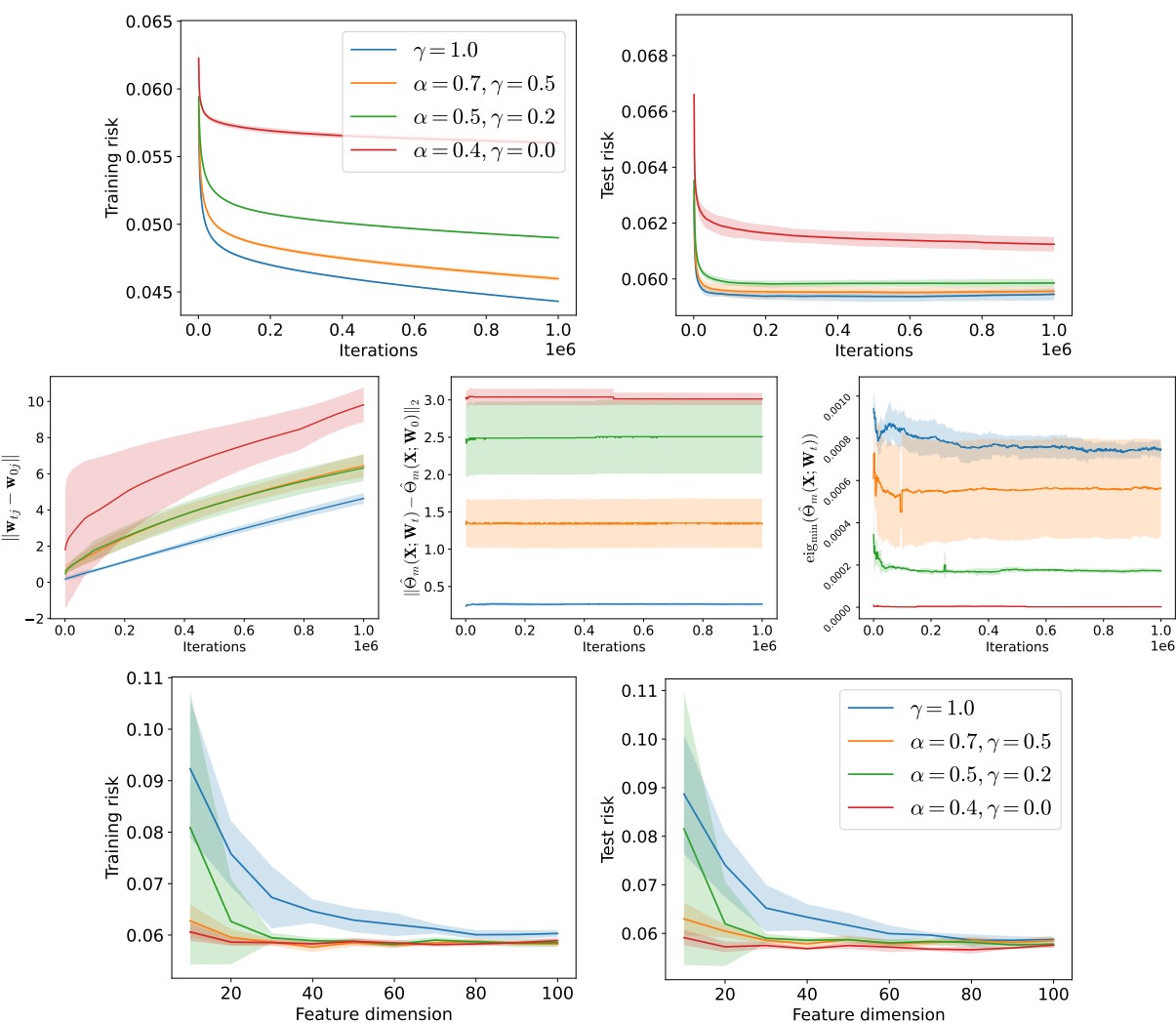

Figure 12: Results for the `plant` dataset (ReLU). From left to right and top to bottom, 1) training risks, 2) test risks, 3) differences in weight norms $\|\mathbf{w}_{tj} - \mathbf{w}_{0j}\|$ with $j$'s being the neurons having the maximum difference at the end of the training, 4) difference in NTG matrices, 5) minimum NTG eigenvalues, 6) training risks for transfer learning, and 7) test risks for transfer learning.

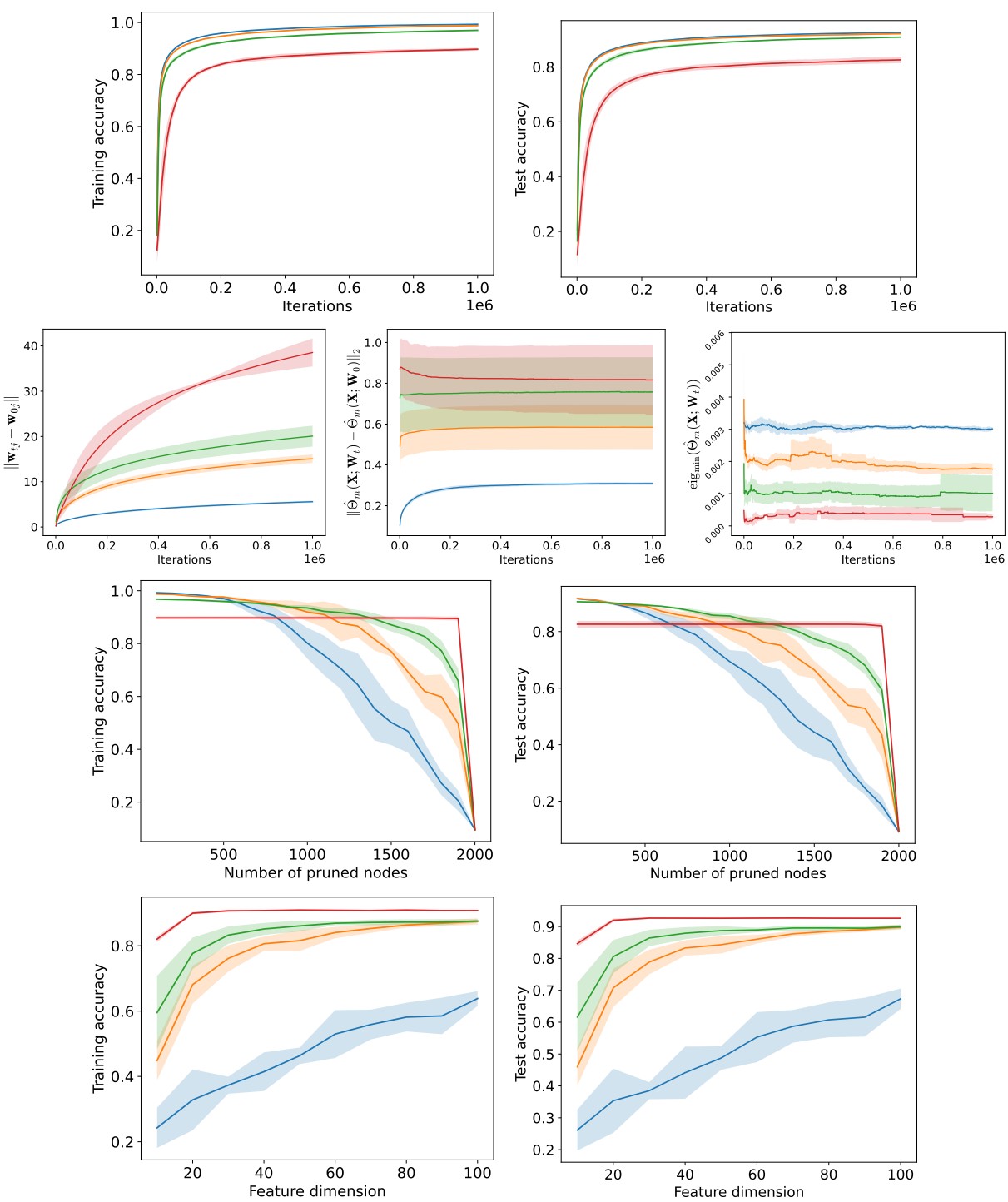

Figure 13: Results for the `MNIST` dataset (ReLU). From left to right and top to bottom, 1) training risks, 2) test risks, 3) differences in weight norms $\|\mathbf{w}_{tj} - \mathbf{w}_{0j}\|$ with $j$'s being the neurons having the maximum difference at the end of the training, 4) difference in NTG matrices, 5) minimum NTG eigenvalues, 6) training accuracies for pruning, 7) test accuracies for pruning, 8) training accuracies for transfer learning, and 9) test accuracies for transfer learning.

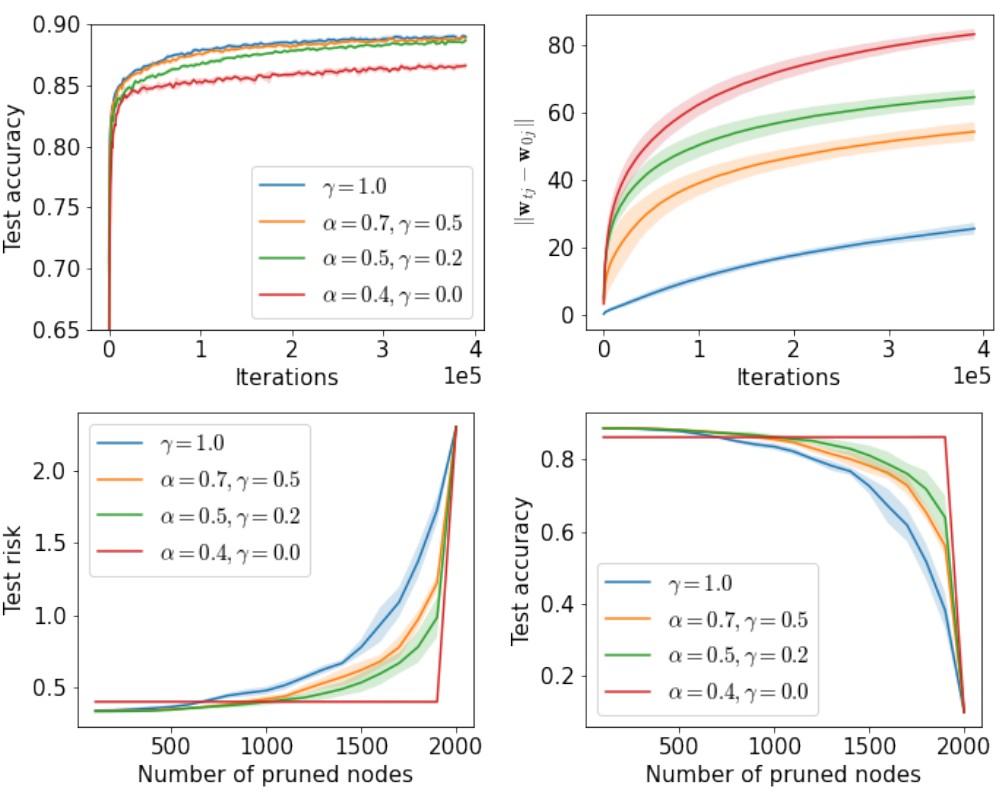

Figure 14: Results for the CIFAR-10 dataset (ReLU). From left to right and top to bottom, 1) test accuracies through training, 2) differences in weight norms $\|\mathbf{w}_{tj} - \mathbf{w}_{0j}\|$ with $j$'s being the neurons having the maximum difference at the end of the training, 3) test risks of the pruned models, and 4) test accuracies of the pruned models.

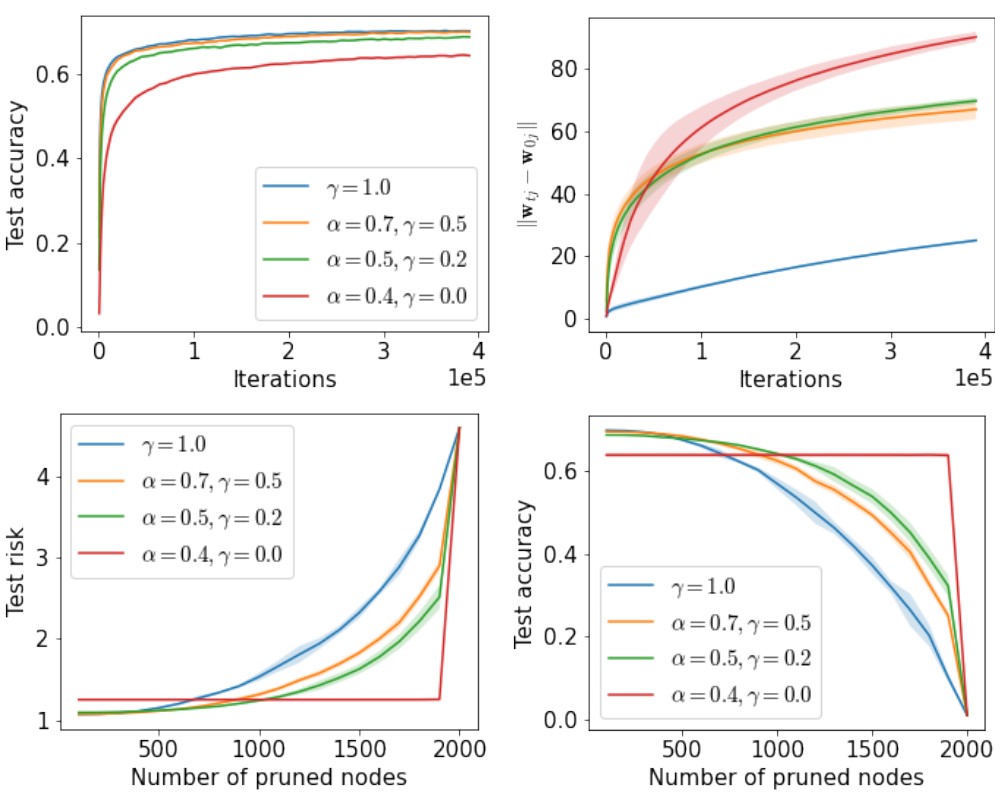

Figure 15: Results for the `CIFAR-100` dataset (ReLU). From left to right and top to bottom, 1) test accuracies through training, 2) differences in weight norms $\|\mathbf{w}_{tj} - \mathbf{w}_{0j}\|$ with $j$'s being the neurons having the maximum difference at the end of the training, 3) test risks of the pruned models, and 4) test accuracies of the pruned models.

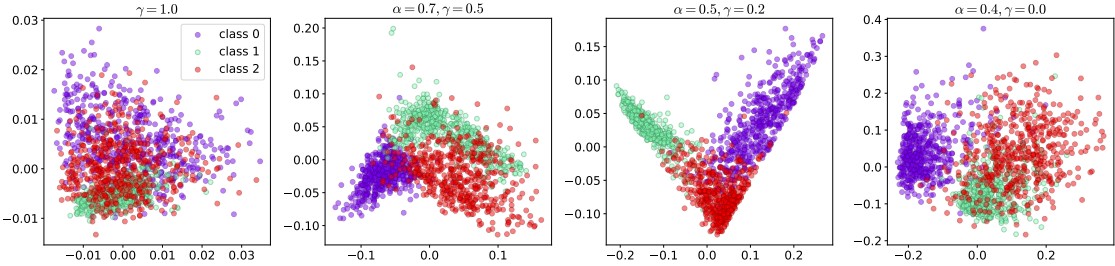

Figure 16: Visualisation of features for MNIST data. We use the top two PCA components to plot the points on a 2D space.

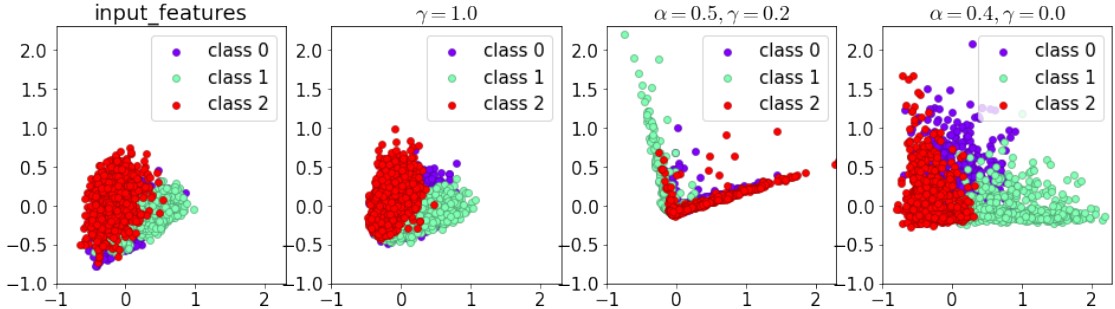

Figure 17: Visualisation of features learnt for the Cifar10 experiment. The models are trained by taking as input the hidden representation of a ResNet18 trained on ImageNet (first figure on the left). We use the top two PCA components to plot the points on a 2D space.

## M   Visualisation of the learned features.

This section aims at visualizing the main features learned in the MNIST and CIFAR experiments reported in the main text. Inspired by (Yang and Hu, 2021), we plot the first two PCA components of the learned features for MNIST (Figure 16) and CIFAR10 (Figure 17) datasets. For the MNIST dataset, as in (Yang and Hu, 2021), the figures show that the features are quasi-random with the symmetric NTK setting, while there is more separation under the asymmetric scaling. For the CIFAR10 experiment, which uses pre-trained features on ImageNet, the features of the symmetric NTK are similar to those of the pre-trained features. The features obtained by PCA better differentiates between the class.

## N   Hyper-parameter transfer.

When scaling-up neural networks, hyper-parameters tuning becomes prohibitively expensive. In practice, one performs hyper-parameter optimization on a smaller version of the model, and uses (transfers) the found values for training the larger model. However, this requires stability of the optimal parameters. As identified in (Yang et al., 2022), the standard pytorch implementation is not stable as the width increases, which can be a major challenge to scale-up models. In this section, we empirically show that the asymmetrical parameterization enjoys stability of the optimal learning rate. We train FFNN with a single hidden layer on Cifar10 for different width $P = 1024, 2048, 4096$. We compare the standard Pytorch parameterization with the asymmetrical one ($\gamma = 0.2$, $\alpha = 0.5$). The results are reported in 18. As expected, in the standard parameterization, the optimal learning rate shifts; as the width increases, the optimal learning rate becomes smaller. On the other hand, with the asymmetrical scaling, the optimal learning rate remains stable as the width increases

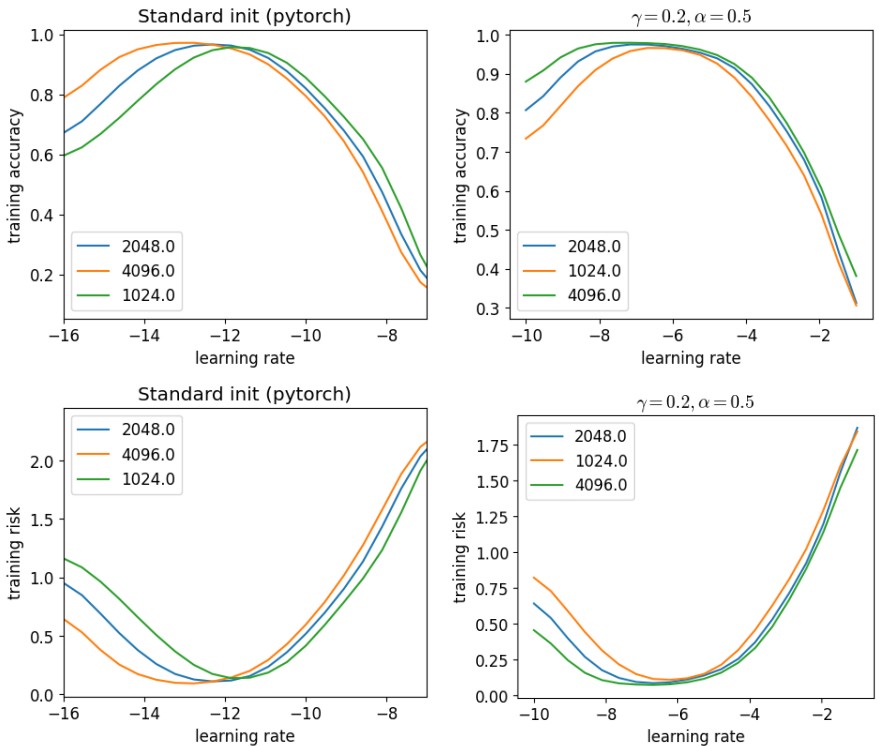

Figure 18: Stability of the optimal learning rate as the width increases. Training error in terms of (top) accuracy (bottom) cross-entropy for (left) standard parameterisation and (right) asymmetrical parameterisation.