# OpenReview forum: "Over-parameterised Shallow Neural Networks with Asymmetrical Node Scaling: Global Convergence Guarantees and Feature Learning"
_TMLR — Accepted by TMLR_

### Review · Reviewer_XJbp · 2024-10-16

**Summary Of Contributions:**

This paper considers the gradient flow/descent training dynamics of infinitely wide two layer neural networks under an asymmetric scaling of the neurons. In the considered formulation, the completely symmetric case reduces to the lazy training regime where the neural tangent kernel remains constant and no feature learning regime is achieved. In the asymmetric case, the authors prove that both feature learning and exponential convergence of the training loss can be achieved. Additionally, the authors provide examples where asymmetric scaling can lead to non-uniform feature learning which further enables pruning. Several experiments support the theoretical findings.

**Audience:**

Yes

**Broader Impact Concerns:**

Not applicable.

**Claims And Evidence:**

Yes

**Requested Changes:**

* Is there an intuitive explanation why the first Gram matrix in Equation (15) does not change significantly even when the weights do change? It might be helpful to highlight this in the main text.
* In the classification experiments, the best test accuracy is achieved in the lazy regime where $\gamma = 1$. This is surprising given the general intuition that feature learning achieves better generalization when the task has some hidden low-dimensional structure. Is there an explanation for this observation?
* The lower bounds on $m$ in Theorem 5.1 and 6.1 are getting better in high dimensions. Does it mean/Is it simply because overfitting is easier in high dimensions when the system becomes underdetermined?
* Where does the statement $\sum_{j \geq \lfloor \rho m\rfloor}\lambda_{m,j} \rightarrow \gamma(1-\rho)$ in Proposition 7.6 come from? Does it not depend on how $\lambda_{m,j}$ are chosen?

**Strengths And Weaknesses:**

# Strengths:
The asymmetric scaling formulation provides flexibility in choosing the training regime, while keeping the mathematical analysis tractable based on tools developed by prior works.

# Weaknesses:
* Theorem 5.1 does not guarantee feature learning, the upper bounds simply suggest that the weights may move significantly from initialization, but the upper bounds may not be sharp. However, feature learning is proved for simpler settings in Section 7.
* Generally, it is not clear why the notion of feature learning introduced here, i.e. simply the weights moving significantly, is useful on its own. Several works have shown that feature learning is useful because it leads to better generalization guarantees by adapting to the task structure, e.g. [1].
* Intuitively, I'm not sure I understand why even though the weights move significantly, the first Gram matrix in Equation (15) stays close to initialization.

[1] B. Ghorbani et al. When Do Neural Networks Outperform Kernel Methods? NeurIPS 2020.

---

> ### Author Response · Authors · 2024-11-25
>
> We thank the reviewer for their detailed and positive review. Thank you also for providing the additional reference, which was added to the paper.
>
> 1. The intuitive reason for the first Gram matrix not changing significantly is that its change is bounded by the average of the changes of the $m$ weight vectors $w_j$ associated with the $m$ nodes up to a constant factor; for large $m$, this average becomes small because only a small number of weight vectors
> change noticeably in that case. We have added a more detailed explanation, for the simplified case of $d=1$ and the activation function $\sigma$ being smooth, at the end of Section 5.
>
>
> 2. In the experiments we performed, the (unpruned) asymmetrical network did not necessarily outperform the symmetrical network. As noted by the reviewer, this is in particular the case in the classification examples, where the symmetric NTK slightly outperforms the asymmetrical one. We agree with the reviewer that this is somewhat counterintuitive, as the learned weights are clearly adapted to the task, as demonstrated on the pruning results. This may be due to the beneficial additional regularisation brought by the symmetric NTK. One other reason could be that "low" is relative when we say low-dimensional structure, since it is not clear how many features are needed to classify MNIST and CIFAR. The single ReLU experiment shows that in at least one experiment with very low-dimensional structure, the asymmetrical scaling performs better.
>
>
>
> 3.  Although $d$ represents the dimension of the input, the dependency
>  of $m$ on $d$ in Theorems 5.1 and 6.1 does not come from the complexity
>  of the input dimension. Instead, it comes from the use of the $1/\sqrt{d}$
>  scaling in the first layer of our model: $ f_m(\mathbf{x}; \mathbf{W}) = \sum_{j = 1}^m \sqrt{\lambda_{m,j}} a_j \sigma
>          \left(\frac{\mathbf{w}_j^\top \mathbf{x}}{\sqrt{d}} \right).$  For instance, if we used a slightly different model without this $1/\sqrt{d}$ factor,  the $d$ terms in Theorems 5.1 and 5.2 would disappear completely  (i.e., we would have had the versions of the theorems with $d$ set to $1$).  We added a comment on the dependency on $d$ in Remark 5.2.
>
> 4. This holds for any scaling in Equation (1). Indeed,
>  $$
>  \sum_{j>\lfloor \rho m \rfloor} \lambda_{m,j}=\frac{\gamma (m-\lfloor \rho m \rfloor)}{m} + (1-\gamma) \frac{\sum_{j=\lfloor \rho m \rfloor}^m \widetilde\lambda_{j}}{\sum_{j=1}^m \widetilde\lambda_j}
>  $$
>  By sandwiching, $\frac{\gamma (m-\lfloor \rho m \rfloor)}{m}\to \gamma (1-\rho)$. Additionally, the series $\sum_{j=1}^m \widetilde\lambda_j$ converges to 1. So, its tail converges to 0 and  $\sum_{j=\lfloor \rho m \rfloor+1}^m \widetilde\lambda_{j}\to 0$. We have added these derivations in Section B.2 of the Supplementary Material.

---

### Review · Reviewer_nBUn · 2024-11-07

**Summary Of Contributions:**

The paper investigates the global convergence and feature learning properties of gradient-based training on wide, shallow neural networks with asymmetrical node scaling, where the output of each hidden node has a non-uniform positive scaling parameter. This scaling method departs from the traditional NTK parameterization, with the intent to achieve feature learning in large-width networks, contrasting with the “lazy training” of NTK. The authors provide theoretical guarantees for convergence under this modified scaling and demonstrate how it allows feature learning. The paper includes numerical experiments that validate the theoretical claims and highlight additional practical benefits, such as enhanced prunability and transfer learning.

**Audience:**

Yes

**Broader Impact Concerns:**

This work is purely theoretical, no Broader Impact Concerns section is necessary.

**Claims And Evidence:**

Yes

**Requested Changes:**

1. **Clarify Definitions and Implications**: The notion of feature learning under asymmetrical scaling could be clarified, especially why it represents a stronger notion than in the standard NTK setting: it is indeed obvious that one definition implies the other, but why this definition ensures that non uniform feeatures learning relate more to the relevance of the learn features for prediction?

2. **Misc** Add $ ||w - w_0|| $ and $ ||\theta - \theta(W_0)|| $) before the big O in section 5 to make the section a bit easier to understand.

3. **Fix Minor Errors**: Address typographical errors, such as “activiation” in Section 3.3.

4. **Explain Assumptions**: Reconsider Assumption 3.3 or provide a justification for the choice of a normal distribution for initialization, especially given that alternative methods exists (like orthogonal initialization, for instance)

5. **Discuss Zipf Law**: Provide reasoning behind the selection of the Zipf law for node scaling, since this choice could appear arbitrary.

6. **Expand on Gradient Descent Stepsize**: Include a discussion about the small stepsize used for gradient descent, particularly how it compares to typical choices in non-convex optimization with smooth functions. This could highlight any advantages or limitations of the approach in practical settings.

**Strengths And Weaknesses:**

#### Strengths
- **Clear Theoretical Contribution**: The paper’s primary theoretical results, e.g. Theorem 5.1 and 6.1, are presented concisely and clearly. Moreover, those results do not require extensive or restrictive assumptions.
- **Experiments and Empirical Validation**: The experiments demonstrate the relevance of the theoretical findings. For instance, they clearly show how the asymmetrical node scaling achieves 0 training loss wityh gradient descent.
- **Relevance to Feature Learning and Transfer Learning**: By addressing the limitations of NTK scaling, which typically does not enable feature learning since the NTK is constant, the paper makes an important contribution to understanding how over-parameterized networks might achieve feature learning.

#### Weaknesses
- **Assumptions and Notations**: Assumption 3.3, which involves a normal distribution, might seem overly restrictive, as other initialization methods (e.g., orthogonal initialization) are common in practice. I beleive this this specific distribution was chosen because it makes the theoretical analysis easier, but how restrictive is this assumptions? Is it possible to generalize it to others?
- **Clarity and Presentation of Some Concepts**: Even tough most of the theoretical results are well explained, certain results lack sufficient explanation. For instance, in Proposition 7.6, the discution  around cases when $\gamma=0$ and $\gamma>0$ is clear but the implications of the inequalities could be made clearer.
- **Arbitrary Choice of Scaling**: The specific use of the Zipf law in defining node scaling appears arbitrary, as it is unclear why this particular law was selected over other potential distributions.
- **Gradient Descent Stepsize**: The stepsize for gradient descent is notably small, and there is no discussion on how slow this choice is when compared to the standard stepsize of non-convex gradient descent (1/smoothness constant). A comparison or at least a brief explanation would be appreciated.

### Overall Assessment

This paper presents an interesting approach with theoretical contributions and empirical validation. However, certain aspects could benefit from further clarification.

**Disclaimer:** I am not an expert in this area, therefore it might be hard for me to judge the impact and also to evaluate this paper w.r.t. existing work in the field.

---

> ### Author Response · Authors · 2024-11-25
>
> We thank the reviewer for their detailed and positive review.
>
> 1/ We show that the weights/neural kernel matrix change significantly with the gradient flow/descent iterations. We show that it does so in a non-uniform way, contrary to mean-field or the symmetric NTK. We show empirically that this is beneficial in terms of pruning and transfer learning. This paper does not address the question of whether the asymmetrical scaling theoretically leads to lower generalisation error for some target functions. We show empirically that this may be the case, but not necessarily, as in the classification examples. In any case, even if the unpruned networks have lower prediction accuracy, we demonstrate empirically that the pruned asymmetric networks always outperform their symmetric NTK counterparts, due to the non-uniform feature learning property.
>
>
> 2/ We have added $||w-w_0||$ and $||\theta-\theta(W_0)||$ in Section 5, thank you for the suggestion.
>
> 3/ We fixed the typo, and checked for other typos. Thank you for catching this typo.
>
> 4/ We have added some discussion regarding the choice of the initialisation. We agree with the reviewer that, although the iid Gaussian assumption is rather standard, it would be interesting to see if the results extend to other initialisations such as the orthogonal one. Our results rely on the positivity of the minimum eigenvalue shown by Du et al. (2019) under an iid Gaussian initialisation. If this result can be extended to an orthogonal initialisation, we are confident that the results in this paper could be extended to this case. We have mentioned the case of the other initialisation in a new discussion section at the end of the article.
>
> 5/ As we need weights that sum to 1, a polynomial function (Zipf) provides a nice trade-off in terms of simplicity (single parameter) and flexibility (one recovers the NTK when $\alpha\to 1$). We have added some discussion about potential alternative choices for the scalings in the discussion section.
>
> 6/ In contrast to the feature learning that occurs under mean-field scaling, our feature learning is independent of the learning rate. With regards to the learning rate as it pertains to global convergence, since our proof heavily relies on the positivity of the minimum eigenvalue $\kappa_n$, the size of the learning rate in the extension from GF to GD also depends on $\kappa_n$. We have added a sentence at the beginning of Section 6 to emphasise that this is the only place where the learning rate is relevant with regards to our proofs. There is also a comment below Theorem 6.1.

---

### Review · Reviewer_3Qfc · 2024-11-18

**Summary Of Contributions:**

The authors study two-layer fully connected neural networks. These networks have been studied intensively in two regimes: a mean-field regime, where the network learns features from data, and the NTK [Jacot et al. NeurIPS '18] or "lazy" regime [Chizat et al. NeurIPS '19] where an over-parametrised network can fit its training data without learning features, instead performing essentially a type of kernel ridge regression. One way to induce a neural network to behave according to one or the other regime is by scaling the weights of the read-out layer: for a network with $m$ neurons, a scaling of $1/m$ is said to put the network in the mean-field regime, while a scaling of $1 / \sqrt m$ puts the network in the NTK regime.

Here, the authors consider an alternative scaling where the readout of each hidden neuron has its own scaling
\begin{equation}
    \lambda_{m, j} = \gamma \frac 1 m + (1 - \gamma) \frac{\tilde \lambda_j}{\sum_{k=1}^m \tilde \lambda_k}
\end{equation}
where $\gamma \in [0, 1]$ and $1 \ge \tilde \lambda_1  \ge \tilde \lambda_2 \ge \ldots  \ge 0$ are nonnegative fixed scalars that sum to 1 (for example, $\tilde \lambda_j \propto j^{-2}$.

In this setting, the authors consider the limit of large $m$ and make similar assumptions as some of the first papers to discuss convergence of neural networks in the lazy regime (Du et al., 2019b;a), most notably that second-layer weights are first fixed, and then trained in a second phase, separately from the first-layer weights. Under these assumptions, the authors show that
- if γ > 0, the training error goes to 0 at a linear rate with high probability;
- feature learning in a weak sense occurs if and only if γ < 1 (specifically, the authors take feature learning to mean that the representation, i.e. the values of the hidden neurons, has a relative change compared to its representation at initialisation at some point along the training trajectory, for some input x -- their definition does not imply that this change benefits the network in any way.

Some simple experiments suggest that this change in the features is however conductive to generalisation.

**Audience:**

Yes

**Broader Impact Concerns:**

None.

**Claims And Evidence:**

Yes

**Requested Changes:**

1. Please give some hint as to the motivation for considering this regime, as well as for some of the scalings that you give as examples. Is the motivation to have global convergence results with feature learning? If so, what does this type of regime bring to the analysis that is not covered by mean-field methods, which also provide global convergence guarantees?
2. Please add a Discussion of your results at the end of the article - what are the lessons? What kind of analyses do you anticipate will be possible in this regime?

**Strengths And Weaknesses:**

## Strengths

- The idea to consider a per-neuron scaling is original, and leads to some interesting results.
- The analysis appears to be clean (although I didn't check all the details carefully)

## Weakness

- The thing that I am missing most from the article is the motivation and a discussion of the implications of the results.

---

> ### Author Response · Authors · 2024-11-25
>
> We thank the reviewer for their detailed and positive review. We have added a discussion section at the end of the paper, that addresses the points raised by the reviewer: (i) summary of the contributions, motivation for the asymmetrical scaling, and benefits of the non-uniform feature learning, in terms of pruning and transfer learning, (ii) discussion of the choice of the scalings, and (iii) limitations and further possible extensions of the work.

---

### Decision · Action_Editor_3N46 · 2025-02-10

**Recommendation:** Accept as is

**Comment:**

All reviewers agree that the paper is technically sound and provides meaningful theoretical insights into feature learning, which is of interest to the deep learning theory community. Given its solid theoretical contributions and relevance, I conclude that this paper satisfies the acceptance criteria of TMLR.

**Audience:**

This paper could be of interest to parts of the TMLR audience.

**Claims And Evidence:**

This paper investigates the behavior of gradient flow/descent for infinitely wide two-layer neural networks and introduces an asymmetric scaling of neurons controlled by the hyperparameter $\gamma$. The authors prove exponential convergence of the optimization under $\gamma > 0$. In the linear activation case, their analysis shows that the proposed scaling induces both weak and strong feature learning based on their definitions if and only if $\gamma < 1$. Moreover, they demonstrate that feature learning also occurs for nonlinear activations under one-step gradient descent. These theoretical contributions are supported some experiments.